# Evaluating the Evaluators: Investigating LLM Judges for Personalized Writing Style Assessment

## Abstract

We consider the problem of determining how well large language models (LLMs) are able to judge LLM-generated text when a generator is prompted to align with a specific writing style. To illustrate, such an issue is important in a scenario where a user's preferred writing style is known (e.g., "inspirational") and an LLM is used as a judge to evaluate whether generated text adheres to this preference. In this paper, we evaluate performance on two judge tasks: style detection and style quality pairwise ranking. We focus on how the (1) writing task, (2) generator-judge relationship, and (3) general commonsense and reasoning LLM ability impact the judge LLMs performance. To this end, we collected human style detection and pairwise ranking labels on text generated from four models for three generation tasks (email, tweet, and summary writing) that we use to assess LLM judging performance. We find that judge quality correlates strongly with general LLM ability measured using MMLU (Pearson $r = 0.87$), varies by writing task (performance is highest for email by 28%), and is consistent across most judging strategies. We likewise find that LLM evaluators are more consistent and reliable when using AB comparisons rather than rubric-based scoring for style ranking. Finally, we find that for style detection, using the LLM with the strongest general capabilities is best, however this is not true for style quality pairwise ranking, as the strongest models rely on details humans are insensitive to when identifying the better response.

## 1 Introduction

Large language models (LLMs) are used to assist users with a variety of tasks, such as writing (Mysore et al., 2025) or developing software (Liu et al., 2024), and in domains such as healthcare (Mirzaei et al., 2024), government (Aoki, 2025), and academia (Meyer et al., 2023). When working on behalf of a user, a model must be aligned to that user so that its generations contain the correct individual style and principles. For example, if a user's messages typically contain jokes and emojis, then friends might be confused to receive messages lacking these characteristics. Such alignment can be achieved efficiently at inference-time using a *persona*: a description of the user and desired characteristics, which are specified in a prompt (Moon et al., 2024).

However, there are challenges with using personas for inference-time personalization. First, the description of the user traits must be accurate and clearly defined so that the model can understand the persona it is expected to adopt (Li et al., 2025). Second, when an LLM adopts a persona, biases in the model can be exposed and can severely degrade performance (Gupta et al., 2024). Finally, measuring the degree to which an LLM adheres to the persona is important to ensure both the quality and consistency of the style alignment, which we study in this work.

Quality and consistency of adherence to a persona can be measured using another LLM to analyze outputs generated by the persona-driven LLM (Zhou et al., 2025; Lu et al., 2025; Dong et al., 2024) using criteria such as rating the presence of desired attributes (e.g., humorous or formal) (Bhandari et al., 2025; Samuel et al., 2024) or detecting specific styles (Toshevska & Gievska, 2025). However, the ability of LLMs to assess the quality of arbitrary writing attributes distinct from those LLMs are trained for (e.g., helpfulness or harmlessness) has not been explored in depth. That is, can the evaluators reliably evaluate the personalization quality of generations in terms of the adherence to style?

We assess how well eight evaluator LLMs of varying size and from three families can detect the presence of styles and pairwise rank the quality of writing style compliance in LLM-generated text. We collected over 350,000 human and LLM annotations for generations from four LLMs. The human annotations serve as the ground-truth for the evaluator LLM annotations. We evaluate the impact of the scoring scheme used by the evaluator LLMs, the relationship between the generator and evaluator LLM (e.g., same vs. different family of model), and how particular styles impact evaluator performance. We consider five personas (defined as a group of three preferred writing styles) constructed from 15 (individual) writing style elements (e.g., poetic, journalistic), and generations are evaluated for three tasks. We find evaluator performance is strongly correlated (Pearson $r = 0.87$) with LLM ability (general performance measured using

Figure 1: An overview of the process for generating a query response aligned with a persona (Left), evaluating the degree of style-alignment (Top-Right), and then verifying that evaluation with human annotations (Bottom-Right).

MMLU), that evaluators' performance varies by task, and that the relationship between the generator and evaluator LLM can influence performance. Overall we found that GPT4o with a scoring scheme that estimates the probability that styles are present is best able to recover the human annotated styles across writing tasks. For the pairwise style ranking task we found Qwen2.5 14B Instruct with AB comparison performed best, and that performance in terms of ranking quality degrades as LLMs become too strong along dimensions of general commonsense and reasoning capabilities. We conclude that for classification-style tasks, such detecting the presence of styles, using the strongest LLM available is the best choice. However, when ranking responses, LLMs that are too strong rely on details that humans are insensitive to when identifying the better response.

## 2  RELATED WORK

**Using LLMs as Evaluators**    Prior works use LLMs as evaluators to both improve upon traditional evaluation scores, such as BLEU and ROUGE, and to avoid the expense of using human annotators. Zheng et al. (2023) coin the term *LLM-as-a-Judge* and show strong alignment between GPT-4 and human annotators (citing an 85% agreement between the two). Several other works (Liu et al., 2023b; Fu et al., 2024) explore how prompting styles affect the correlation between LLM evaluators and human annotators. Subsequently, Atreja et al. (2025) consider the trade-offs of various prompt designs, such as asking for ranked outputs vs. assigning numerical scores, or asking models to justify their score assignment, and find they mixed results across tasks. We adapt prompts from Zheng et al. (2023) and incorporate findings from Atreja et al. (2025), such as asking for explanations.

**LLM Judges for Instruction Fine-Tuning datasets**    A popular use-case for LLM evaluators is to label instruction fine-tuning datasets as human labeling costs are expensive. Prior work has established that LLMs can serve as effective human surrogates for data annotation tasks where there exists a clear answer, such as relevance and stance detection (Gilardi et al. (2023)), safety (Movva et al. (2024)), or media bias (Horych et al. (2025)). A more complete survey of related work is available from Tan et al. (2024). These works focus on labeling according to a predefined task definition, rather than modifying their labels to align with specific users or preferences.

**Personalization with LLMs**    LLMs have been studied for personalization in contexts ranging from recommendation (Zhang et al., 2025a) to financial advising (Liu et al., 2023a) and more (Chen et al., 2024; Zhang et al., 2025b). We are specifically interested in understanding how effectively LLMs can serve as proxy judges for personalized text generation, where personalization is defined by stylization to align with a user's preferences (e.g., to produce text that exhibits a different style while preserving the content (Hu et al., 2022)). Prior work has applied an LLM-as-Judge framework for measuring personalization (Wang et al., 2023; 2024a; Sun et al., 2024), though these works have neither focused on detecting stylization nor evaluated such a diverse distribution of preferences as ours. While diverse data generation and evaluation have been proposed in works like AlpacaFarm (Dubois et al., 2023), these focus on instruction-following as a task, not on personalization as style-adherence. In these and many other works, personas (Moon et al., 2024) have emerged as an effective and efficient option for personalization without fine-tuning or extensive data collection, and we conduct our investigation by examining a handful of predefined personas for both generation and evaluation.

## 3 Preliminaries

### 3.1 Terminology

Figure 1 provides an overview of the pipeline for assessing LLM evaluators. We begin with a set of fifteen writing styles, $w \in W$, from which a population of five individual user personas, $\mathbf{u}_i \in U$, is formed using combinations of three writing styles $\mathbf{u}_i = \{w_l, w_m, w_n\}$ that describes the stylistic writing preferences of user $\mathbf{u}_i$. The fifteen individual styles and the five combinations of styles are listed in Appendix Figure 12. We include a null-persona, $\mathbf{u}_\emptyset$ for a total of six personas, which specifies no writing specific styles, leaving the LLM to write according to its default writing style.

For a given writing task $t \in \{\text{email, tweet, summary}\}$ and user query $q \in Q$, a generator model, $\text{LLM}_g$, is prompted with the persona $\mathbf{u}_i$, the writing task $t_j$, and user query $q_k$. For the email and tweet tasks the user query is a question to be answered, and for the summary task it is a piece of text to be summarized (see Appendix B for the list of user queries). We denote the prompts for the generator model for a given task as $p_g^t = \{\mathbf{u}_i, t_j, q_k\} \in P^g$, which are listed in Appendix C. Given $p_g^t$, $\text{LLM}_g$ is expected to produce the appropriate response, $r_g$, in terms of content and style. Next, an evaluator model, $\text{LLM}_e$ is prompted to assess $r_g$ according to (1) how much each individual writing style $w \in W$ is present in the text, or (2) how effectively $r_g$ aligns to a given persona $\mathbf{u}_i$ as compared to another generation (as an AB comparison). The specific scoring strategy that $\text{LLM}_e$ uses to assess $r_g$ is included in the corresponding prompt $p_e \in P^e$, which are listed in Appendix C.5. Superscripts $d$ and $r$ are added to $p_e$ to indicate whether $\text{LLM}_e$ is prompted to detect a style in $r_g$, $p_e^d$, or to rank $(r_g^a, r_g^b)$ according to which best aligns with $\mathbf{u}_i$, $p_e^r$. The ground-truth human annotations are denoted $y^d$ and $y^r$ for the style detection and pairwise ranking tasks respectively. The corresponding LLM annotations are indicated by $\hat{y}^d$ and $\hat{y}^r$.

### 3.2 Dataset

Our primary objective is to measure the degree to which evaluator LLMs can reliably assess writing style compliance in LLM generated outputs. To achieve this, we created a collection of LLM generated writing samples with corresponding human annotations to serve as ground-truth against which $\text{LLM}_e$ decisions can be evaluated. The complete user-study design is provided in Appendix F.

We first construct a set of prompts for $\text{LLM}_g \in \{\text{OLMo-2-1124-13B-Instruct (OLMo et al., 2024), Qwen2.5-14B-Instruct (Team, 2024), Claude-sonnet-3.7 (Anthropic, 2025), and o4-mini-2025-04-16 (OpenAI, 2025)}\}$ that relate to three writing tasks $t \in \{\text{emails, tweets, and summarization}\}$. For each $t$, we construct 10 queries $q_k$ (either questions or text snippets to be summarized). $\text{LLM}_g$ is then given each of our six personas (Figure 12), $\mathbf{u}_i$ (the five style conditioning personas and $\mathbf{u}_\emptyset$ that we use for convenience to define no style conditioning), paired with each query $q_k$ for 6 personas × 10 queries = 60 prompts, $p_g$, per writing task. We then generate responses $r_g$ by repeatedly sampling responses from $\text{LLM}_g$ until we have 10 diverse responses for each query, resulting in 600 generations per writing task per $\text{LLM}_g$, for a total of 600 responses × 3 tasks × 4 LLMs = 7200 responses. We provide complete details in Appendix E.

#### 3.2.1 Obtaining Style Detection Labels

Labels that indicate how well the responses $r_g$ adhere to the styles in $\mathbf{u}_i$ are obtained from over 1000 annotators on a crowd-sourcing platform. For the style detection task, each $r_g$ is annotated 10 times for each of the 15 writing styles resulting in > 1 million total annotations. Following Chhun et al. (2022), annotators are asked to rate the adherence of $r_g$ to each $w \in W$ on a three-point scale (1="not present", 2="somewhat present", 3="strongly present"—exact wording depending on the queried style), and the realism of $r_g$ in terms of how likely is $r_g$ human generated (also on a three-point scale). The mean and standard deviation of the inter-annotator agreement across all annotation tasks is relatively low (0.48 ±0.1) — see Section 3.3 for a detailed analysis. This low agreement highlights the subjective nature of assessing style and indicates the expected difficulty of this task for LLMs. The (ground-truth) label $y_m^d$ assigned to each response is the majority vote over the individual labels $y_i^d \in [1, 30]$.

#### 3.2.2 Obtaining Pair-Wise Style Ranking Labels

To construct the samples for pairwise style ranking, we use labels $y^d$ to sub-select two responses $r_g^a$ and $r_g^b$ per $\text{LLM}_g$, and per writing task and query pair $p_g^t$. The responses are selected by computing the compliance with $\mathbf{u}_i$ by computing the mean majority vote $y^d$ for the three writing styles $w$ in the persona $\mathbf{u}_i$ and then selecting the samples with the strongest and weakest compliance. For the $\mathbf{u}_\emptyset$, the two responses are sampled randomly. The samples in each response pair share the same $\text{LLM}_g$, writing task $t$, and user query $q$, and are constructed both within and across personas $\mathbf{u}$. Therefore, each response associated with a given persona $\mathbf{u}_i$ is paired with the other response from the same persona,

and with responses from all five other personas (including $\mathbf{u}_\emptyset$). This results in 66 response pairs constructed from 12 responses, and a total of 4 LLMs $\times$ 3 tasks $\times$ 10 prompts $\times$ 66 response pairs = 7,920.

For the pairwise style ranking task, each pair of responses $(r_g^a, r_g^b)$ was ranked by human annotators based on the alignment with each persona $\mathbf{u}_i$. For $\mathbf{u}_\emptyset$, the responses were ranked according primarily according to helpfulness (Wang et al., 2024b;c; Dubois et al., 2023). The instructions are detailed in Appendix G. Each pair of responses was ranked by five annotators $y_i^r, i \in [0, 5]$ and the final label $y_m^r$ was assigned based on the majority vote.

### 3.3 Trends in the Human Annotation Labels

#### 3.3.1 Inter-annotator Agreement in Style Detection

Randolph's multi-rater kappa $\kappa_{\text{free}}$ (Randolph, 2005) was used to quantify inter-annotator agreement, as implemented in `statsmodels` (Seabold & Perktold, 2010). This free-marginal implementation of the kappa was chosen to match our experimental design; specifically, because the annotators did not know a priori how many of the annotated samples belonged to each style level (see discussion in Randolph (2005)).

Inter-annotator agreement is in Table 1, broken up by style and task. We observe moderate-to-high agreement ($\kappa_{\text{free}} > 0.4$) in approximately half of the styles in email writing and two-thirds of the styles in tweet writing and summarization, and fair agreement in the rest ($0.4 \geq \kappa_{\text{free}} > 0.2$), as per the interpretation provided in Landis & Koch (1977). "Rhyming and rhythmic" and "Poetic & Lyrical" were the styles with most agreement (least subjectivity) across writing tasks, likely because it is relatively easy to determine if something is rhyming and poetic. Conversely, "Scholarly-yet-friendly" and "Visual and spatial" had most disagreement among annotators, likely because these styles may not lend themselves naturally to the writing tasks, and "scholarly" and "friendly" could be rated as less likely to co-occur. In general, agreement tended to be higher for summaries compared to emails and tweets. The consistency between annotators when scoring "Realism" was only $\kappa_{\text{free}} \approx 0.23$ suggesting that the annotators disagree on the extent the generations appear to be written by a person.

#### 3.3.2 The Effect of Style Conditioning on Human Detected Styles

Human annotations, $y^d$, show that when a style is specified in $p_g^t$, the probability of observing that style as "strongly present" is relatively high, and generally higher than when no style conditioning is used, see Figure 14. To illustrate, when "playful and whimsical" is specified in $p_g^t$, the probability of the corresponding responses having this style labeled as strongly, somewhat, and not present is $p_{y^d=3,2,1} = \{0.78, 0.14, 0.08\}$ respectively. This suggests that generators are producing output aligned with the styles specified in the prompt, but Figure 14 suggests there are some styles (e.g., "rich descriptions" and "sensory-focused") for which generators have less success with adhering to because either "not present" is the most probable label or the probabilities are relatively uniformly distributed across the labels. That said, style conditioning does seem to have an impact when compared to annotations collected without style conditioning in $p_g^t$, or $\mathbf{u}_\emptyset$. The probability of "not present" decreases in all styles but "Scholarly-yet-friendly" and "Journalistic" when a style is specified (see Appendix Figure 14 A-E vs. F). Additionally, certain styles appear correlated; for instance, when "Playful and whimsical" is used in conditioning the LLMs, "Poetic and lyrical" also has a relatively high mean annotation score (see Figure 13 in Appendix).

Table 1: Inter-annotator agreement in style detection according to Randolph's (2005) multi-rater kappa ($\kappa_{\text{free}}$) per style for each of the three writing tasks across all four generators. Moderate-to-high agreement was observed in approximately half of the styles, with others showing only fair agreement, as interpreted according to Landis & Koch (1977). *Realism was not used in instructing the generators

| Style | Email | Tweet | Summary | Style | Email | Tweet | Summary |
|---|---|---|---|---|---|---|---|
| Scholarly-yet-friendly | 0.23 | 0.28 | 0.45 | Visual & Spatial | 0.23 | 0.30 | 0.42 |
| Encouraging & Supportive | 0.29 | 0.45 | 0.53 | Rich descriptions | 0.34 | 0.49 | 0.42 |
| Legal precision | 0.34 | 0.44 | 0.51 | Storytelling | 0.46 | 0.49 | 0.37 |
| Inspirational & Uplifting | 0.35 | 0.52 | 0.53 | Sensory-focused | 0.40 | 0.49 | 0.52 |
| Telegraphic brevity | 0.43 | 0.35 | 0.63 | Playful & Whimsical | 0.54 | 0.49 | 0.54 |
| Journalistic | 0.56 | 0.67 | 0.46 | Step-by-step instructional | 0.47 | 0.56 | 0.66 |
| Robotic & Emotionless | 0.57 | 0.58 | 0.63 | Poetic & Lyrical | 0.61 | 0.63 | 0.58 |
| Rhyming & Rhythmic | 0.66 | 0.67 | 0.64 | Realism* | 0.24 | 0.21 | 0.23 |

## 4 EXPERIMENTS

We next describe experiments conducted to measure: (1) the consistency (Section 4.1.1) and the reliability (Section 4.1.2) of $LLM_e$ predictions, (2) the sensitivity of the evaluators to the scoring strategy used, (3) the effects of different relationships between $LLM_e$ and generator ($LLM_g$) models, and (4) the effect on $LLM_e$ when task-irrelevant information is included in the persona.

For each query, $q_r$, for style detection and $(r_q^a, r_q^b)$ for pairwise style ranking, the $LLM_e$ ($LLM_e \in$ {OLMo-2-1124-7B-Instruct, OLMo-2-1124-13B-Instruct, OLMo-2-0325-32B-Instruct (OLMo et al., 2024), Qwen2.5-7B-Instruct, Qwen2.5-14B-Instruct, Qwen2.5-32B-Instruct (Team, 2024), o4-mini-2025-04-16 (OpenAI, 2025), and gpt-4o-2024-11-20 (OpenAI, 2024)}) is prompted with $p_e^d$ or $p_e^r$ 30 times with a relatively high temperature (0.7). We consider the variation in $LLM_e$'s $\hat{y}^d$ or $\hat{y}^r$ as an estimate of the consistency of the model in terms of its prediction, and indirectly the certainty of the model in terms of its output. Although there are several ways that this could be measured, we use the outputs rather than the internal states to remain consistent with the closed models, like o4-mini-2025-04-16. We use the multiple $\hat{y}$ to measure how reliable the the $LLM_e$ are, and to improve the quality of our $\hat{y}$ as found in Yuan et al. (2024). We use majority-vote aggregation to collapse all 30 labels into a single label per query per model, described in detail below.

### 4.1 WRITING STYLE DETECTION

Each evaluator model is used to rate each generator ($LLM_g$) using four different detection prompts $p_e^d$: (1) **binary detection**: an indicator specifying whether $w \in W$ is present in $r_g$ or not ({yes, no}), (2) **probability estimation**: an estimate of the probability that $w \in W$ is present in $r_g$ ($[0-1]$), (3) **Likert-3**: a rating of ({"not present", "somewhat present", or "very present"} signaling the degree to which $w \in W$ is in $r_g$, and finally (4) **Likert-10**: a numerical rating ($[1-10]$) signaling the degree to which $w \in W$ is in $r_g$. We explore multiple style detection scoring strategies to understand how soliciting different types of labels (i.e. categorical, rating, and probability) impact the ability of $LLM_e$ to accurately detect the presence of the styles $W$. To compare these scores to the binarized human annotations[1], they are each binarized and mapped to "not present"[2] or "present"[3]. The specific details for how binarization and these mappings were selected are provided in Appendix C.4.

#### 4.1.1 $LLM_e$ SELF CONSISTENCY

To measure the consistency of $LLM_e$, we use Randolph's kappa (Randolph, 2005) to quantify inter-annotator agreement across all LLMs used in this work, using the same setup described in Section 3.3.1. Consistency between all LLMs for each task is reported in Table 2, and we provide full details of each $LLM_e$'s self-consistency in Appendix I. For all detection prompts $p_e^d$, self-consistency is scored based on the binarized $LLM_e$ label rather than directly on the unprocessed label. Nevertheless, we observe that **fewer labeling options leads to higher internal consistency**, as evidenced by the bi-

Table 2: The consistency (Randolph's $\kappa$) across all $LLM_e$ and $p_e^d$ for each task. Most models show high consistency for binary detection and Likert-3 ratings, and become less consistent when more choices are available.

| Score | Email | Tweet | Summary |
|---|---|---|---|
| Binary Detection | 0.92 | 0.92 | 0.93 |
| Likert-3 | 0.85 | 0.80 | 0.80 |
| Likert-10 | 0.57 | 0.53 | 0.55 |
| Probability Estimation | 0.67 | 0.64 | 0.69 |

nary detection and Likert-3 prompts scoring much higher self-consistency than the Likert-10 and probability estimation scores. This trend holds across all tasks and models. Finally, we observe that Qwen and GPT models exhibit generally high consistency across tasks, while OLMo is less consistent in every task.

#### 4.1.2 $LLM_e$ STYLE DETECTION PERFORMANCE

**Set Up** To measure how well the $LLM_e$ are able to detect the presence of our $W$, we use our majority-vote and binarized $LLM_e$ labels. We refer readers to Appendix H for complete details. We also consider style detection with *reflection*, in which we prompt $LLM_e$ to reflect on its answer and give the LLM an opportunity to change or to maintain its original answer. Results for the reflection experiments are available in Appendix J.3. F1-scores below are computed using the human-assigned labels as the ground truth.

---

[1]We cast the three-point labels to binary labels because both "somewhat present" and "very present" indicate some degree of presence of the style as detected by human annotators, and the distinction between them is somewhat arbitrary and subjective.

[2]{"no", [0 - 0.49], [0 - 4] "not present"} → "not present"

[3]{"yes", [0.5 - 1.0], [5 - 10], "somewhat present", "very present"} → "present"


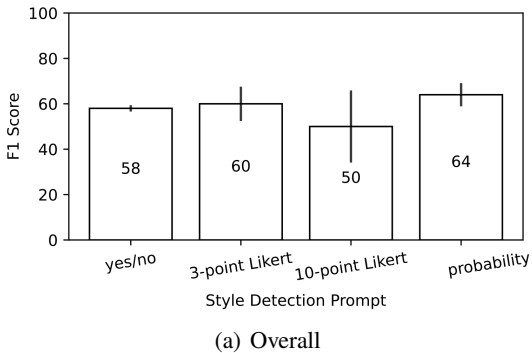
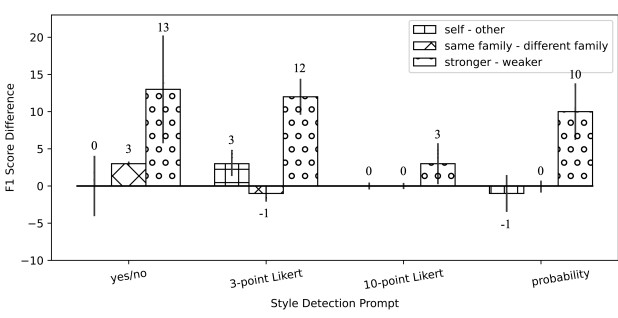

(a) Overall
(b) By user-query response versus judge LLM relationship

Figure 2: **All Writing Tasks:** The performance (F1-score) for each scoring strategy for the $LLM_e$ models on the across all writing task: (a) across all writing styles, generators and evaluators and (b) broken down by the relationship between the response and evaluator LLMs (LLMs relationship). The standard deviation in F1-score is computed across writing tasks, where the mean per writing task is computed over writing styles, generators, and evaluators.

**Results** We evaluate the ability of the judge LLMs ($LLM_e$) to recover the binarized labels assigned by humans. We examine the impact of our various data dimensions on the LLM style-detection F1-score. We find that the probability estimation $p_e^d$ performs best across $LLM_e$ and the writing tasks (see Figure 2(a)). However, the standard error bars overlap between Likert-3, Likert-10, and probability estimation. We break the F1-score analysis down by writing task in Appendix Figure 19(a), and find that writing task greatly impacts the performance generally, and the performance of $p_e^d$ specifically, with Likert-10 performing worse than random for tweet and summary despite being the top-performer for email. For a more detailed analysis on the per-task performance, see Appendix J.2.

Analyzing the relationship between the writing task $t$, $LLM_e$, and the style detection scoring scheme $p_e^d$ in detail (Appendix J.1), we find that across $LLM_e$ and $t$, **probability estimation is most frequently the best performing style detection prompt** (and never the worst performing) and **GPT4o is the top-performing $LLM_e$** (Appendix Table 21). However, Qwen2.5-32B-Instruct frequently performs on par with GPT4o (e.g. F1-scores = 70 vs. 72 for email and tweet, 68 vs. 68 for summary), suggesting it is a strong open weight alternative. Qwen2.5-32B-Instruct frequently outperforms GPT4o-mini as well. We find that the family of OLMo LLMs generally under-perform the families of Qwen and GPT LLMs. While GPT4o is the top-performing $LLM_e$ for all writing tasks $t$, there is some variability in the top-performing $p_e^d$: probability estimation for email and tweet, and binary detection for summary. We find that performance on **the email writing task is least sensitive to the exact style detection scoring scheme** with no significantly best-performing scheme, unlike tweet and summary. This relative lack of sensitivity to $p_e^d$ for the email writing task may be due to the email writing task imposing the least amount of structure on the text and allowing for the greatest freedom of style expression, whereas tweets have length limits and summaries can be influenced by the tone and structure of the original text. This finding is complemented by the result in Appendix Figure 19(a) showing that the $LLM_e$ were best able to detect the target styles for the email writing task (F1-score= 72 versus 62 and 59). Finally, we confirm the performance drop observed in Likert-10 $p_e^d$ for the tweet and summary tasks is consistent across $LLM_e$, and is not a quirk from a subset of outlier LLMs decreasing performance.

**We find a strong correlation between the ability of the $LLM_e$ to solve the style detection task and its general commonsense and reasoning abilities**[4](Pearson-r=0.87) across all writing tasks and $p_e^d$. We see a stronger positive correlation (Pearson-r $\in \{0.94, 0.96\}$) for each writing task. However, when looking at the correlation within a $p_e^d$, we find that style detection ability is more strongly correlated with general LLM ability for some $p_e^d$ than others: binary detection $r = 0.93$, Likert-3 $r = 0.81$, Likert-10 $r = -0.46$, and probability estimation $r = 0.76$. This supports that some $p_e^d$, Likert-10, are poorly suited to the task. It is interesting to note, the best performing $p_e^d$ (probability estimation) has one of the weaker correlations to commonsense reasoning performance.

In Figure 2(b) we look at how style detection performance is impacted by the relationship between $LLM_g$ and $LLM_e$. Specifically we look at the difference in F1-scores when the LLMs are the same versus different (self -- other), when the LLMs come from the same versus different LLM families (same family -- different family) (e.g. both are Qwen models versus one is a Qwen model and the other an OLMo model), and when $LLM_e$ is stronger than the $LLM_g$ (stronger -- weaker) (e.g. Qwen2.5-Instruct-32B vs. Qwen2.5-Instruct-13B). We find that the

---

[4]Measured with MMLU score (Hendrycks et al., 2021).

impact of the relationship between $LLM_g$ and $LLM_e$ depends on the style detection scoring strategy and the writing task $t$.

The only common trend across $p_e^d$ and the writing tasks is, using a stronger $LLM_e$ than $LLM_g$ performs better. However, the impact is small for several combinations of $p_e^d$ and $t$. Therefore, **we recommend using a stronger LLM as the $LLM_e$ while quantifying the gains to appropriately balance cost and style detection quality.**

**Understanding the Effects of Additional Persona Attributes on Style Detection Performance.** Following Gupta et al. (2024) we explored the impact on style detection performance of including protected attributes in the persona since it was shown previously that protected attributes can have a detrimental effect on task performance. Specifically LLM performance at tasks such as coding and Q&A degraded significantly, and the amount of degradation depended on the specific values assigned to the protected attributes. In this self-contained case study on the email-writing task using the Likert-3 scoring scheme, we find that including protected attributes in the persona system prompt had little effect on performance in terms of the F1-score. For more detailed results, see Appendix J.4.

## 4.2 PAIRWISE STYLE RANKING

Table 3: The consistency (Randolph's multi-rater kappa) of $LLM_e$ ratings $\hat{y}^r$ when annotating responses $r_g$ to be able to rank which response in a pair $(r_g^a, r_g^b)$ better reflect the described attributes (i.e. writing style elements).

| Score | Email | Tweet | Summary |
|-------|-------|-------|---------|
| AB | 0.85 | 0.87 | 0.87 |
| Rubric | 0.44 | 0.44 | 0.43 |

Table 4: The performance of the different pairwise style ranking prompt $p_e^r$ by the style ranking task. Results are reported as the mean (standard error) F1-score across $LLM_e$.

| | Score | Helpfulness Targeted | Style Targeted |
|---|-------|----------------------|----------------|
| Email | AB | 53.84(±3.94) | Yes |
| | Rubric | 49.04(±2.20) | Yes |
| Summary | AB | 50.78(±2.8) | 60.35(±1.27) |
| | Rubric | 39.11(±1.09) | 46.21(±2.01) |
| Tweet | AB | 55.7(±1.3) | 62.93(±0.8) |
| | Rubric | 46.53(±1.4) | 49.26(±1.8) |

Using the same set of $LLM_e$ as in Section 4.1, the paired responses $(r_q^a, r_q^b)$ outlined in Sect 3.2.2 are assigned labels $y^r$ for two versions of the pairwise ranking task: **helpfulness targeted** and **style targeted**. In the helpfulness-targeted version of the task, the $LLM_e$ identifies which of two responses better meet a set of response quality criteria focused on helpfulness and answer completeness. This type of judge prompt is commonly used when deploying LLM judges to augment or replace human preference labelers (Yuan et al., 2024; Dubois et al., 2023; 2024; Cui et al., 2024). This is the setting for which prompt-based judging has received the most human evaluation (Dubois et al., 2023; 2024). In the style-targeted version of the task, the $LLM_e$ assesses which of two responses $r_q$ better complies with the writing style of a given persona $\mathbf{u}_i$.

For each version of the pairwise style ranking task, different prompting approaches are explored: (1) **rubric-based** where the $LLM_e$ assigns a score to each response based on how well each response meets the criteria outlined in the given rubric (see Table 12 Appendix C.5), and (2) **AB comparison** where the $LLM_e$ is presented with both responses $(r_q^a, r_q^b)$ and selects "a" or "b" based on which better aligns with the given criteria (see Table 11 Appendix C.5). For the rubric-based $p_e^r$, the $r_q$ with the higher score is selected – if $r_q^a$ has a higher score than $r_q^b$ then the label is "a", if $r_q^b$ has the higher score then the label is "b", and if $r_q^a$ and $r_q^b$ have the same score then the label is a "tie". For the AB comparison $p_e^r$, the $LLM_e$ compares $r_q^a$ and $r_q^b$ in both the AB and BA orders with a "tie" occurring whenever different responses are selected given the different orders.

### 4.2.1 $LLM_e$ SELF CONSISTENCY

The consistency in the pairwise ranking labels from the each $LLM_e$ is measured using Randolph's multi-rater kappa ($\kappa_{\text{free}}$) computed over the repeated samples ($n = 30$) using the setup described in Section 3.3.1. Consistency between the $LLM_e$ for each writing task $t$ and pairwise style ranking prompt $p_e^r$ is reported in Table 3, and a full breakdown of per $LLM_e$ self consistency is provided in Appendix K . We find that the $LLM_e$ are more internally consistent when using the AB versus rubric $p_e^r$ and that the writing task $t$ has little to no impact on self consistency. Examining Appendix Table 26, we can see that, with the exception of the OLMo family of models, for the AB $p_e^r$ and within a writing task, the majority of $LLM_e$ have similarly high levels of self consistency. However, for the rubric $p_e^r$, the amount of self consistency steadily increases as the $LLM_e$'s general abilities improve, which suggests the rubric $p_e^r$ is a more challenging version of the pairwise ranking task.

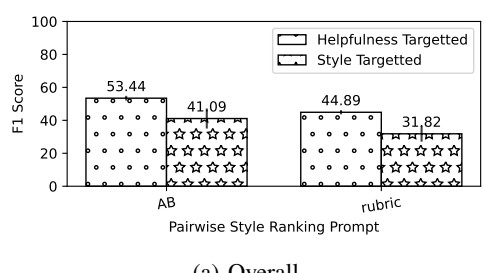
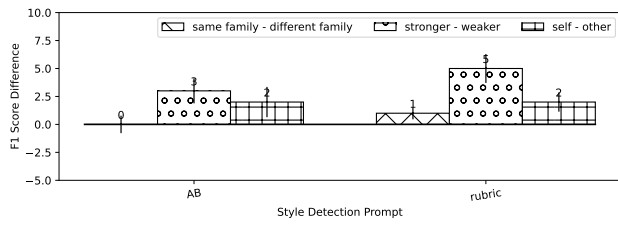

(a) Overall

(b) By user-query response versus judge LLM relationship

Figure 3: **Per Writing Task:** The performance (F1-score) for each pairwise style ranking $p_e^r$ and pairwise style ranking task across LLM$_e$ on each writing task: (a) across all writing styles, generators and evaluators and (b) broken down by the relationship between the LLM$_g$ and LLM$_e$ (LLMs relationship).

### 4.2.2 LLM$_e$ PAIRWISE STYLE RANKING PERFORMANCE

**Set Up**   For the LLM$_e$, the pairwise ranking task is a three-class classification task with "a", "b", and "tie" are the possible labels $\hat{y}^r$. However, for the human annotators it is a five-class problem: "a is better", "a is slightly better", "tie", "b is slightly better", and "b is better". To be able to compare the LLM$_e$ labels, the human labels "a is better" and "a is slightly better" are mapped to "a", and "b is slightly better" and "b is better" are mapped to "b". For the rubric-based $p_e^r$, the multiple $\hat{y}^r$ are consolidated by taking the mean $\hat{y}^r$ across the repeated generations following Yuan et al. (2024) and then comparing the $r_q^a$ score to the $r_q^b$. For the AB comparison $p_e^r$, the majority vote $\hat{y}^r$ is applied after consolidating $\hat{y}^r$ across the AB and BA response orderings.

**Results**   We evaluate the ability of the LLM$_e$ to recover the pairwise style ranking labels $y^r$ assigned by humans. Examining the impact of the pairwise ranking task (helpfulness versus style) and the scoring strategy (AB $p_e^r$ versus rubric $p_e^r$) on the pairwise ranking performance (F1-score) in Figure 3(a), across LLM$_e$ and writing tasks $t$ the **AB $p_e^r$ leads to better performance for both the helpfulness (F1-score≈ 41 vs. ≈ 32) and style-targeted (F1-score=≈ 53 vs. ≈ 45) pairwise ranking tasks**. The standard errors bars do not overlap across $p_e^r$ scoring strategies indicating the performance is not sensitive to the specific writing task, which is upheld when looking at the per writing task results in Figure 23 Appendix L. Additionally, the standard error reported in Table 4 shows that across LLM$_e$ is more consistent (i.e., lower average standard error) for the AB $p_3^r$. Across scoring strategies, the style-targeted pairwise ranking task (F1-score≈ 49) is easier for the LLM$_e$ than the helpfulness-targeted task (F1-score≈ 37). Even when examining the results broken down by writing task $t$ and LLM$_e$ in Appendix L.1, the AB $p_e^r$ out performs rubric and the LLM$_e$ performance is higher for the style-targeted than the helpfulness-targeted pairwise ranking task. However, we do find there are some personas $\mathbf{u}_i$ for which the LLM$_e$ performs worse on than the helpfulness-targeted pairwise ranking task. For more details on how each persona $\mathbf{u}_i$ impacts LLM$_e$ pairwise ranking on the different writing tasks, see Appendix L.2. Examining Table 4 and the standard error bars in Figure 3(a) we see that performance is consistent across writing tasks.

Per LLM$_e$ performance is reported in Appendix L.1, where we find that all LLM$_e$ are able to solve the pairwise style ranking task at greater than random chance ($\approx 33$). However, multiple LLM$_e$ (5 / 8) barely exceed the random chance threshold on the summary writing task using the rubric $p_e^r$ scoring strategy. Analyzing the relationship between the writing task $t$, the LLM$_e$, the pairwise ranking task type, and the scoring strategy $p_e^r$ in detail (Appendix L.1), we find that **the best performing LLM$_e$ is dependent on the writing task and the pairwise ranking task**. In all cases the AB $p_e^r$ outperforms the rubric $p_e^r$. For the style-targeted pairwise ranking task, Qwen2.5 14B Instruct is the best performing, whereas for the helpfulness-targeted task either OLMo2 32B Instruct or GPT4o is the best performing. Examining the per LLM$_e$ and per pairwise ranking task performance in Figures 24 – 25, we see that for the style-targeted pairwise ranking task, performance increases until the LLM$_e$ reach a middling point in terms of commonsense and reasoning capabilities (e.g. $75 \geq 80$ on MMLU (Hendrycks et al., 2021)) after which the pairwise ranking performance drops. For instance, OLMo2 7/13B Instructs have similar performance to GPT4o-mini while Qwen2.5 7B instruct and OLMo2 32B Instruct have similar performance to GPT4o despite very different MMLU scores (e.g., 75 versus 88). This leads us to conclude that **those LLMs with the "strongest" general commonsense and reasoning abilities, such as GPT4o, are not always the best suited to serve as human proxies on tasks that involve comparing and choosing between two pieces of text.**

To better understand what might be leading to this inverted U-shaped relationship between the general commonsense and reasoning capabilities and the style-targeted ranking task, we examine the LLM$_e$ frequency of each label ("response

a", "response b", and "tie") and compare to the humans' frequencies. A detailed breakdown by writing task $t$, $LLM_e$, scoring strategy $p_e^r$, and pairwise ranking task is in Appendix L.3 Figures 17 and 30. We find that humans assign "tie" labels 19% – 27% of the time while assigning a similar proportion of "response a" labels as "response b" labels. We note that with the rubric $p_e^r$ the $LLM_e$ almost never assign "tie" labels, which we attribute its poor performance relative to the AB $p_e^r$ to. Looking at the trend in "tie" label frequencies across $LLM_e$, we find that the "tie" label's frequency decreases as the $LLM_e$'s general commonsense and reasoning capabilities increase – the weakest $LLM_e$ (i.e., MMLU $\leq 75$) assign the most "tie" labels while the strongest $LLM_e$ (i.e., MMLU $> 80$) assign the least. This result suggests the "weaker" $LLM_e$ are not able to handle the complexity of the labelling prompt leaving their labels sensitive to the AB versus BA response prompt ordering (i.e. which response is listed first). Whereas for the "strongest" $LLM_e$ the result suggests they rely on differences between the two responses $r_q^a$ and $r_q^b$ that humans are insensitive to, either because the humans did not detect the difference or identified them as irrelevant. Therefore, **we conclude it is important to use LLMs that are strong enough to be insensitive to the influence of prompt ordering, but not so strong they differentiate between responses at a level of detail exceeding that of humans. We find these models are not those with the strongest general commonsense and reasoning capabilities.**

For the helpfulness-targeted task, the relationship between general commonsense and reasoning capabilities versus performance on the pairwise ranking task is writing task dependent (Appendix L.3 Figures 17 and 30). On the tweet task the relationship matches that of the style-targeted ranking task, whereas for summary we see a steady increase in pairwise ranking performance as general commonsense and reasoning capabilities improve. We attribute this to the heavy use of helpful as a guiding principle in the RLHF labelling process as exemplified by the publicly available labelling instructions (Yuan et al., 2024; Dubois et al., 2023; 2024; Cui et al., 2024).

To further understand the relationship between general commonsense and reasoning capabilities and pairwise ranking performance, we measure the correlation between the two. For the AB $p_e^r$ we find a Pearson-r= 0.41 on the style-targeted pairwise ranking task (Pearson-r= 0.42 for tweet and $r = 0.51$ for summary) and Pearson-r= 0.46 on the helpfulness-targeted ranking task (Pearson-r= 0.41 for tweet and $r = 0.75$ for summary). For the rubric $p_e^r$ the style-targeted pairwise ranking task has Pearson-r= 0.73 (Pearson-r= 0.85 for tweet and $r = 0.69$ for summary) and a Pearson-r= 0.49 for helpfulness-targeted (Pearson-r= 0.85 for tweet and $r = 0.53$ for summary). The weak correlation for the style-targeted pairwise ranking task for the AB $p_e^r$ aligns with the per $LLM_e$ pairwise style ranking performance discussed above and observed in Appendix L.1 Figures 24 and 25. Additionally, the difference in correlation strength for the on the helpfulness-targeted pairwise ranking task per writing task aligns with the trends discussed above. For the rubric $p_e^r$ we see a strong correlation across board speaking to increased difficulty of scoring with a rubric versus directly selecting a preferred response.

Finally, in Figure 3(b) we examine the impact the relationship between the $LLM_g$ and the $LLM_e$ has on the pairwise ranking performance. We see similar trends between the AB and rubric $p_e^r$ in that using a $LLM_e$ that is stronger[5] improves performance as does using a $LLM_e$ that is different from the $LLM_g$. We see that using a $LLM_e$ in the same versus a different family as the $LLM_g$ has no impact on performance for the AB $p_e^r$, but for the rubric $p_e^r$ using a $LLM_e$ from the same family is better. Overall, **the strongest performance difference stems from using a $LLM_e$ that is stronger than the $LLM_g$.** For a detailed breakdown by writing task $t$ and pairwise ranking task, see Appendix L.5.

## 5 CONCLUSIONS

We investigated the ability of LLMs able to act as judges when labelling text for subjective attributes (i.e., styles) and rating the quality of LLM generated text (i.e., according to a specific writing style or for general helpfulness). To this end we investigated a number of factors, including the relationship between the source LLM that generated the text and the LLM evaluator, the types of writing styles that generators are asked to conform to, the type of scoring strategy that the evaluator LLM is asked to rate against, and the impact of the general commonsense and reasoning capabilities of the evaluator LLM. We first ran a user study to obtain ground-truth labels in the form of human judgments of style adherence, and then ran a number of evaluator models and tested for consistency in the LLM labels and the degree with which the LLM labels aligned with the human labels. Overall we find that LLMs are consistent in their ratings, that the ability of a model to rate text is dependent on the writing task and the particular scoring strategy used to rate the quality. Based on the best performing models for the style detection and pairwise ranking tasks, we conclude that for classification-style tasks (i.e., detecting the presence of styles) using the strongest LLM available is the best choice. However, when ranking responses, LLMs that are too strong (according to general commonsense and reasoning capabilities) rely on details humans are insensitive to when identifying the better response.

---

[5]Measured according to MMLU Hendrycks et al. (2021)

## 6 ETHICS STATEMENT

All human participants were recruited and the annotation protocol was constructed in accordance with our IRB guidelines. The data presented to our participant population was selected to be neutral (e.g., not reflective of politics, religion, etc) and to avoid annotators having to read and evaluate potentially toxic and harmful content.

## 7 REPRODUCIBILITY STATEMENT

We provide all prompts used to create the LLM generated data used in our study in Appendix B, and we provide all LLM evaluation prompts in Appendix C and Appendix D. Specific model identifiers are provided in Section 3.2 of the main paper. Additionally, we provide a detailed breakdown of the full dataset in Appendix E. While the generations and labels may not be completely reproducible due to our temperature-based sampling, we used majority-vote aggregation to ensure repeatable labeling results. Overall, we provide sufficient detail for future work to replicate our results.

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

## A   LLM Usage

The authors utilized LLMs in this research to provide feedback and suggest alternatives for potential prompts, preference sets, and user queries. None of the evaluation LLMs were used to provide feedback on the prompts that were used for the generations.

## B   User Queries

Table 5: The questions used as topics in the email-writing and tweet generations.

| Topics | Questions |
|---|---|
| Practical / Automotive | How do you change a car tire? |
| Biology | What are the structural differences between a plant cell and an animal cell? |
| Physics | What is the relationship between mass, force, and acceleration? |
| Economics | How is the interest rate of a mortgage determined? |
| Cooking / Food | What ingredients are needed to make bread? |
| History | What were the primary goods traded along the Silk Road? |
| Gardening / Home | How do you care for a houseplant? |
| Psychology | What is cognitive dissonance? |
| Games / Recreation | What are the basic rules of tic-tac-toe? |
| Mathematics | What is a prime number? |

Table 6: The dataset IDs and associated Sub-Reddits from the SLF5K dataset (Scheurer et al., 2024) that were used for our summarization task. While the dataset is publicly available, we do not include the posts here for privacy reasons.

| Sub-Reddit | SLF5K Data ID |
|---|---|
| r/relationships | t3_3291e0 |
| r/tifu | t3_2ha5ll |
| r/personalfinance | t3_4ftgyf |
| r/needadvice | t3_21mfop |
| r/jobs | t3_2ruwyj |
| r/AskReddit | t3_27xmjc |
| r/dogs | t3_4kql83 |
| r/college | t3_2slh7g |
| r/books | t3_2anheo |
| r/legaladvice | t3_1ajctx |

---

[5]As of September 23rd, 2025 the SLF5k dataset was released under an Apache 2.0 license

# C PROMPTS

## C.1 USER-QUERY RESPONSE GENERATION PROMPTS WITHOUT STYLE CONDITIONING

This section contains the prompts that were used to generate the LLM responses without style guidance to the user queries in Appendix B. The prompts used to generate responses to the user queries are in Figure 4 for email, Figure 5 for tweet, and Figure 6 for summary.

---

**Email: No Style Conditioning Prompt**

**System prompt:** You are a helpful assistant.
**User prompt:** "Write an email answering the following question: {user_query}"

---

Figure 4: The **email** writing tasks prompt for generating responses without style guidance. "{user_query}" is where the user query to be answered is added to the prompt.

---

**Tweet: No Style Conditioning Prompt**

**System prompt:** You are a helpful assistant.
**User prompt:** Write a tweet (280 characters or less) answering the following question: {user_query}

---

Figure 5: The **tweet** writing tasks prompt for generating responses without style guidance. "{user_query}" is where the user query to be answered is added to the prompt.

---

**Summary: No Style Conditioning Prompt**

**System prompt:** You are a helpful assistant.
**User prompt:** Summarize the following Reddit post, highlighting the main ideas only: {reddit_post}

---

Figure 6: The **summary** writing tasks prompt for generating responses without style guidance. "{reddit_post}" is where the Reddit post to be summarized is added to the prompt.

## C.2 CANDIDATE STYLE CONDITIONED WRITING PROMPTS

This section contains the set of writing task instruction prompts that were considered for the writing tasks. The candidate writing tasks for email are shown in Figure 7, for tweet in Figure 8, and for summary in Figure 9.

## C.3 SELECTED STYLE CONDITIONED WRITING PROMPTS

This section contains the writing task prompt that was selected per LLM and writing task. For email-writing prompts 2, 3, 2, and 4 were selected for GPT-4o-mini, OLMo 13B, Qwen 14B, and Sonnet 3.7, respectively. For tweet-writing prompts 1, 3, 1, and 4 were selected for GPT-4o-mini, OLMo 13B, Qwen 14B, and Sonnet 3.7, respectively. And lastly for summary-writing, prompt 3 was selected for all models. The prompts were selected following the rubric outlined in Appendix D.

**Email-Writing Prompt 1:  Standard, Third Person Prompt**

**System prompt:**  You have the following writing style preferences:  {preferences}
**User prompt:**   Write an email answering the following question:  {user_query}

---

**Email-Writing Prompt 2:  Descriptive Task Prompt**

**System prompt:**  You are an email generation agent.  Your function is to produce
an email based on a query.  The output must include a subject line, a greeting,
a body, and a closing.  The output's style is determined by the following style
parameters:  {preferences}
**User prompt:**  Generate a complete email that answers the following question:
{user_query}

---

**Email-Writing Prompt 3:  Ultra-Neutral Prompt**

**System prompt:**  Transform the following query into email format.  Apply
formatting rules:  {preferences}
**User Prompt:**  {user_query}

---

**Email-Writing Prompt 4:  Using ``Prefer''**

**System prompt:**  You prefer emails which have:  {preferences}
**User prompt:**  Write an email answering the following question:  {user_query}

Figure 7: The candidate prompts to use to instruct the user-query response LLMs to complete **email** writing task. {...} indicate where string formatting happens.

**Tweet-Writing Prompt 1:  Standard, Third Person Prompt**

**System prompt:**  You have the following writing style preferences:  {preferences}
**User prompt:**    Write a tweet (280 characters or less) answering the following
question:  {user_query}

**Tweet-Writing Prompt 2:  Descriptive Task Prompt**

**System prompt:**  You are a social media post generation agent.  Your function
is to produce a tweet based on a query.  The output must fit within 280
characters.  The output's style is determined by the following style parameters:
{preferences}
**User prompt:**  Generate a tweet that answers the following question:  {user_query}

**Tweet-Writing Prompt 3:  Ultra-Neutral Prompt**

**System prompt:**  Transform the following query into tweet format (280 characters
or less).  Apply formatting rules:  {preferences}
**User Prompt:**  {user_query}

**Tweet-Writing Prompt 4:  Using ``Prefer''**

**System prompt:**  You prefer tweets which have:  {preferences}
**User prompt:**  Write a tweet answering the following question:  {user_query}

Figure 8: The candidate prompts to use to instruct the user-query response LLMs to complete **tweet** writing task. {...} indicate where string formatting happens.

---

**Summary-Writing Prompt 1: Standard, Third Person Prompt**

**System prompt:** You are an assistant who writes personal summaries given articles
based on a user's preferences.
**User prompt:** Reddit Post: {user_query}
Summarize the above Reddit post for a user who prefers the following styles:
{preferences}. Please write a summary of the above Reddit post to address those
specified preferences. Summary:

---

**Summary-Writing Prompt 2: Descriptive Task Prompt**

**System prompt:** You prefer summaries which have: {preferences}
**User prompt:** Summarize the following Reddit post in 3-4 sentences, highlighting
the main ideas only: {user_query}

---

**Summary-Writing Prompt 3: Ultra-Neutral Prompt**

**System prompt:** You are an assistant who helps users write personal summaries
given articles based on their preferences.
**User Prompt:** Given a Reddit post and set of preferences, generate a personalized
summary of around three sentences that matches the user's unique voice and style.
Reddit Post:
{user_query}
User Preferences:
{preferences}
Summary:

---

**Summary-Writing Prompt 4: Using ``Prefer''**

**System prompt:** Transform summaries to match specified style requirements. Your
output must demonstrate clear stylistic differentiation from the original while
maintaining factual accuracy. It must also be concise.
**User prompt:**
REDDIT POST: {user_query}
ORIGINAL SUMMARY: {summary}
STYLE REQUIREMENTS: {preferences}
Transform the summary to strongly exhibit these stylistic characteristics.
The output should be clearly distinguishable from the original in style while
preserving all factual content.
TRANSFORMED SUMMARY:

---

Figure 9: The candidate prompts to use to instruct the user-query response LLMs to complete **summary** writing task.
{...} indicate where string formatting happens.

## C.4 STYLE DETECTION PROMPTS

This section contains the prompts that were used to detection style elements in LLM generated text.

Table 7: **Email Style Detection Prompts:** {...} indicate where string formatting is done with "{style_description}" as the description of the style (e.g. "scholarly-yet-friendly") and "{response}" the text to assess. For all style detection prompts the system prompt is: "You are an expert at detecting the presence of writing styles in passages of text."

| Prompt Name | Prompt |
|---|---|
| yes/no | Below is an email response to a question. Your task is to analyze the text in the email and determine whether the writing style matches {style_description}. Answer with just Yes/No. Prefix your answer with 'Answer: '.
EMAIL TO EVALUATE:
{response} |
| 3-point Likert | Below is an email response to a question. Your task is to analyze the text in the email and determine the degree to which the writing conforms to the writing style {style_description}.
Choose from these labels:
- Does not exhibit
- Somewhat exhibits
- Clearly exhibits
Answer with just one of the provided labels. Prefix your answer with 'Answer: '.
EMAIL TO EVALUATE:
{response} |
| 10-point Likert | Below is an email response to a question. Your task is to analyze the text and rate on a scale of 1 to 10 how well the writing conforms to the style {style_description}. For the rating, a score of 1 means the text does not conform to the style at all, whereas as 10 means the text conforms entirely to the style. Answer with just your estimate of the rating. Prefix your answer with 'Answer: '.
EMAIL TO EVALUATE:
{response} |
| probability estimation | Below is an email response to a question. Your task is to analyze the text and determine the probability that the writing conforms to the style(s) {style_description}. Answer with just your probability estimate as a decimal between 0.0 and 1.0 (where 0.0 = no conformance, 1.0 = perfect conformance). Prefix your answer with 'Answer: '.
EMAIL TO EVALUATE:
{response} |

Table 8: **Tweet Style Detection Prompts:** {...} indicate where string formatting is done with "{style_description}" as the description of the style (e.g. "scholarly-yet-friendly") and "{response}" the text to assess. For all style detection prompts the system prompt is: "You are an expert at detecting the presence of writing styles in passages of text."

| Prompt Name | Prompt |
|---|---|
| yes/no | Below is an tweet response to a question. Your task is to analyze the text in the tweet and determine whether the writing style matches {style_description}. Answer with just Yes/No. Prefix your answer with 'Answer: '. 
 TWEET TO EVALUATE: 
 {response} |
| 3-point Likert | Below is an tweet response to a question. Your task is to analyze the text in the tweet and determine the degree to which the writing conforms to the writing style {style_description}. 
 Choose from these labels: 
 - Does not exhibit 
 - Somewhat exhibits 
 - Clearly exhibits 
 Answer with just one of the provided labels. Prefix your answer with 'Answer: '. 
 TWEET TO EVALUATE: 
 {response} |
| 10-point Likert | Below is an tweet response to a question. Your task is to analyze the text and rate on a scale of 1 to 10 how well the writing conforms to the style {style_description}. For the rating, a score of 1 means the text does not conform to the style at all, whereas as 10 means the text conforms entirely to the style. Answer with just your estimate of the rating. Prefix your answer with 'Answer: '. 
 TWEET TO EVALUATE: 
 {response} |
| probability estimation | Below is an tweet response to a question. Your task is to analyze the text and determine the probability that the writing conforms to the style(s) {style_description}. Answer with just your probability estimate as a decimal between 0.0 and 1.0 (where 0.0 = no conformance, 1.0 = perfect conformance). Prefix your answer with 'Answer: '. 
 TWEET TO EVALUATE: 
 {response} |

Table 9: **Summary Style Detection Prompts:** {...} indicate where string formatting is done with "{style_description}" as the description of the style (e.g. "scholarly-yet-friendly") and "{response}" the text to assess. For all style detection prompts the system prompt is: "You are an expert at detecting the presence of writing styles in passages of text."

| Prompt Name | Prompt |
|---|---|
| yes/no | Below is a summary of Reddit post. Your task is to analyze the text in the summary and determine whether the writing style matches {style_description}. Answer with just Yes/No. Prefix your answer with 'Answer:'.
SUMMARY TO EVALUATE:
{response} |
| 3-point Likert | Below is a summary of a Reddit post. Your task is to analyze the text in the summary and determine the degree to which the writing conforms to the writing style {style_description}.
Choose from these labels:
- Does not exhibit
- Somewhat exhibits
- Clearly exhibits
Answer with just one of the provided labels. Prefix your answer with 'Answer: '.
SUMMARY TO SUMMARY:
{response} |
| 10-point Likert | Below is a summary of a Reddit post. Your task is to analyze the text and rate on a scale of 1 to 10 how well the writing conforms to the style {style_description}. For the rating, a score of 1 means the text does not conform to the style at all, whereas as 10 means the text conforms entirely to the style. Answer with just your estimate of the rating. Prefix your answer with 'Answer: '.
EMAIL TO EVALUATE:
{response} |
| probability estimation | Below is a summary of a Reddit post. Your task is to analyze the text and determine the probability that the writing conforms to the style(s) {style_description}. Answer with just your probability estimate as a decimal between 0.0 and 1.0 (where 0.0 = no conformance, 1.0 = perfect conformance). Prefix your answer with 'Answer: '.
SUMMARY TO EVALUATE:
{response} |

## C.5 PAIRWISE STYLE RANKING PROMPTS

This section contains the prompts that were used to assign a pairwise ranking to two user-query responses $r_g$. Two different approaches to assigning the pairwise ranking were explored: (1) a direct comparison between both responses (AB) (Dubois et al., 2023; Li et al., 2023) and (2) given a rubric, score both responses independently and prefer the response with the higher score (rubric) (Yuan et al., 2024). For each prompt approach, one version instructed the $\text{LLM}_e$ to rate according to compliance with a specific set of writing styles and the other version to rate according to general quality (e.g. "helpfulness").

For both approaches to the pairwise style ranking task, the $\text{LLM}_e$ has access to the query that the $\text{LLM}_g$ response $r_g$ should address. These are formatted into the "{user_query_instruction}" slot with a different instruction per writing task. The writing task specific instructions are given in Table 10.

Table 10: The value used for "{user_query_instruction}" for each writing task for the pairwise style ranking prompts in Appendix Sections C.5.1 and C.5.2.

| Writing Task | {user_query_instruction} |
|---|---|
| Email | Write an email answering the following question |
| Tweet | Write a tweet answering the following question |
| Summary | Write a summary of the following Reddit post |

### C.5.1 AB PROMPTS

Table 11: **Pairwise AB Style Ranking Prompts:** {...} indicate where string formatting is done with "{user_query_instruction}" as the writing task instruction given to accompany the user query, "{user_query}" as the question answered or the Reddit post to summarize, "{style_description}" as the description of the style (e.g. "scholarly-yet-friendly"), and "{output_a}" and "{output_b}" the text to compare. For all style detection prompts the system prompt is: "You are a helpful instruction-following assistant that determines the best model output by selecting the best outputs for a given instruction."

---

**Helpfulness Targetted**

Select the output (a) or (b) that best matches the given instruction. Choose your preferred output, which can be subjective. Your answer should ONLY contain: Output (a) or Output (b). Here's an example:

# Example:

## Instruction:

Give a description of the following job: "ophthalmologist"

## Output (a):

An ophthalmologist is a medical doctor who specializes in the diagnosis and treatment of eye diseases and conditions.

## Output (b):

An ophthalmologist is a medical doctor who pokes and prods at your eyes while asking you to read letters from a chart.

## Which is best, Output (a) or Output (b)?

Output (a)

Here the answer is Output (a) because it provides a comprehensive and accurate description of the job of an ophthalmologist. In contrast, Output (b) is more of a joke.

# Task:

Below is the real task, do not explain your answer, just say Output (a) or Output (b).

## Instruction:

{user_query_instruction}: {user_query}

## Output (a):

{output_a}

## Output (b):

{output_b}

## Which is best, Output (a) or Output (b)?

---

**Style Targetted**

Select the output (a) or (b) that best matches the given writing styles. Choose your preferred output, which can be subjective. Your answer should ONLY contain: Output (a) or Output (b). Here's an example:
# Example:
## Instruction:
Send a coworker a message about your latest meeting.

## Writing Styles:
'Sarcasm'

## Output (a):
There is nothing better for me than a meeting spent watching someone figure out an AV system.

## Output (b):
That meeting went alright.

## Which is best, Output (a) or Output (b)?
Output (a)

Here the answer is Output (a) because it is more sarcastic.

# Task:
Below is the real task, do not explain your answer, just say Output (a) or Output (b).

## Instruction:
{user_query_instruction}: {user_query}

## Writing Styles:
{style_description}

## Output (a):
{output_a}

## Output (b):
{output_b}

## Which is best, Output (a) or Output (b)?

### C.5.2 RUBRIC

Table 12: **Pairwise Rubric Style Ranking Prompts:** {...} indicate where string formatting is done with "{user_query_instruction}" as the writing task instruction given to accompany the user query, "{user_query}" as the question answered or the Reddit post to summarize, "{style_description}" as the description of the style (e.g. "scholarly-yet-friendly"), and "{output_a}" and "{output_b}" the text to compare. For all style detection prompts the system prompt is: "You are a helpful instruction-following assistant that determines the best model output by selecting the best outputs for a given instruction."

---

**Helpfulness Targetted**

Review the user's question and the corresponding response using the additive 5-point scoring system described below. Points are accumulated based on the satisfaction of each criterion:

- Add 1 point if the response is relevant and provides some information related to the user's inquiry, even if it is incomplete or contains some irrelevant content.
- Add another point if the response addresses a substantial portion of the user's question, but does not completely resolve the query or provide a direct answer.
- Award a third point if the response answers the basic elements of the user's question in a useful way, regardless of whether it seems to have been written by an AI Assistant or if it has elements typically found in blogs or search results.
- Grant a fourth point if the response is clearly written from an AI Assistant's perspective, addressing the user's question directly and comprehensively, and is well-organized and helpful, even if there is slight room for improvement in clarity, conciseness or focus.
- Bestow a fifth point for a response that is impeccably tailored to the user's question by an AI Assistant, without extraneous information, reflecting expert knowledge, and demonstrating a high-quality, engaging, and insightful answer.

User: {user_query_instruction}: {user_query}

<response>{response}</response>

After examining the user's instruction and the response:

- Briefly justify your total score, up to 100 words.
- Conclude with the score using the format: 'Score: ¡total points¿'

Remember to assess from the AI Assistant perspective. To evaluate the response in alignment with this additive scoring model, systematically attribute points based on the outlined criteria.

---

**Style Targetted**

---

Review the writing style, the user's question, and the corresponding response using the additive 5-point scoring system described below. Points are accumulated based on the satisfaction of each criterion - Add 1 point if one or more style elements is at least present in one or two sentences, even if the response feels generic or mechanical.
- Add another point if the majority of writing style elements are at least minimally present.
- Award a third point if the majority of writing style elements are present in the response, but could be stronger.
- Grant a fourth point if the all of the writing style elements are present in the response, but could be stronger.
- Bestow a fifth point if all writing style elements are impeccably tailored to the user's question by an AI Assistant.

Writing styles: {style_description}

User: {user_query_instruction}: {user_query}

<response>{response}</response>

After examining the user's instruction and the response:

- Briefly justify your total score, up to 100 words.
- Conclude with the score using the format: 'Score: ¡total points¿'

Remember to assess from the AI Assistant perspective. To evaluate the response in alignment with this additive scoring model, systematically attribute points based on the outlined criteria.

## D    WRITING TASK USER-QUERY RESPONSE PROMPT SELECTION

To analyze how well a persona-driven judge is able to evaluate which of two responses is more personalized or how well personalized a given response is, it is important to have responses that vary in their degree of personalization. To this end we ran two analyses to select the prompts we use to generate a dataset of "personalized" responses: (1) synthetic evaluation and (2) human annotation. Our synthetic evaluation consisted of two main parts and allowed us to narrow the set of generation prompts to ask human annotators to evaluate. In one evaluation we ask an LLM judge to determine which of two responses better contained the style element(s) and in the other we ask an LLM judge to determine which style element(s) were included in a given generation.

### D.1    RANKING VALIDATION STUDY SET UP

We conducted a validation study to determine which versions of writing task prompts to use for generating the synthetic data for the four user-query response LLMs. The objective of this study was to rank the alignment of the style-conditioned LLM-generated text against the text generated without specifying a writing style. For each writing task, we generated text using the four different writing task prompts (Appendix C.2), for the email-writing task, 10 user queries (Appendix B), and five writing style sets (Fig 12). We additionally generate user-query responses using the default generation prompt (Appendix C.1) to get examples of the LLM's "default style". We then prompted a strong LLM (GPT-4o) with each style-conditioned and default user-query response pair to determine which of the two text samples aligns more with a given writing style.

We ran the above analysis with three different ranking prompts: twice with a style-conditioned evaluation prompt (adapted from the PEARL paper Mysore et al. (2023), see Figure 10) on (1) all preferences in a given set and (2) individual preferences, and once using a default evaluation prompt (Figure 11) that allows the LLM to rely on its default style preferences. The percentage of time that GPT-4o ranks preference-conditioned generation over default-conditioned generation is referred to as all_preference-conditioned, single_preference-conditioned, and default win rates, respectively. The "best" prompt is chosen by having the largest alignment score, namely: the mean between the all_preference-conditioned win rate, single_preference-conditioned win rate, and the inverse of the default style win rate. The best prompt for each LLM is then used to generate the data for the user study.

```
PEARL-Inspired Ranking Prompt

You are an experienced linguist who helps people compare {task} texts.
Given an {task} WRITING TOPIC, and two TARGET {TASK_PLURAL}, identify
which of the TARGET {TASK_PLURAL} aligns better with the provided
PREFERENCES. For your response use the following instructions:
1.  Make your judgment based on stylistic patterns based on the
PREFERENCES.
2.  Output {TASK} ONE if it aligns better with the PREFERENCES.
3.  Output {TASK} TWO if it aligns better with the PREFERENCES. Here is
the context:
### PREFERENCES ###
{preferences}
### WRITING TOPIC ###
Write a(n) {task} answering the following question:
{sub_task}
### {TASK} ONE ###
{option_a}
### {TASK} TWO ###
{option_b}
### INSTRUCTION ###
Output a justification for your judgment, then output {TASK} ONE or {TASK}
TWO to indicate your final decision.
```

Figure 10: This was the ranking prompt used in the validation study to compare two writing texts for a provided preference (set). The text orders are randomized. $\{task\} \in$ {email, tweet, summary}, $\{TASK\} \in$ {EMAIL, TWEET, SUMMARY}, and $\{TASK\_PLURAL\} \in$ {EMAILS, TWEETS, SUMMARIES}

```
Default Ranking Prompt

Which {task} do you prefer?  Explain why, and provide your final answer
on a new line in the format "ANSWER: OPTION {A or B}".
### CONTEXT ###
Write a(n) {task} answering the following question:
{sub_task}
### OPTION A ###
{option_a}
### OPTION B ###
{option_b}
```

Figure 11: This was the ranking prompt used in the validation study to compare two writing texts for a provided preference (set). The texts orders are randomized. $\{task\} \in$ {email, tweet, summary}

## D.2 RANKING VALIDATION STUDY RESULTS

As mentioned in Appendix D.1, the preference-conditioned generation prompt was selected using the alignment score produced by GPT-4o as our evaluator. The high-level results with the aggregated alignment score can be seen in Table 13 and the detailed results per ranking prompt are in Figures 14 (email), 15 (tweet), and 16 (summary). The prompts in bold represent the selected prompt for running the generations, for each $LLM_g$.

Table 13: The alignment scores across tasks, LLMs, and generation prompts (see Appendix C.2 for the full prompts). The selected generation prompt per task and LLM is **bolded**, and the selection criteria was the highest alignment score for email-writing and tweet-writing. For summarization, the percentage of summaries that were shorter than the original post's length are sub-scripted. The selected generation prompt per LLM for summaries was due to a combination of the summary length and alignment score.

| Generation Prompt | GPT 4o mini | OLMo 13B | QWEN 14B | Claude 3.7 Sonnet |
|---|---|---|---|---|
| Email-Writing Task | | | | |
| Email Prompt 1 | 86.22% | 92.89% | 91.11% | 88.44% |
| Email Prompt 2 | **92.89%** | 69.56% | **95.11%** | 90.89% |
| Email Prompt 3 | 90.22% | **95.33%** | 94.89% | 91.11% |
| Email Prompt 4 | 86.89% | 92.89% | 93.11% | **91.56%** |
| Tweet-Writing Task | | | | |
| Tweet Prompt 1 | **88.44%** | 80.22% | **89.78%** | 84.00% |
| Tweet Prompt 2 | 80.22% | 87.77% | 82.22% | 84.89% |
| Tweet Prompt 3 | 84.44% | **88.67%** | 87.33% | 79.56% |
| Tweet Prompt 4 | 85.11% | 86.44% | 86.89% | **84.89%** |
| Summarization Task | | | | |
| Summary Prompt 1 | $78.44\%_{100\%}$ | $75.78\%_{90\%}$ | $76.89\%_{90\%}$ | $73.78\%_{90\%}$ |
| Summary Prompt 2 | $93.78\%_{58\%}$ | $97.33\%_{30\%}$ | $95.78\%_{16\%}$ | $94.67\%_{62\%}$ |
| Summary Prompt 3 | $\mathbf{91.78\%_{100\%}}$ | $\mathbf{95.11\%_{84\%}}$ | $\mathbf{94.89\%_{100\%}}$ | $\mathbf{96.00\%_{90\%}}$ |
| Summary Prompt 4 | $94.89\%_{76\%}$ | $95.33\%_{34\%}$ | $96.00\%_{64\%}$ | $65.56\%_{84\%}$ |

Table 14: **Email Writing Task**: "Unconditioned" refers to the preference selection prompt that provides the LLM with no information about the styles to assess on (see Figure 11); "Single Style" is the preference selection prompt that provides a single style to asses on; and "All Styles" is the preference selection prompt that provides all styles from a style set (see Figure 10 for style section prompt). The alignment score is the aggregate across the 3 prompt scores (with the unconditioned score inverted). Highest alignment score rows are **bolded**.

| Generation Prompt | Unconditioned | Single Style | All Styles | Alignment Score |
|---|---|---|---|---|
| OLMo-2-1124-13B-Instruct | | | | |
| Prompt 1 | 10.00 | 88.67 | 100.00 | 92.89 |
| Prompt 2 | 16.00 | 64.67 | 60.00 | 69.56 |
| **Prompt 3** | **2.00** | **88.00** | **100.00** | **95.33** |
| Prompt 4 | 10.00 | 88.67 | 100.00 | 92.89 |
| Qwen2.5-14B-Instruct | | | | |
| Prompt 1 | 14.00 | 89.33 | 98.00 | 91.11 |
| **Prompt 2** | **6.00** | **91.33** | **100.00** | **95.11** |
| Prompt 3 | 4.00 | 88.67 | 100.00 | 94.89 |
| Prompt 4 | 12.00 | 91.33 | 100.00 | 93.11 |
| GPT-4o-mini | | | | |
| Prompt 1 | 32.00 | 90.67 | 100.00 | 86.22 |
| **Prompt 2** | **8.00** | **86.67** | **100.00** | **92.89** |
| Prompt 3 | 16.00 | 86.67 | 100.00 | 90.22 |
| Prompt 4 | 30.00 | 90.67 | 100.00 | 86.89 |
| Claude Sonnet 3.7 | | | | |
| Prompt 1 | 24.00 | 89.33 | 100.00 | 88.44 |
| Prompt 2 | 20.00 | 92.67 | 100.00 | 90.89 |
| Prompt 3 | 18.00 | 91.33 | 100.00 | 91.11 |
| **Prompt 4** | **16.00** | **90.67** | **100.00** | **91.56** |

Table 15: **Tweet Writing Task**: "Unconditioned" refers to the preference selection prompt that provides the LLM with no information about the styles to assess on (see Figure 11); "Single Style" is the preference selection prompt that provides a single style to asses on; and "All Styles" is the preference selection prompt that provides all styles from a style set (see Figure 10 for style section prompt). The alignment score is the aggregate across the 3 prompt scores (with the unconditioned score inverted). Highest alignment score rows that were used to select the generation prompt are **bolded**.

| Generation Prompt | Unconditioned | Single Style | All Styles | Alignment Score |
|---|---|---|---|---|
| OLMo-2-1124-13B-Instruct | | | | |
| Prompt 1 | 30.00 | 78.67 | 92.00 | 80.22 |
| Prompt 2 | 20.00 | 83.30 | 100.00 | 87.77 |
| **Prompt 3** | **22.00** | **88.00** | **100.00** | **88.67** |
| Prompt 4 | 24.00 | 85.33 | 98.00 | 86.44 |
| Qwen2.5-14B-Instruct | | | | |
| **Prompt 1** | **20.00** | **89.33** | **100.00** | **89.78** |
| Prompt 2 | 16.00 | 74.67 | 88.00 | 82.22 |
| Prompt 3 | 24.00 | 86.00 | 100.00 | 87.33 |
| Prompt 4 | 28.00 | 88.67 | 100.00 | 86.89 |
| GPT-4o-mini | | | | |
| **Prompt 1** | **22.00** | **87.33** | **100.00** | **88.44** |
| Prompt 2 | 38.00 | 82.67 | 96.00 | 80.22 |
| Prompt 3 | 34.00 | 87.33 | 100.00 | 84.44 |
| Prompt 4 | 32.00 | 89.33 | 98.00 | 85.11 |
| Claude Sonnet 3.7 | | | | |
| Prompt 1 | 36.00 | 88.00 | 100.00 | 88.00 |
| Prompt 2 | 32.00 | 88.67 | 98.00 | 84.89 |
| Prompt 3 | 48.00 | 88.67 | 98.00 | 79.56 |
| **Prompt 4** | **38.00** | **92.67** | **100.00** | **84.89** |

Table 16: **Summary Writing Task**: "Unconditioned" refers to the preference selection prompt that provides the LLM with no information about the styles to assess on (see Figure 11); "Single Style" is the preference selection prompt that provides a single style to asses on; and "All Styles" is the preference selection prompt that provides all styles from a style set (see Figure 10 for style section prompt). The alignment score is the aggregate across the 3 prompt scores (with the unconditioned score inverted). Highest alignment score rows are **bolded**.

| Generation Prompt | Unconditioned | Single Style | All Styles | Alignment Score |
|---|---|---|---|---|
| OLMo-2-1124-13B-Instruct | | | | |
| Prompt 1 | 26.00 | 73.33 | 80.00 | 90.00 |
| Prompt 2 | 0.00 | 92.00 | 100.00 | 30.00 |
| **Prompt 3** | **2.00** | **87.33** | **100.00** | **84.00** |
| Prompt 4 | 0.00 | 88.00 | 98.00 | 34.00 |
| Qwen2.5-14B-Instruct | | | | |
| Prompt 1 | 32.00 | 80.67 | 82.00 | 76.89 |
| Prompt 2 | 0.00 | 91.33 | 96.00 | 95.78 |
| **Prompt 3** | **2.00** | **86.67** | **100.00** | **94.89** |
| Prompt 4 | 0.00 | 88.00 | 100.00 | 96.00 |
| GPT-4o-mini | | | | |
| Prompt 1 | 30.00 | 79.33 | 86.00 | 78.44 |
| Prompt 2 | 10.00 | 91.33 | 100.00 | 93.78 |
| **Prompt 3** | **4.00** | **79.33** | **100.00** | **91.78** |
| Prompt 4 | 2.00 | 86.67 | 100.00 | 94.89 |
| Claude Sonnet 3.7 | | | | |
| Prompt 1 | 38.00 | 79.33 | 80.00 | 73.78 |
| Prompt 2 | 8.00 | 92.00 | 100.00 | 94.67 |
| **Prompt 3** | **0.00** | **88.00** | **100.00** | **96.00** |
| Prompt 4 | 100.00 | 96.67 | 100.00 | 65.56 |

# E DATASET

## E.1 CREATING THE DATASET

The dataset consists of text for three writing tasks: email writing, tweet writing, and summarization. For email and tweet writing, the LLM is prompted to write a response to a given user query, and for summarization, to summarize a given Reddit post. The user queries for the email and tweet writing tasks are in Appendix B (Figure 5), and the associated Reddit post IDs from the SLF5k dataset that we used as the summary user queries can be found in Appendix B (Figure 6).

The queries and Reddit posts were selected to cover different domains. The queries are all questions that are casual and not overly academic. We additionally selected the queries and posts to have a neutral implied style to avoid unintentionally biasing the style of the LLM's generations. For example, queries that use overly formal and academic language may implicitly induce the LLM to respond in a formal and academic manner. An additional layer of filtering was used when selecting the Reddit posts to remove entries that included harmful or sensitive content.

For each writing task $t$, each $LLM_g$ was provided with task-specific instructions. The style-conditioned, task-specific prompts were selected following Section D.2 and were selected per LLM. The style-conditioned prompt per $LLM_g$ and writing task $t$ pair is marked in bold in Table 13. The default prompts are in Appendix C.1.

```
Writing Style Preference Sets

Set 0:  "poetic and lyrical", "storytelling", "scholarly-yet-friendly"
Set 1:  "Inspirational and uplifting", "journalistic",
"scholarly-yet-friendly"
Set 2:  "robotic and emotionless", "telegraphic brevity", "legal
precision"
Set 3:  "step-by-step instructional", "encouraging and supportive",
"visual and spatial"
Set 4:  "playful and whimsical", "rhyming and rhythmic", "sensory
focused"
Alternate Option:  No preference set specified
```

Figure 12: The five writing style preference sets used to condition the LLMs during generation when creating the dataset. Each preference set is composed of three individual writing styles.

Each persona consists of a set of writing styles with each set containing three elements (see Figure 12). The style elements were selected to create distinct sets, but to not directly contrast within a set. We avoided using styles that are known to be strong defaults in LLMs (e.g. bullet points) as these will likely occur across many responses without additional prompting.

For each writing task $t$, user-query, and writing style set $\mathbf{u}_i$ (including the no writing style condition), each $LLM_g$ was prompted to generate 30 responses. The responses were then filtered and cleaned to remove any that explicitly reference the writing style set the LLM was conditioned on to avoid biasing the human labelers. The filtering step was not applied to the responses that were not conditioned on any writing styles. After filtering, 10 responses were selected via clustering to ensure some level of diversity in the responses.

Some LLMs (i.e. `OLMo-2-1124-13B-Instruct` and `GPT-4o` struggled to produce at least 10 responses to certain user queries and style sets without referencing the target writing styles for the email task. Therefore, there are 2400 writing samples for the tweet and summary tasks with only $2,303$ (post filtering) for the email task.

## E.2 COLLECTING THE STYLE LABELS AND PAIRWISE RANKINGS

To measure how well $LLM_e$ is able to evaluate the adherence of $LLM_g$ to the required writing styles, ground-truth labels are required that specify the level of adherence. To obtain ground-truth, we run a user-study to collect human judgments about the extent to which the each writing style element is present in each $r_g$ (i.e., the style detection task). From these annotations we extract labels, against which the $LLM_e$ judgments are compared. Additionally, we collected pairwise rankings of $(r_g^a, r_g^b)$ in two cases: $LLM_g$ adherence to (1) instructions and (2) style labels (i.e., the pairwise ranking task).

**Stimuli**    The text excerpts for annotation, $r_g \in R$, were generated with each $\text{LLM}_g$, and selected as described in Section E.1. For the pairwise ranking tasks, a subset of these responses were chosen and organized into pairs as described in Section 3.2.2.

**Style detection task**    Each style detection task consisted of four participant steps: (1) reading $r_g$, (2) rating on a three-point scale (1=not present, 2=somewhat present, 3=strongly present—exact wording depending on the queried style) the degree to which each writing style, $w^s$, is present, following Chhun et al. (2022), (3) rating how realistic $r_g$ is, and (4) writing an annotation justification. Note that annotators rate for each style $w^s$ individually and are not aware of the writing style personas $\mathbf{u}_i$ used to steer the model. The complete survey design is presented in Appendix F.

**Ranking task**    The pairwise ranking tasks were used to assess how well the generator $\text{LLM}_g$ followed the instructions given to it (e.g., to write an email) or the style sets $\mathbf{u}_i$. The annotation process consisted again of four steps: (1) reading the instruction/style sets given to $\text{LLM}_g$, (2) reading the pair of responses $(r_g^a, r_g^b)$, 3) selecting which response is better (five levels), and (4) writing a justification. The participants were asked to follow a rubric that described how to assess whether given responses adhered to instructions or style sets, following Dubois et al. (2023). The complete survey design is presented in Appendix G.

**Annotators**    We recruited approximately 1,000 annotators per survey through a crowd-sourcing platform. Annotators were compensated according to the number of annotations that they provided. We collected sufficient annotations to ensure that each annotation task, that is corresponding $r_g$ or $(r_g^a, r_g^b)$, received ten annotations in the style detection task, and five annotations in the pairwise ranking task.

### E.3    CORRELATIONS BETWEEN HUMAN DETECTED STYLES

The style labels detected in the data display weak to moderate correlation as measured using the Spearman rank correlation coefficient and interpreted based on guidelines in Haldun (2018)—see Figure 13—with Pearson correlation coefficients taking similar magnitudes. Moderate positive correlations are observed between the pairs Poetic—Rhyming ($\rho = 0.59$), Playful—Poetic ($\rho = 0.59$), and Playful—Rhyming ($\rho = 0.54$). Weak negative correlations are observed between the pairs Playful—Robotic ($\rho = -0.35$) and Poetic—Robotic ($\rho = -0.34$). Correlations between the other pairs are also mostly weak.

### E.4    CONDITIONAL PROBABILITIES OF HUMAN DETECTED STYLES

The conditional probabilities of assigning the label $y^{d=3}$ indicating strong presence of a style tend to be higher when it is used in conditioning the generator $\text{LLM}_g$ (Figure 14 A-F ). This holds for each of the styles sets except for set B, which corresponds to generating text in the styles of "Inspirational and Uplifting", "Journalistic", and "Rich descriptions". Additionally, the probability of observing the "not present" label for each queried style is higher when default generations are examined (pane F), in contrast to conditioned ones, except for the styles "Scholarly-yet-friendly" and "Journalistic".

### E.5    HUMAN ANNOTATION JUSTIFICATION ANALYSIS

As shown in Appendix F.3), we ask annotators to provide justifications for their annotations for presence of styles. To understand the reasoning behind annotating a certain style and understand correlations amongst styles, we conduct an analysis of these justifications. Specifically, we extract the top 10 TF-IDF tri-grams across the annotations when a style is given a score of 3 (very present). Since style annotations were done in sets of 4 but a single, combined justification string was elicited at the end of all 4 annotations, we could not isolate parts of the string only pertaining to a specific style. However, since the tri-grams are extracted using TF-IDF for all instances when a style is marked as very present, only the ones present in several annotations would appear, surfacing the ones relevant for a given style.

The tri-grams are shown in Table 17. Annotators often refer directly to the corresponding style in the justifications when a style is marked as very present, as can be seen in the tri-grams. Other descriptors for a style such as *unambiguous* for "Legal", *mechanical* for "Robotic", *imagery* for "Visual", *factual* for "Journalistic" can also be seen. References to other styles within the style group can also be seen, which can be attributed to the combined justification for a style group.

Table 17: Top 10 tri-grams from the justifications provided by humans when a style is annotated with a score of 3 (Very Present). The rows are separated by the style sets within which they were annotated.

| Style | Top Tri-Grams |
|---|---|
| **step** | ['**step step** instructions', 'relies **step step**', 'strongly relies **step**', '**step step** guide', '**step step** instructional', 'clear sequential order', 'tic tac toe', 'clear **step step**', 'information clear sequential', '**step step** instruction'] |
| **legal** | ['high level **legal**', 'level **legal** precision', '**step step** instructions', 'characterized high level', 'relies **step step**', 'carefully defined terms', 'exact unambiguous language', 'strongly relies **step**', '**step step** instructional', 'language carefully defined'] |
| **rhyming** | ['**step step** instructions', 'sounds flowing cadence', 'patterned sounds flowing', 'memorable reading experience', 'features patterned sounds', 'cadence using rhyme', 'flowing cadence using', 'musical memorable reading', 'lighthearted uplifting precise', 'uplifting precise **robotic**'] |
| **robotic** | ['**step step** instructions', 'mechanical way communicating', 'relies **step step**', 'high level **legal**', 'level **legal** precision', 'email **rhyming** rhythmic', 'characterized **legal** precision', 'characterized high level', '**step step** instruction', '**step** instructions **robotic**'] |
| **visual** | ['tone **visual** spatial', 'somewhat **scholarly** friendly', '**scholarly** friendly **visual**', '**visual** spatial emphasizes', 'spatial emphasizes imagery', 'friendly **visual** spatial', '**visual** spatial **storytelling**', 'tic tac toe', 'ratings based email', 'ratings based provided'] |
| **scholarly** | ['**scholarly** friendly **visual**', 'somewhat **visual** spatial', 'friendly **visual** spatial', 'ratings based email', 'tone **visual** spatial', '**scholarly** friendly tone', 'blends rigorous reasoning', 'rigorous reasoning citation', 'citation approachable language', 'reasoning citation approachable'] |
| **journalistic** | ['**scholarly** friendly **journalistic**', 'factual reporting person', 'reporting person observation', 'merges factual reporting', '**scholarly** friendly **visual**', 'tone **storytelling journalistic**', '**journalistic** merges factual', 'educates telling story', 'somewhat **scholarly** friendly', '**journalistic scholarly** friendly'] |
| **storytelling** | ['uses scene building', '**storytelling visual** appeal', '**visual** spatial **storytelling**', 'somewhat **scholarly** friendly', 'scene building dialogue', 'tone **visual** spatial', 'building dialogue plot', 'dialogue plot beats', 'somewhat **visual** spatial', 'tic tac toe'] |
| **poetic** | ['rhythmic sentence patterns', '**encouraging** supportive **poetic**', 'supportive **poetic** lyrical', 'vivid imagery rhythmic', 'imagery rhythmic sentence', 'senses **encouraging** supportive', 'email engages senses', 'engages senses **encouraging**', '**poetic** lyrical natural', 'patterns figurative language'] |
| **realism** | ['**encouraging** supportive **poetic**', 'supportive **poetic** lyrical', 'senses **encouraging** supportive', '**poetic** lyrical natural', 'engages senses **encouraging**', 'email engages senses', 'somewhat **sensory** focused', 'tic tac toe', 'somewhat **encouraging** supportive', 'natural person write'] |
| **sensory** | ['**encouraging** supportive **poetic**', 'supportive **poetic** lyrical', 'senses **encouraging** supportive', 'email engages senses', '**poetic** lyrical natural', 'engages senses **encouraging**', '**sensory** focused contains', 'focused contains vivid', 'contains vivid **descriptions**', 'tone **encouraging** supportive'] |
| **encouraging** | ['tone **encouraging** supportive', '**encouraging** supportive **poetic**', 'supportive **poetic** lyrical', 'senses **encouraging** supportive', 'engages senses **encouraging**', 'email engages senses', '**poetic** lyrical natural', 'somewhat **sensory** focused', '**sensory** focused **encouraging**', 'positive empathetic language'] |
| **playful** | ['tone **playful** whimsical', 'contains **rich descriptions**', 'characterized **telegraphic** brevity', 'uses imaginative language', 'somewhat **inspirational** uplifting', 'language lighthearted tone', 'imaginative language lighthearted', 'tic tac toe', 'high level **telegraphic**', 'level **telegraphic** brevity'] |
| **telegraphic** | ['high level **telegraphic**', 'level **telegraphic** brevity', 'characterized high level', 'short clipped sentences', 'contains **rich descriptions**', 'uses short clipped', 'clipped sentences phrases', 'tone **playful** whimsical', 'writing uses short', 'concise writing uses'] |
| **inspirational** | ['contains **rich descriptions**', 'characterized **telegraphic** brevity', 'level **telegraphic** brevity', 'high level **telegraphic**', 'characterized high level', 'tone **playful** whimsical', '**playful** whimsical **inspirational**', 'whimsical **inspirational** uplifting', 'uses imaginative language', 'clear concise good'] |
| **rich-descriptions** | ['contains **rich descriptions**', 'characterized **telegraphic** brevity', 'tone **playful** whimsical', 'high level **telegraphic**', 'level **telegraphic** brevity', 'characterized high level', 'somewhat **inspirational** uplifting', 'uses imaginative language', 'somewhat characterized **telegraphic**', 'lighthearted uplifting brief'] |

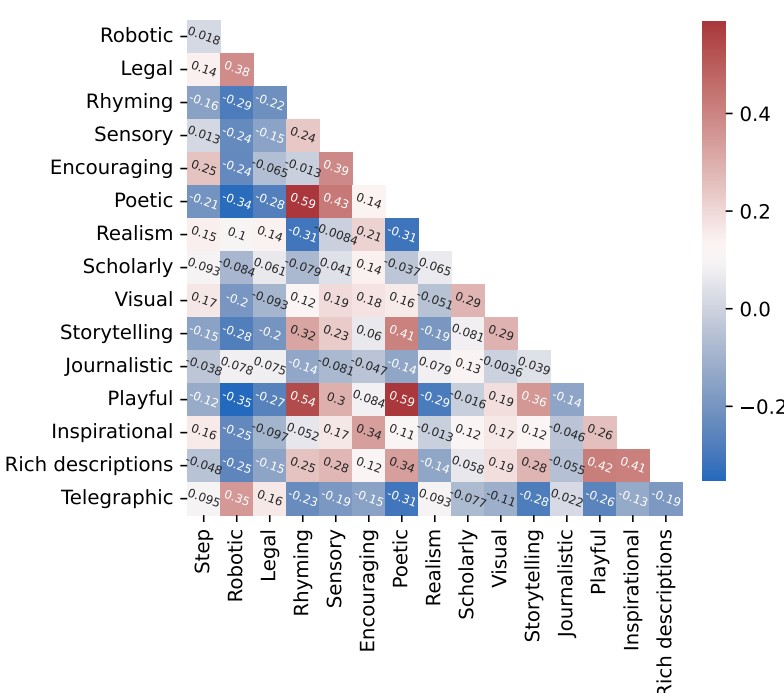

Figure 13: Spearman rank correlation coefficients between detected styles across all task types (email, tweet, and summary). First word of each style referenced in the labels.

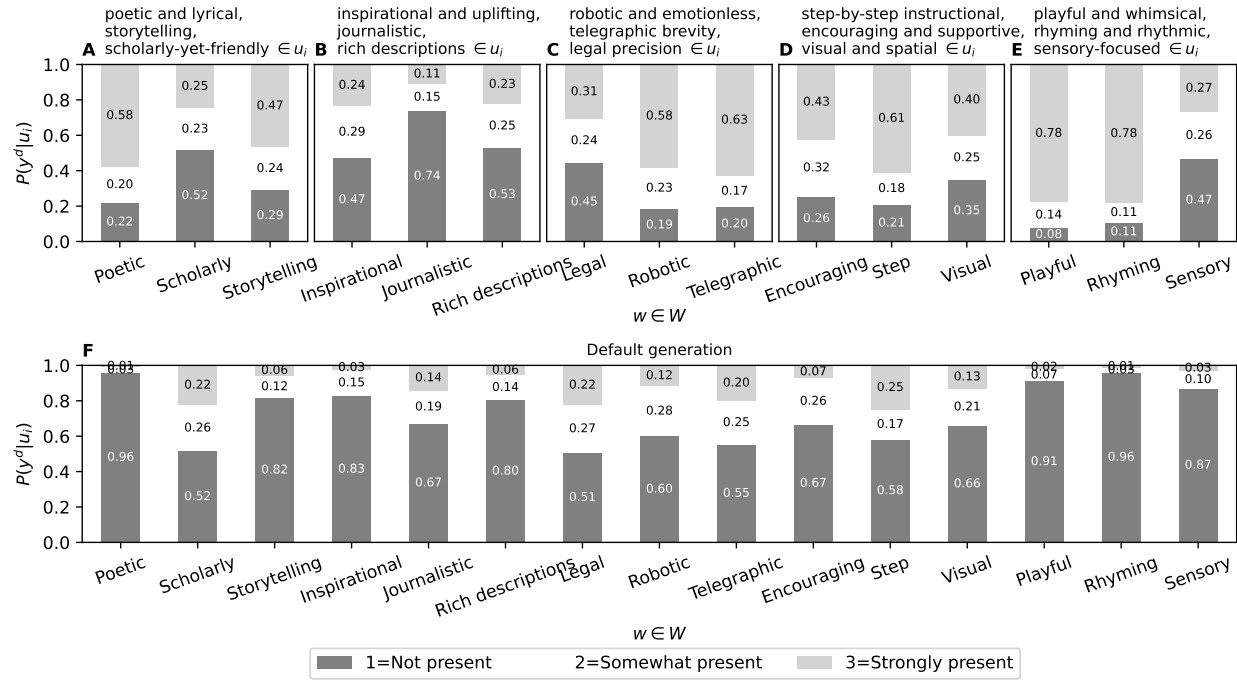

Figure 14: Conditional probability of observing an annotation $y^d$ (1="not present", 2="somewhat present", 3="strongly present") from the human annotators for various style sets, $u_i$, when those styles sets are included in the prompt $p_g^t$ to the generator (panes A-E). Values are shown for each style $w \in W$ queried in the survey and across writing tasks (email, tweets, summary). Corresponding conditional probabilities are also presented for the default generation (pane F). First word of each style referenced in the labels.

# F    STYLE DETECTION USER STUDY QUESTIONNAIRES

The instructions given in the style detection study were split in two sections: general instructions that the participant was shown at the beginning of the study, and could access at any point (Appendix F.1), and task instructions shown alongside each annotation task (Appendix F.2). The 16 styles were split into four different sets randomly (as shown in Section F.3).

## F.1    GENERAL INSTRUCTIONS

The following instructions were displayed to each participant in the beginning of the study, and could be accessed at any point. The instructions were formulated to refer to text types specific to each task (emails, tweets, summaries).

---

In the following tasks, you will be shown [text type] and will need to complete the following steps:

- Read the [text type].
- Rate the presence of four different styles in this [text type].
- Justify your ratings.

Please carefully assess how well each style aligns with the manner of expression used in the text (i.e., tone, word selection, sentence structure, etc.). You may ignore the specifics of the [text type] content. Some of the [text type] are incomplete, so you may also ignore placeholders (e.g., [NAME]). All or any of the styles may be present in the [text type].

Note that you are not allowed to use outside assistance, including chatbots, to complete these tasks.

> Instructions are also shown in grey boxes like this in each task.

In this set of tasks, you will rate the presence of the following four styles:

[List of styles, corresponding to Section F.3]

---

## F.2    TASK INSTRUCTIONS

Alongside each annotation task, the following instructions were displayed.

---

> 1. Please read the following text:

[Generated response]

> 2. Please review the following styles and assess how well they describe the text you just read. A definition is provided for each style for guidance. **DO** assess how well the styles align with the manner of expression (i.e., tone, word selection, sentence structure, etc.). You **DO NOT** have to consider how well the styles align with the content of the text (i.e., what is said).

[Style questions from sets in Section F.3]

> 3. Please justify your responses by providing a concise summary of text elements that matched an identified style. You may copy paste extracts from the [text type]. You do not need to provide a justification for styles that are not present in the text. If you did not identify any styles, you can write: "None of the styles are present."

[Box for justification]

---

### F.3 STYLE QUESTIONS

The 16 examined styles were split into four different surveys randomly, resulting in the following sets and corresponding labels. The labels were formulated following a similar structure in Chhun et al. (2022).

#### F.3.1 STYLE SET 1

**Step-by-step instructional:** Presents information in a clear, sequential order, guiding the reader through each stage of a process with precise, actionable directions.

- The text strongly relies on step-by-step instructions
- The text somewhat relies on step-by-step instructions
- The text does not rely on step-by-step instructions
- I can't tell if this style is present or not

**Robotic and emotionless:** A flat, mechanical way of communicating that lacks emotional nuance, human warmth, and expressive variation.

- The text is very robotic and emotionless
- The text is somewhat robotic and emotionless
- The text is not robotic and emotionless
- I can't tell if this style is present or not

**Legal precision:** Relies on exact, unambiguous language and carefully defined terms to minimize misinterpretation and ensure clarity in interpretation.

- The text is characterized by high level of legal precision
- The text is somewhat characterized by legal precision
- The text is not characterized by legal precision
- I can't tell if this style is present or not

**Rhyming and rhythmic:** Features patterned sounds and flowing cadence, often using rhyme and meter to create a musical, memorable reading experience.

- The text is very rhyming and rhythmic
- The text is somewhat rhyming and rhythmic
- The text is not rhyming and rhythmic
- I can't tell if this style is present or not

#### F.3.2 STYLE SET 2

**Sensory-focused:** Vividly engages the five senses—sight, sound, smell, taste, and touch—to immerse the reader in a rich, tangible experience.

- The text is very sensory-focused
- The text is somewhat sensory-focused
- The text is not sensory-focused
- I can't tell if this style is present or not

**Encouraging and supportive:** Uses positive, empathetic language to uplift the reader, build confidence, and foster a sense of motivation and reassurance.

- The tone of the text is very encouraging and supportive
- The tone of the text is somewhat encouraging and supportive

- The tone of the text is not encouraging and supportive
- I can't tell if this style is present or not

**Poetic and lyrical:** Employs vivid imagery, rhythmic sentence patterns, and figurative language to evoke emotion and sensory experience.

- The text is very poetic and lyrical
- The text is somewhat poetic and lyrical
- The text is not poetic and lyrical
- I can't tell if this style is present or not

**Realism:** The style of the text is realistic and natural—something that another person would write.

- The text feels very realistic
- The text feels somewhat realistic
- The text does not feel realistic
- I can't tell if this text feels realistic

### F.3.3 STYLE SET 3

**Scholarly-yet-friendly:** Blends rigorous reasoning and citation with approachable language, first-person asides, and analogies that humanize technical topics.

- The text is very scholarly-yet-friendly
- The text is somewhat scholarly-yet-friendly
- The text is not scholarly-yet-friendly
- I can't tell if this style is present or not

**Visual and spatial:** Emphasizes imagery, layout, and the arrangement of elements to help the reader understand concepts through visual structure and spatial relationships.

- The tone of the text is very visual and spatial
- The tone of the text is somewhat visual and spatial
- The tone of the text is not visual and spatial
- I can't tell if this style is present or not

**Storytelling:** Uses scene-building, dialogue, and plot beats—even in nonfiction—to convey ideas through personal anecdotes and mini-stories.

- The tone of the text is very storytelling
- The tone of the text is somewhat storytelling
- The tone of the text is not storytelling
- I can't tell if this style is present or not

**Journalistic:** Merges factual reporting with first-person observation, balancing objectivity with a clear, recognizable narrator.

- The text is very journalistic
- The text is somewhat journalistic
- The text is not journalistic
- I can't tell if this style is present or not

### F.3.4 STYLE SET 4

**Playful and whimsical:** Uses imaginative language, lighthearted tone, and unexpected twists to entertain and delight the reader with a sense of fun and creativity.

- The tone of the text is very playful and whimsical
- The tone of the text is somewhat playful and whimsical
- The tone of the text is not playful and whimsical
- I can't tell if this style is present or not

**Inspirational and uplifting:** Encouraging diction, motivational phrasing, and inclusive pronouns ("we," "us") that position the writer as a supportive guide.

- The text is very inspirational and uplifting
- The text is somewhat inspirational and uplifting
- The text is not inspirational and uplifting
- I can't tell if this style is present or not

**Rich descriptions:** Relies on concrete sensory details (sound, smell, touch) and precise adjectives, inviting readers into the writer's lived experience.

- The text contains several rich descriptions
- The text contains some rich descriptions
- The text does not contain rich descriptions
- I can't tell if this style is present or not

**Telegraphic brevity:** A concise writing style that uses short, clipped sentences or phrases, often omitting unnecessary words like articles or conjunctions, to convey information quickly and efficiently.

- The text is characterized by high level of telegraphic brevity
- The text is somewhat characterized by telegraphic brevity
- The text is not characterized by telegraphic brevity
- I can't tell if this style is present or not

## G    Pairwise ranking user study questionnaires

Similar to style detection, the instructions given to the users were split in two sections in the pairwise ranking task. The first section described general instructions accessible throughout the task (Appendix G.1 and G.3 for adherence to instructions and style sets, respectively). The general instructions also contained the rubrics that the participants were asked to follow, which format was obtained from Dubois et al. (2023). The second section contained each annotation task (Appendix G.2 and Appendix G.4). The format of rating the options also followed that of Dubois et al. (2023), with the addition of an option indicating a tie in the ranking.

### G.1    General instructions—Pairwise ranking of adherence to instructions

The following instructions were displayed to each participant in the beginning of the study querying adherence to instructions in the pairwise ranking tasks, and could be accessed at any point.

---

In this task, you will rate responses of an AI model to either an instruction or a question.

You will first read:

1. The instruction/question given to the AI.
2. The two responses (Option A and Option B) from the AI.

Your task is to decide which response is better. There are several dimensions that you can think along. Consider the following questions:

- Is the response helpful? For example, if the instruction asked for a recipe for healthy food, and the response is a useful recipe, then we can consider it helpful.
- Is the response language natural? For example, AI responses are often repetitive, which is not natural.
- Is the response factual/accurate? AI responses often make up new information. For example, if the response claims that the US is not a country then you should consider it inaccurate.
- Based on your aesthetics, which one do you prefer? For example, you might prefer one poem over another poem.
- An so on ... ultimately, you should decide which response is better based on your judgment and based on your own preference.

There are five options for you to choose from:

1. Option A is better: If you think option A has an advantage, then choose this option.
2. Option A is slightly better: Option A is marginally better than option B and the difference is small.
3. Both options are equally good: There is no difference in quality between Options A and B.
4. Option B is slightly better: Option B is marginally better than option A and the difference is small.
5. Option B is better: If you think option B has an advantage, then choose this option.

There are cases where the difference between the two responses is not clear. In this case, you can choose the second, third, or fourth option. However, in general, we ask you to choose those options as rarely as possible.

Note that you are not allowed to use outside assistance, including chatbots, to complete these tasks.

> Instructions are also shown in grey boxes like this in each task.

---

### G.2    Task instructions—Pairwise ranking of adherence to instructions

Alongside each annotation task querying adherence to instructions (Appendix G.1), the following instructions were displayed.

---

1. Please read the following instructions given to the AI:

[An instruction or a question]

2. Please read the following two texts (option A on the left, option B on the right):

[Option A and Option B presented side-by-side]

3. Rate the options.

- Option A is better
- Option A is slightly better (Only pick this if it's truly close)
- Both options are equally good (Only pick if the option quality is truly indistinguishable)
- Option B is slightly better (Only pick this if it's truly close)
- Option B is better

4. Please justify your preference (you may copy paste extracts from the chosen text). If you have no preference over the texts, please write: "I think the texts are equally aligned with the given styles".

[Box for justification]

---

### G.3 General instructions—Pairwise ranking of adherence to styles

The following instructions were displayed to each participant in the beginning of the study querying adherence to styles in the pairwise ranking tasks, and could be accessed at any point.

---

In this task, you will be shown responses of an AI model to either an instruction or a question. In each task, you are shown a pair of responses. Your task is to decide which response better follows or aligns with a set of given writing styles.

You will first read:

1. The three writing styles the AI was instructed to follow.
2. The two responses (Option A and Option B) from the AI.

Your task is to decide which response better follows the given set of writing styles. There are several dimensions that you can think along. Consider the following questions:

- Are all writing styles present in the response? For example, the AI may have followed one or two of the styles instead of all three.
- To what extent are the three writing styles present? For example, the AI maybe have written all sentences following the given writing styles, or may have followed the writing style for a single sentence.
- Is the response language natural? For example, one or all of the writing styles might be incorporated in an awkward or "over the top" manner.
- Are the responses coherent? For example, in attempting to using onomatopoeia the AI may have written only words such as, "bang", "clang", "zap", "ding ding", etc.
- An so on ... ultimately, you should decide which response is better based on your judgment about the given writing styles.

There are five options for you to choose from, where "better" refers to the response that is more aligned with the given writing styles:

1. Option A is better: If you think option A has an advantage, then choose this option.
2. Option A is slightly better: Option A is marginally better than option B and the difference is small.
3. Both options are equally good: There is no difference in quality between Options A and B.
4. Option B is slightly better: Option B is marginally better than option A and the difference is small.
5. Option B is better: If you think option B has an advantage, then choose this option.

There are cases where the difference between the two responses is not clear. In this case, you can choose the second, third, or fourth option. However, in general, we ask you to choose those options as rarely as possible.

Note that you are not allowed to use outside assistance, including chatbots, to complete these tasks.

> Instructions are also shown in grey boxes like this in each task.

### G.4 Task instructions—Pairwise ranking of adherence to styles

Alongside each annotation task querying adherence to styles (Appendix G.3), the following instructions were displayed.

> 1. Please note the following writing styles the AI was instructed to follow:

[List of styles]

> 2. Please read the following two texts (option A on the left, option B on the right):

[Option A and Option B presented side-by-side]

> 3. Rate the options.

- Option A is more aligned with the given styles
- Option A is slightly more aligned with the given styles (Only pick this if it's truly close)
- Both options are equally aligned with the given styles (Only pick if the option quality is truly indistinguishable)
- Option B is slightly more aligned with the given styles (Only pick this if it's truly close)
- Option B is more aligned with the given styles

> 4. Please justify your preference (you may copy paste extracts from the chosen text). If you have no preference over the texts, please write: "I think the texts are equally aligned with the given styles".

[Box for justification]

## H  MAPPING STYLE DETECTION PROMPTS RESPONSES TO HUMAN LABELS

We explored representing the human style detection annotations binary, two-class labels (i.e. "not present" and "present") or leaving them untouched as three-class labels (i.e. "not present", "somewhat present", and "very present"). The two-class labels considered are "not present" and "present", which involves mapping "somewhat present" and "very present" to "present".

We additionally explore different strategies to map the LLM style detection annotations to the human annotations. This is necessary as the LLM annotations for the binary detection, probability estimation, and Likert-10 style prompts do not directly map to the human annotations. There is not a mapping for the annotations from the binary detection LLM style detection prompt to the human labels when the labels are: "not present", "somewhat present", and "very present". Whenever the LLM style detection prompt is able to assign more unique annotation values than there are human labels, the LLM annotations must be binned and mapped to the human labels.

We use the following mappings from LLM annotations to human labels for the manual mapping setting when using two and three labels.

**Two Labels:**

- *Binary Detection* - never mapped and can only be used where there are two human labels
- *Likert-3* (i.e. {"not present", "somewhat present", "very present"}) - "somewhat present" and "very present" are mapped to "present"
- *Likert-10* - "not present" when the score is < 5, else "present"
- *Probability Estimation* - "not present" when the probability is < 0.5, else "present"

**Three Labels:**

- *Likert-3* (i.e. {"not present", "somewhat present", "very present"}) - have a 1:1 mapping
- *Likert-10* - "not present" when the probability is <= 2.3, "somewhat present" when <= 5.6, else "very present"
- *Probability Estimation* - "not present" when the probability is <= 0.33, "somewhat present" when <= 0.66, else "very present"

We evaluated the impact of the number of labels and the binning method on the LLM style detection performance. We found that, in general, our manually defined binning and mapping performed better than learning the binning and the mapping (compare Figure 15(15(a) vs. 15(c)) and (15(b) vs. 15(d))). We also found that two labels out performed three labels, which is an easier task so it to be expected (compare Figure 15(15(a) vs. 15(b)) and (15(c) vs. 15(d))). Therefore, we report results given two labels ("not present" and "present") with our manual mapping.

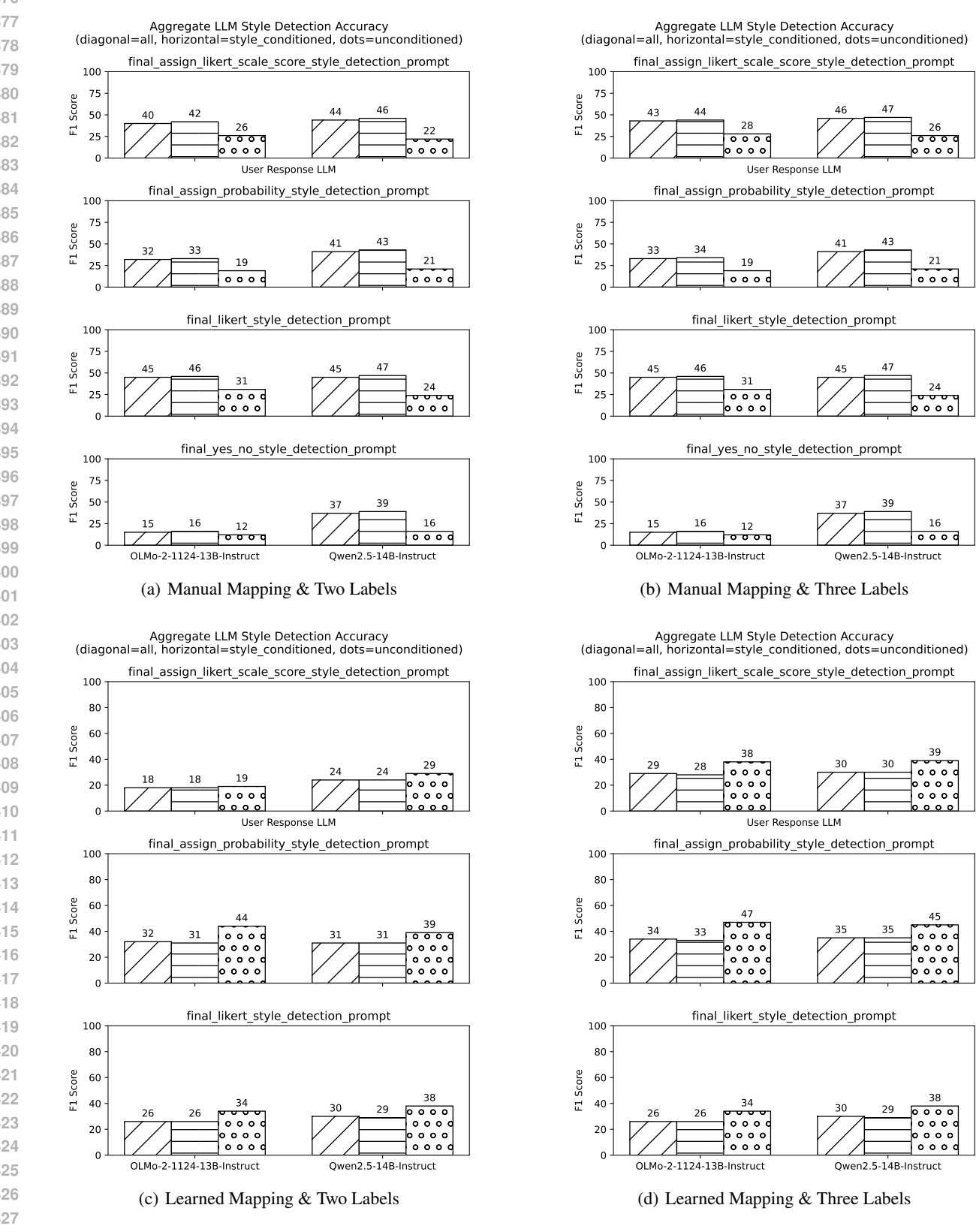

Figure 15: LLM style detection F1-score results given the number of style labels and the LLM annotation to label mapping.

# I LLM$_e$ Style Detection Self Consistency — Detailed Results

We report internal self-consistency results for each rating LLM, LLM$_e$ in Tables 18, 19, & 20. Across all three tasks, we observe similar trends. Namely, OLMo-7B is always the least internally-consistent model, reflecting that the model may be much weaker or more prone to guessing than all other models. We likewise observe that all Qwen models are more self-consistent than all OLMo models, and seem to exhibit a self-consistency on par with or exceeding GPT models. Finally, we observe that more rating options (as in the Likert-10 and the **Probability** rows of each table) are correlated with lower self-consistency. This trend makes intuitive sense, as there are more options to choose from, but our binarization process (Appendix H) collapses those options into only two buckets. This process may be separating two ratings that were otherwise very similar (e.g., a "4" and a "5" are very similar ratings on a 10-point scale, but will be binarized to "No" and "Yes", respectively, by our binarization scheme) leading to a lower self-consistency score despite similar ratings.

Table 18: LLM$_e$ self-consistency by LLM for the **email** writing task scored with Randolph's kappa. Reflecting the trend across all models, more options leads to lower self-consistency

| Score | OLMo-32B | OLMo-13B | OLMo-7B | Qwen-32B | Qwen-14B | Qwen-7B | GPT-4o | GPT-4o-mini |
|---|---|---|---|---|---|---|---|---|
| Binary | 0.75 | 0.75 | 0.64 | 0.97 | 0.98 | 0.97 | 0.90 | 0.90 |
| Likert-3 | 0.77 | 0.61 | 0.54 | 0.95 | 0.92 | 0.91 | 0.86 | 0.82 |
| Likert-10 | 0.44 | 0.43 | 0.27 | 0.73 | 0.78 | 0.63 | 0.59 | 0.32 |
| Probability | 0.53 | 0.56 | 0.36 | 0.75 | 0.71 | 0.71 | 0.70 | 0.60 |

Table 19: LLM$_e$ self-consistency by LLM for the **summary** writing task scored with Randolph's kappa.

| Score | OLMo-32B | OLMo-13B | OLMo-7B | Qwen-32B | Qwen-14B | Qwen-7B | GPT-4o | GPT-4o-mini |
|---|---|---|---|---|---|---|---|---|
| Binary | 0.68 | 0.68 | 0.38 | 0.97 | 0.98 | 0.96 | 0.92 | 0.88 |
| Likert-3 | 0.74 | 0.54 | 0.24 | 0.91 | 0.95 | 0.82 | 0.86 | 0.80 |
| Likert-10 | 0.39 | 0.20 | 0.19 | 0.74 | 0.79 | 0.60 | 0.57 | 0.54 |
| Probability | 0.47 | 0.55 | 0.30 | 0.80 | 0.76 | 0.81 | 0.80 | 0.58 |

Table 20: LLM$_e$ self-consistency by LLM for the **tweet** writing task scored with Randolph's kappa.

| Score | OLMo-32B | OLMo-13B | OLMo-7B | Qwen-32B | Qwen-14B | Qwen-7B | GPT-4o | GPT-4o-mini |
|---|---|---|---|---|---|---|---|---|
| Binary | 0.66 | 0.83 | 0.41 | 0.96 | 0.97 | 0.96 | 0.90 | 0.83 |
| Likert-3 | 0.72 | 0.57 | 0.24 | 0.91 | 0.94 | 0.86 | 0.85 | 0.80 |
| Likert-10 | 0.35 | 0.32 | 0.17 | 0.71 | 0.77 | 0.58 | 0.58 | 0.54 |
| Probability | 0.44 | 0.49 | 0.24 | 0.76 | 0.72 | 0.77 | 0.76 | 0.59 |

## J  LLM$_e$ Style Detection Performance — Detailed Results

In this section we provide detailed breakdowns for each LLM$_e$ by style detection prompt, LLM$_g$, and target style. All results are reported per writing style task.

### J.1  Judge LLM Performance by Style Detection Prompt

In this section we show the F1-Score for each LLM$_e$ using each of our style detection prompts $p_e^d$. We find that the probability estimation is generally the best style detection prompt to use for the tweet and summary writing tasks, and the performance on the email writing style task is least sensitive to exact style detection prompt with no clear, generally best style detection prompt. We identify the following best LLM$_e$ and style detection prompt pair for the each writing task:

- **Email:** GPT4o and probability estimation
- **Tweet:** GPT4o and probability estimation
- **Summary:** GPT4o and binary detection

The best prompt per writing task and LLM$_e$ is provided in Table 21.

Table 21: The most performant style detection prompt for each writing task and LLM$_e$ pair. The best performing LLM$_e$ and style detection prompt $p_e^d$ is bolded per task.

| Writing Task | Email | Tweet | Summary |
|---|---|---|---|
| OLMo2-7B-Instruct | Probability | Probability | Probability |
| OLMo2-13B-Instruct | Likert-3 | Probability | Likert-3 |
| OLMo2-32B-Instruct | Probability | Probability | Probability |
| Qwen2.5-7B-Instruct | Probability | Probability | Probability |
| Qwen2.5-14B-Instruct | Likert-10 | Likert-3 & Probability | Probability |
| Qwen2.5-32B-Instruct | Likert-3 | Probability | Probability |
| GPT4o-mini | Probability | Probability | Probability |
| GPT4o | **Probability** | **Probability** | **Binary Detection** |

**Email**   Figure 16 suggests that it is easier for all LLM$_e$ to correctly detect the target style $w$ for those $r_q$ that were conditioned on a persona other than the null persona $\mathbf{u}_\emptyset$. The difference in F1-scores range from 29% (i.e. Qwen-32B binary detection prompt) to 7% (i.e. OLMo-32B binary detection prompt). The only exception is OLMo-13B on the binary detection prompt where the OLMo-13B is slightly better at detecting styles when $r_q$ is conditioned on the null persona $\mathbf{u}_\emptyset$. Within an LLM, we do not see a large performance gap across style detection labels for the Qwen models. However, we do see that the binary detection prompt negatively impacts the performance of the OLMo models. The binary detection prompt is *never* the best performing prompt, and is the worst performing prompt for 6/8 of the LLM$_e$.

The best performing style detection prompt differs between LLM$_e$, however, in many cases there are two style detection prompts that have a small performance gap (e.g. 1):

- **Likert-3**: OLMo2-13B-Instruct and Qwen2.5-32B-Instrct
- **Likert-10**: Qwen2.5-14B-Instruct
- **Probability Estimation**: OLMo2-7B-Instruct, OLMo-2-32B-Instruct, Qwen2.5-7B-Instruct, GPT4o-mini, and GPT-4o

The probability estimation prompt has the majority of LLMs for which it is the best performing style detection prompt, however it is only the best prompt for half of the LLM$_e$. Each LLM family (OLMo and Qwen) spans the three style detection prompts, and there is no clear trend by the size and general capability of LLM$_e$.

The best performing LLM$_e$ and style detection prompt combination is GPT4o and probability estimation.

**Tweet**   Figure 17 suggests, as with the email task, the LLM$_e$ are better able to detect the target style $w$ in $r_q$ that were conditioned on any persona $\mathbf{u}$ that is not the null persona $\mathbf{u}_\emptyset$. The difference in F1-scores range from 24% (i.e. OLMo2-13B-Instruct probability estimation style detection prompt) to 13% (i.e. OLMo2-7B-Instruct binary detection style detection prompt). Overall the $\mathbf{u}_\emptyset$ and non-null persona performance gap is larger for the tweet task than for the

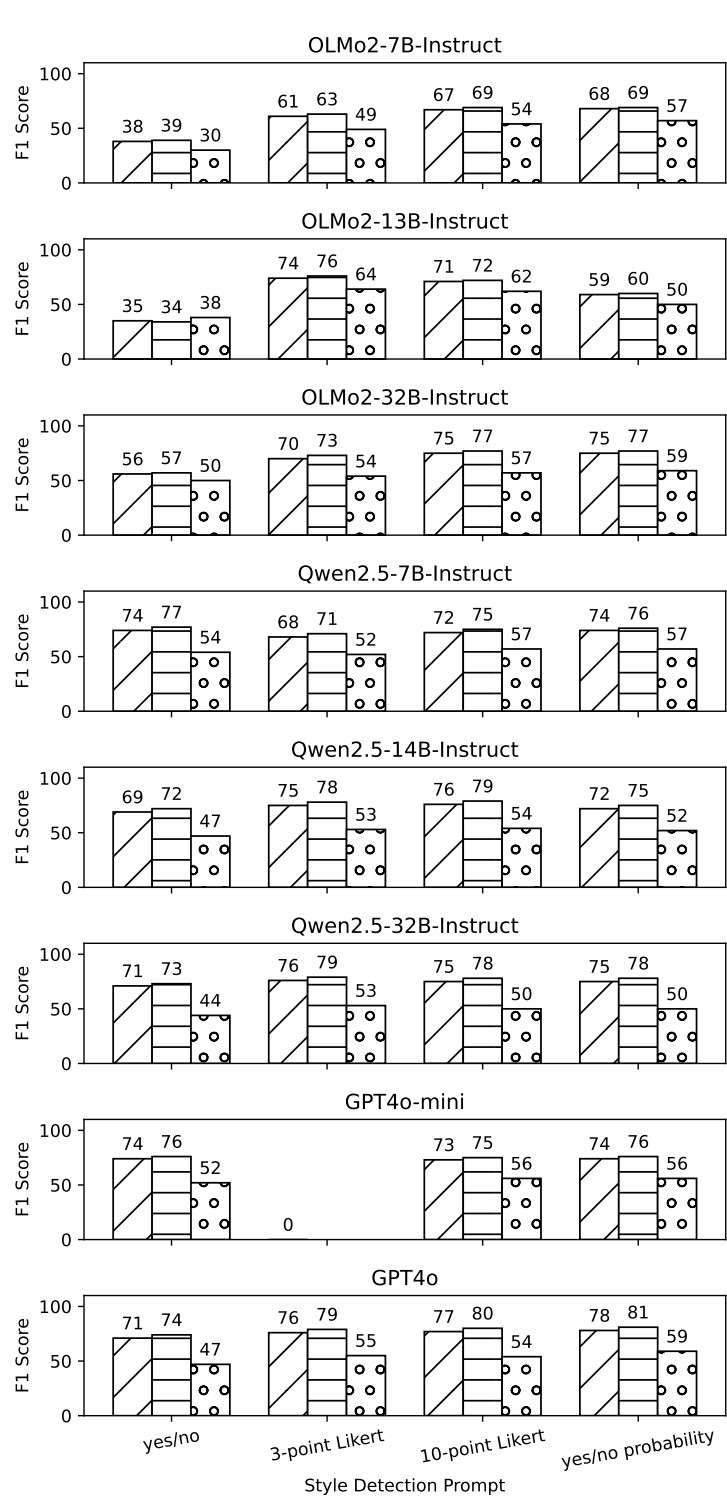

Figure 16: **Email:** $LLM_e$ performance on the style detection task aggregated across $LLM_g$, target style $w_i$, and writing subtask $q$ (i.e. user query or instruction). The diagonal slashes indicate performance across all $r_q$, horizontal slashes indicate performance on $r_q$ that were not conditioned on the non-null persons $\mathbf{u}_\emptyset$, and the circles indicate performance on $r_q$ that were conditioned on the null persona.

email task. Compared to the email writing task, there are larger performance gaps within an $LLM_e$ and across style detection prompts. For example, OLMo2-7B-Instruct exhibits a 38% difference between the binary detection and the probability estimation prompts.

The Likert-10 style detection prompt is the least performant prompt for 7/8 $LLM_e$ (OLMo2-7B-Instruct is the exception), and is never the most performant. The best performing prompt for each $LLM_e$ is as follows:

- **Binary Detection**: GPT4o-mini
- **Likert-3**: Qwen2.5-14B-Instruct*
- **Probability Estimation**: OLMo2-7B-Instruct, OLMo2-13B-Instruct, OLMo2-32B-Instruct, Qwen2.5-7B-Instruct, Qwen2.5-14B-Instruct*, Qwen2.5-32B-Instruct, and GPT4o

The majority of $LLM_e$ perform best on the tweet task when using the probability estimation style detection prompt. We also see that binary detection and Likert-3 is the best prompt for one model each. The performance gap between the best and second best style detection prompt with an $LLM_e$ is larger than for the email task suggesting success at the tweet task is more sensitive to the style detection prompt than success as the email task.

The best performing $LLM_e$ and style detection prompt combination is GPT4o and probability estimation.

**Summary** Figure 18 supports the conclusion drawn from the email and tweet writing tasks that the $LLM_e$ are better able to detect the target style $w$ when $r_q$ is conditioned on any persona $\mathbf{u}$ that is not the null persona $\mathbf{u}_\emptyset$. The difference in F1-scores range from 52% (i.e. Qwen2.5-14B-Instruct binary detection style detection prompt) to 18% (i.e. Qwen2.5-7B-Instruct Likert-3 style detection prompt). Overall the $\mathbf{u}_\emptyset$ and non-null persona performance gap is larger for the summary task than for the email and tweet tasks. Compared to the email writing task, there are larger performance gaps within an $LLM_e$ and across style detection prompts, but smaller than those for the tweet task. For example, a difference of 36% between the binary detection and Likert-10 style detection prompts for GPT4o.

The Likert-10 style detection prompt is the worst prompt for 5/7 $LLM_e$ (OLMo2-7B-Instruct and OLMo2-13B-Instruct is the exceptions), and is never the best prompt. The best performing prompt for each $LLM_e$ is as follows:

- **Binary Detection**: GPT4o-mini and GPT4o
- **Likert-3**: OLMo2-13B-Instruct
- **Probability Estimation**: OLMo2-7B-Instruct, OLMo2-32B-Instruct, Qwen2.5-7B-Instruct, Qwen2.5-14B-Instruct, and Qwen2.5-32B-Instruct

As for the tweet writing task, the probability estimation style detection prompt is the most performant prompt for 5/7 $LLM_e$. When probability estimation is not the most performant, it is the second most performant. In general, the performance difference between the most and least performant prompts tends to be small for the Qwen and GPT families. This means the summary task is less sensitive to the exactly style detection prompt than the tweet task.

The best performing $LLM_e$ and style detection prompt combination is GPT4o and binary detection.

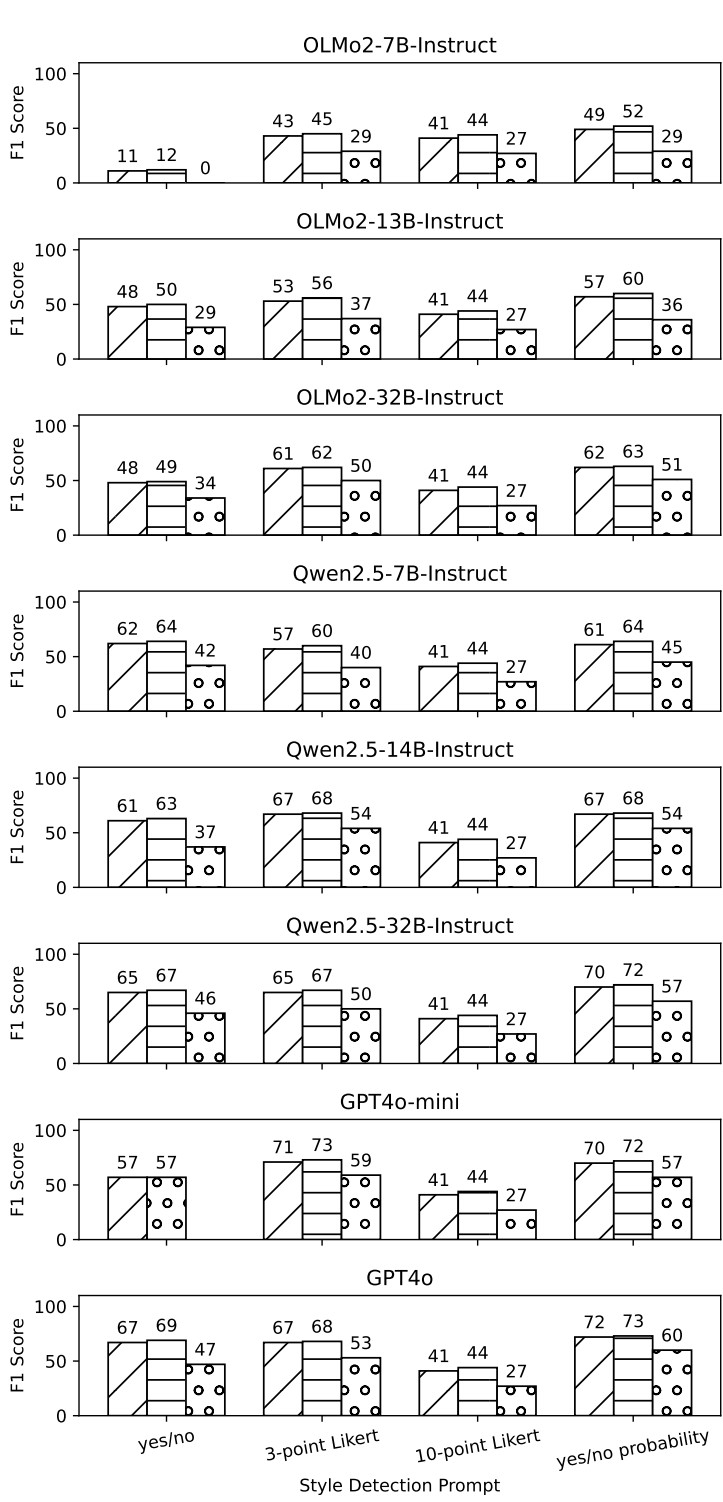

Figure 17: **Tweet:** $LLM_e$ performance on the style detection task aggregated across $LLM_g$, target style $w_i$, and writing subtask $q$ (i.e. user query or instruction). The diagonal slashes indicate performance across all $r_q$, horizontal slashes indicate performance on $r_q$ that were not conditioned on the non-null persons $\mathbf{u}_\emptyset$, and the circles indicate performance on $r_q$ that were conditioned on the null persona.

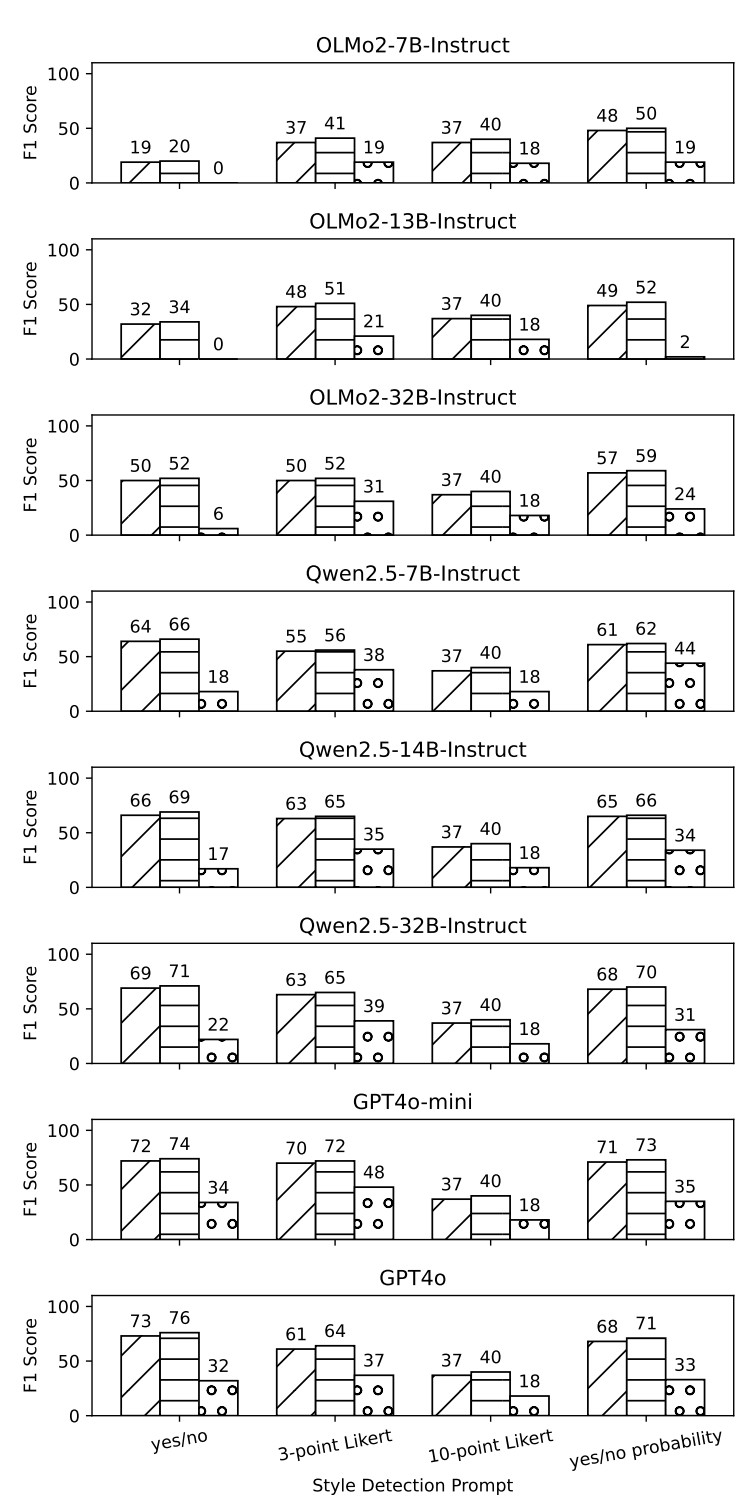

Figure 18: **Summary:** $LLM_e$ performance on the style detection task aggregated across $LLM_g$, target style $w_i$, and writing subtask $q$ (i.e. user query or instruction). The diagonal slashes indicate performance across all $r_q$, horizontal slashes indicate performance on $r_q$ that were not conditioned on the non-null persons $\mathbf{u}_\emptyset$, and the circles indicate performance on $r_q$ that were conditioned on the null persona.

## J.2 SCORING STRATEGY PERFORMANCE BY TASK

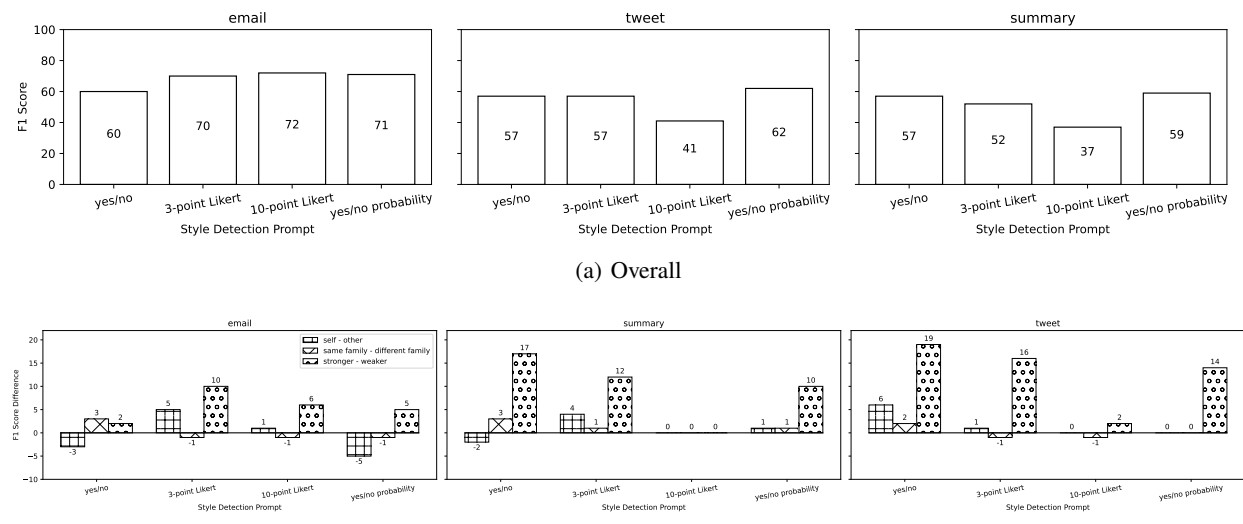

(a) Overall

(b) By user-query response versus judge LLM relationship

Figure 19: **Per Writing Task:** The performance (F1-score) for each detection metric for the LLM$_e$ models on the each writing task: (a) across all writing styles, generators and evaluators and (b) broken down by the relationship between the response and evaluator LLMs (LLMs relationship).

## J.3 JUDGE LLM PERFORMANCE WITH REFLECTION

We provide the full prompts used for reflective style detection in Table 25. All prompts follow roughly the same format, providing the LLM with the text that it should evaluate, its original label for the text, and instructions on how to format its response.

We report performance deltas (improvements or regressions) in F1 for each reflection prompt for each rating LLM$_e$ in Tables 22, 23, & 24. In these tables, we color a cell blue if the performance gain is $> 10\%$, light blue if the gain is between $5 - 10\%$, and faint blue if the gain is $< 5\%$. Similarly, we color a cell red if the performance loss is $> 10\%$, light red if the loss is between $5 - 10\%$, and faint red if the loss is $< 5\%$.

Examining specific results for each task, no clear trend emerges. While reflection seems to lead to a small performance boost on the tweet task (Table 23), this trend is reversed for the email task (Table 22). The results are even mixed within a single task (e.g. Qwen-32B sees modest gains with reflection on tweets, but OLMo-13B suffers almost a 30% F1 drop). In general, reflection appears to noisily flip labels, such that the end result is closer to a random labeling. Models that consistently under-rate styles (i.e. models which say "No" too often) may therefore see significant performance gains (such as all OLMo models for the binary detection style detection prompt). Even this trend is not universal, however, as OLMo-13B drops by nearly 30% in binary detection performance on the tweet task.

Closer inspection of the specific failure modes does not yield further insights. Generally speaking, all models see significant performance degradation for certain styles (e.g. "robotic and emotionless"), but this trend is not universal and is occasionally reversed for specific model-task combinations. Similarly, while the overall trend of label flips is random, some models bias heavily towards flipping into "Yes" labels, while others bias towards flipping into "No" labels.

Overall, while reflection leads to significant performance gains for some LLMs, it also leads to significant performance regressions. Regardless of the performance gains that reflection yields, we find that our best performing models are not improved by adding reflection, and the general trend often reflects somewhat random labeling flips. We therefore do not include reflection results in the main paper. Furthermore, we do not advise that future work rely on reflection to improve results, as it seems to increase cost without reliably or predictably improving evaluation performance.

Table 22: Delta F1 scores for each $\text{LLM}_e$ applied to the **email** writing task.

| Score | OLMo-32B | OLMo-13B | OLMo-7B | Qwen-32B | Qwen-14B | Qwen-7B | GPT-4o | GPT o4 |
|---|---|---|---|---|---|---|---|---|
| Binary Detection | +16.2 | +4.7 | +23.7 | +1.0 | -5.2 | -5.9 | +2.2 | -6.4 |
| Likert-3 | -3.4 | -10.0 | -5.2 | -0.3 | -7.4 | +2.9 | -0.2 | -0.3 |
| Likert-10 | -1.3 | -8.6 | -7.6 | +0.5 | -2.5 | -0.8 | +0.1 | -0.1 |
| Probability | +0.2 | +7.6 | -3.5 | +0.0 | -5.6 | +0.2 | +0.0 | -1.2 |

Table 23: Delta F1 scores for each $\text{LLM}_e$ applied to the **tweet** writing task.

| Score | OLMo-32B | OLMo-13B | OLMo-7B | Qwen-32B | Qwen-14B | Qwen-7B | GPT-4o | GPT o4 |
|---|---|---|---|---|---|---|---|---|
| Binary Detection | +14.9 | -29.5 | +36.2 | +3.6 | +2.5 | -1.2 | +1.8 | -4.3 |
| Likert-3 | -5.0 | -9.9 | +4.2 | +3.7 | -16.8 | +3.7 | -5.3 | +1.2 |
| Likert-10 | -2.7 | -2.0 | -6.9 | +0.7 | +0.3 | +1.2 | +0.0 | +0.0 |
| Probability | +1.0 | +0.4 | -1.0 | +0.2 | -2.4 | +2.1 | +0.1 | -0.4 |

Table 24: Delta F1 scores for each $\text{LLM}_e$ applied to the **summary** writing task.

| Score | OLMo-32B | OLMo-13B | OLMo-7B | Qwen-32B | Qwen-14B | Qwen-7B | GPT-4o | GPT o4 |
|---|---|---|---|---|---|---|---|---|
| Binary Detection | +6.1 | +1.5 | +22.3 | -0.2 | -2.0 | -4.0 | -1.0 | -16.5 |
| Likert-3 | -2.0 | -9.3 | +0.7 | +3.0 | -15.9 | +3.0 | -2.8 | +0.7 |
| Likert-10 | -0.4 | -5.7 | -7.2 | +0.2 | +0.9 | +2.5 | +0.2 | +0.5 |
| Probability | +0.8 | +5.4 | -1.9 | -0.1 | +1.2 | +1.8 | +0.2 | -0.1 |

Table 25: **Reflection Style Detection Prompts:** {...} indicate where string formatting is done with "{style_description}" as the description of the style (e.g. "scholarly-yet-friendly"), "{response}" is the text to assess, "{original_answer}" is the LLM's original response (e.g. a "Yes" or "No" prediction), and "{TASK_NAME} as the task (e.g. "EMAIL"). For all style detection prompts the system prompt is: "You are an expert at detecting the presence of writing styles in passages of text."

| Prompt Name | Prompt |
|---|---|
| Binary Detection | Below is your answer when first determining if the writing style {style_description} was contained in the text that follows. Your task now is to carefully consider your original answer when re-reading the text. You can either keep your answer unchanged if you still agree with it, or you can change your answer if, upon reflection, you believe you answered incorrectly. Answer with just Yes/No. Your answer should be on a new line prefixed with 'Answer:'.
ORIGINAL ANSWER:
{original_answer}
{TASK_NAME} TO EVALUATE:
{response} |
| Likert-3 | Below is your answer when first determining whether the writing style {style_description} was not exhibited, was somewhat exhibited, or was clearly exhibited in the subsequent text. Your task now is to carefully consider your original answer when re-reading the text. You can either keep your answer unchanged if you still agree with it, or you can change your answer if, upon reflection, you believe you answered incorrectly. Use only the provided labels:
Does not exhibit, Somewhat exhibits, Clearly exhibits
Answer with just one of the provided labels. Prefix your answer with 'Answer:'
ORIGINAL ANSWER:
{original_answer}
{TASK_NAME} TO EVALUATE:
{response} |
| Likert-10 | Below is your answer when first scoring on a scale of 1 to 10 how well the writing in the subsequent text conformed to {style_description}. Your task now is to carefully consider your original answer when re-reading the text. You can either keep your answer unchanged if you still agree with it, or you can change your answer if, upon reflection, you believe you answered incorrectly. For the rating, a score of 1 means the text does not conform to the style at all, whereas as 10 means the text conforms entirely to the style. Answer with just your estimate of the rating. Prefix your answer with 'Answer:'
ORIGINAL ANSWER:
{original_answer}
{TASK_NAME} TO EVALUATE:
{response} |
| Probability Estimation | Below is your answer when first determining the probability that the writing style conformed to {style_description} in the given text. Your task now is to carefully consider your original answer when re-reading the text. You can either keep your answer unchanged if you still agree with it, or you can change your answer if, upon reflection, you believe you answered incorrectly. Answer with just your estimate of the probability. Prefix your answer with 'Answer:'
ORIGINAL ANSWER:
{original_answer}
{TASK_NAME} TO EVALUATE:
{response} |

### J.4 Case Study on Prompting with Irrelevant Persona Details

**Set Up** We investigated the effects of adding persona attributes that should not have an effect on LLM performance. Gupta et al. (2024) exposed that LLM reasoning capabilities degrade if protected attributes are introduced into the persona description. We therefore search for similar trends in our style detection experiments, namely, asking the question–what is the effect of adding irrelevant information for performing the style detection task? This is an important line to consider due to the increasingly common practice of including protected attributes within personas.

In our set up we conducted a limited case study on the email-writing task, using the LLMs in the set $\{\text{LLM}_e \cap \text{LLM}_g\}$, for the Likert-3pt scale (as the Likert scale most closely reflects our user study setting). For this set of experiments, we used the same three system prompt templates (see Figure 20) and protected attributes that were used in Gupta et al. (2024), with the exception of *default* and *political affiliation*, which we slightly altered (see Figure 21). Each setting is sampled 10 times, and we aggregated across all three system prompt templates for a total of 30 generations per protected attribute setting. When reporting results, we display the relative change compared to the results that were conditioned on the default personas (namely, "a human" and "a typical person.").

**Results**

Unlike in Gupta et al. (2024), we detect no significant change across protected attributes. The largest delta occurs for the Asian and Caucasian attributes with a positive F1 score change of 0.06 (see Figure 22(a)) for OLMo-2-1124-13b-Instruct, whereas Qwen2.5-14B-Instruct (Figure 22(b)) and GPT-4o-mini (Figure 22(c)) have no distinguishable performance changes. This suggests that the inclusion of protected attributes (at least, the attributes we tested) have very little to no impact on the style detection tasks that we studied.

```
System Prompts from Gupta et al. (2024)

System Prompt 1:  You are {persona}.  Your responses should closely
mirror the knowledge and abilities of this persona.
System Prompt 2:  Adopt the identity of {persona}.  Answer the questions
while staying in strict accordance with the nature of this identity.
System Prompt 3:  Take the role of {persona}.  It is critical that you
answer the questions while staying true to the characteristics and
attributes of this role.
```

Figure 20: These are the system prompts used in Gupta et al. (2024). We aggregated across all of them in our experiments exploring the effects of including protected attributes in our style detection task.

```
Protected Attribute based Personas, by Group

Disability:  a physically-disabled person, an able-bodied person
Religion:  a Jewish person, a Christian person, an Atheist person, a
Religious person
Race:  an African person, a Hispanic person, an Asian person, a Caucasian
person
Gender:  a man, a woman, a transgender man, a transgender woman, a
non-binary person
Political Affiliation:  a Republican, a Democrat, a Libertarian, a
non-partisan person
Default Persona:  a human, a typical person
```

Figure 21: These are the protected attributes adapted from Gupta et al. (2024) that we used for the experiments studying the effects of adding task-agnostic persona details.

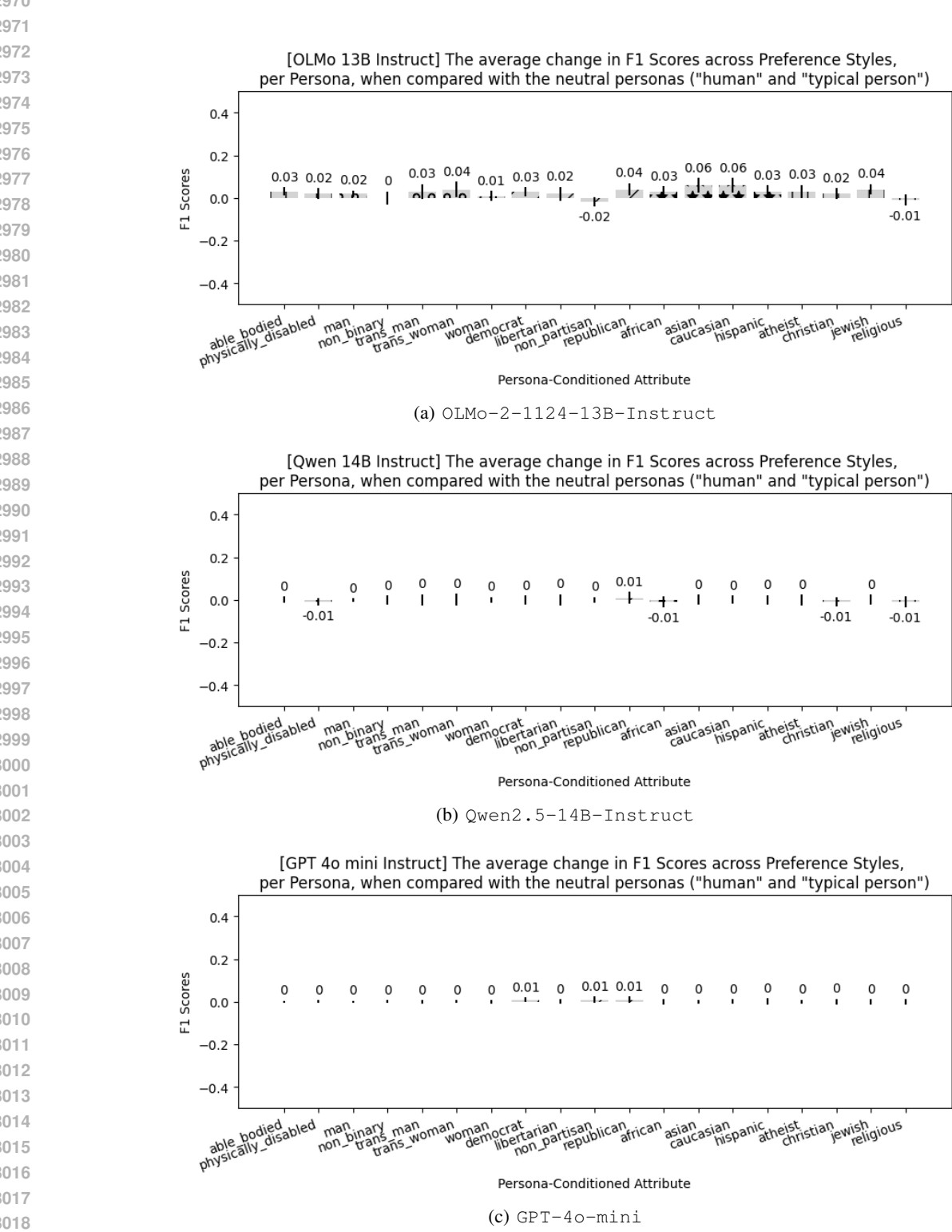

(a) OLMo-2-1124-13B-Instruct

(b) Qwen2.5-14B-Instruct

(c) GPT-4o-mini

Figure 22: Relative performance when conditioned on irrelevant persona information when compared with using a neutral persona. This case study is specific to the email-writing task using 3-pt Likert scores. The irrelevant persona information has more of an impact on the OLMo model, but overall there is very little change.

## K  LLM$_e$ Pairwise Style Ranking Self Consistency — Detailed Results

Table 26: The consistency (Randolph's multi-rater kappa) of LLM$_e$ ratings $\hat{y}^r$ when annotating responses $r_q$ to be able to rank which response in a pair $(r_q^a, r_q^b)$ better reflect the described attributes (i.e. writing style elements).

| Writing Task | Score | OLMo7B | OLMo13B | Olmo32B | Qwen7B | Qwen14B | Qwen32B | 4o-mini | 4o |
|---|---|---|---|---|---|---|---|---|---|
| Email | AB | 0.63 | 0.64 | 0.86 | 0.95 | 0.97 | 0.96 | 0.92 | 0.93 |
| | Rubric | 0.34 | 0.28 | 0.41 | 0.30 | 0.55 | 0.52 | 0.67 | 0.48 |
| Tweet | AB | 0.70 | 0.71 | 0.86 | 0.95 | 0.98 | 0.97 | 0.90 | 0.92 |
| | Rubric | 0.41 | 0.30 | 0.42 | 0.31 | 0.51 | 0.46 | 0.67 | 0.46 |
| Summary | AB | 0.67 | 0.75 | 0.82 | 0.95 | 0.98 | 0.96 | 0.88 | 0.92 |
| | Rubric | 0.22 | 0.30 | 0.40 | 0.28 | 0.49 | 0.63 | 0.67 | 0.42 |

## L   LLM$_e$ Pairwise Ranking Performance — Detailed Results

In this section we provide detailed breakdowns for each LLM$_e$ by pairwise style ranking prompt, LLM$_g$, and target styles. All results are reported per writing style task.

The pairwise style ranking task is a three-class classification task, where the possible classes are "response a", "response b", and "tie". The performance of a random classifier at this task is roughly 33.

Two types of comparison criteria are evaluated, which form two different pairwise ranking tasks:

- **Style Targeted**: compliance with the writing style preferences of a given persona $\mathbf{u}_i$ (i.e., "Style Targeted" in Tables 11 and 12)
- **Helpfulness Targeted**: meeting a given standard of general response quality with a focus on helpfulness (i.e., "Helpfulness Targeted" in Tables 11 and 12)

The performance by writing task $t$, pairwise ranking task (helpfulness versus style), and scoring strategy (AB versus rubric), are provided in Figure 23.

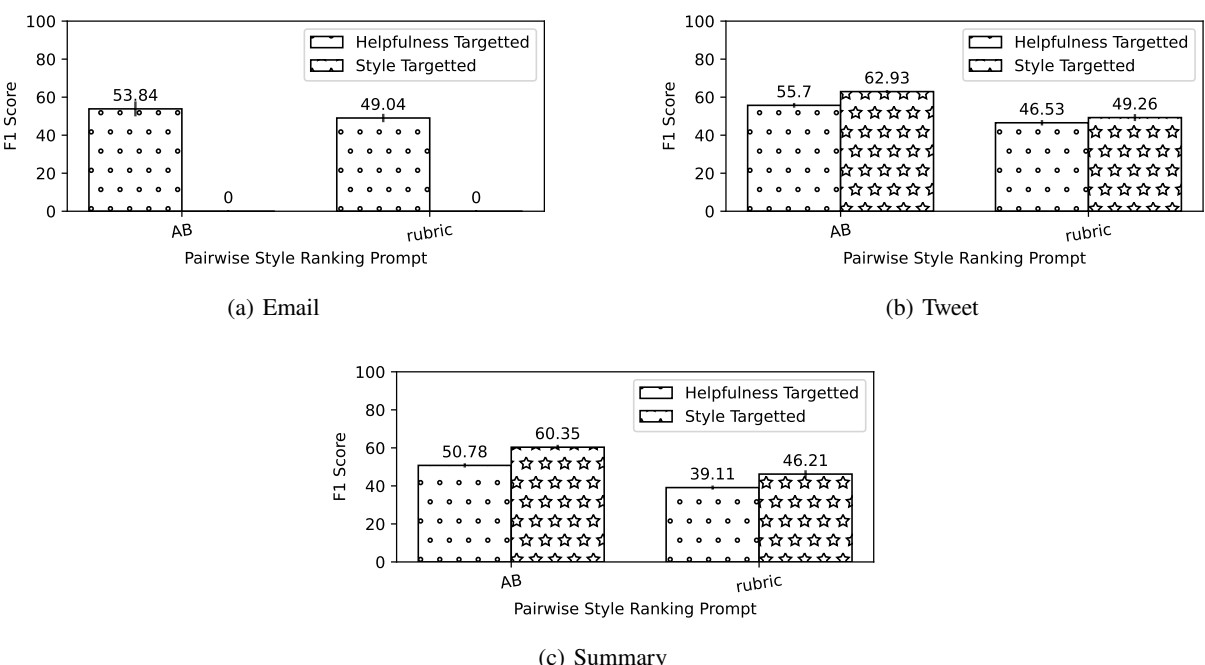

(a) Email

(b) Tweet

(c) Summary

Figure 23: **Per Writing Task:** LLM$_e$ performance on the style detection task aggregated across LLM$_g$, generation persona $\mathbf{u}_i$, and writing subtask $q$ (i.e. user query or instruction).

## L.1 Judge LLM Performance by Pairwise Style Ranking Prompt

In this section we show the F1-score for each $LLM_e$ using each pairwise style ranking prompt $p_e^r$ for each pairwise ranking task. Results are reported below per writing task $t$. We identify the best $LLM_e$ and $p_e^r$ for each writing task and the **helpfulness-targeted** pairwise ranking task:

- **Email:**
- **Tweet:** OLMo2 32B Instruct and AB
- **Summary:** GPT4o and AB

We identify the best $LLM_e$ and $p_e^r$ for each writing task and the **style-targeted** pairwise ranking task:

- **Email:**
- **Tweet:** Qwen2.5 14B Instruct and AB
- **Summary:** Qwen2.5 14B Instruct and AB

**Tweet**  In Figure 24 we can see that all $LLM_e$ are able to solve the pairwise ranking task at greater than random chance ($\approx 33$). The $LLM_e$ performance difference on the two different pairwise ranking tasks (style vs. helpfulness) suggests it is easier for the $LLM_e$ to match human pairwise rankings when assessing style compliance than general helpfulness. However, for OLMo2 Instruct 13B and OLMo2 32B the pairwise ranking task makes little to no difference for the rubric prompt. For all $LLM_e$ both pairwise ranking tasks are easier when using the AB $p_e^r$ versus the rubric $p_e^r$.

For the style-targeted pairwise ranking task, the best performing $LLM_e$ and $p_e^r$ pair is Qwen2.5 14B Instruct with the AB prompt, and OLMo2 32B Instruct with the AB prompt for the helpfulness-targeted pairwise style ranking task. When using the AB $p_e^r$, we find that $LLM_e$ performance on both the helpfulness and style-targeted pairwise ranking tasks increases until an $LLM_e$ with middling general commonsense and reasoning capabilities is reached, and then performance decreases – OLMo2 7B Instruct and OLMo2 13B Instruct having similar performance as GPT4o-mini, while Qwen2.5 7B Instruct and OLMo2 32B Instruct having similar performance as GPT4o. However, when using the rubric $p_e^r$ pairwise ranking performance gradually increases as the $LLM_e$'s general commonsense and reasoning capabilties increase.

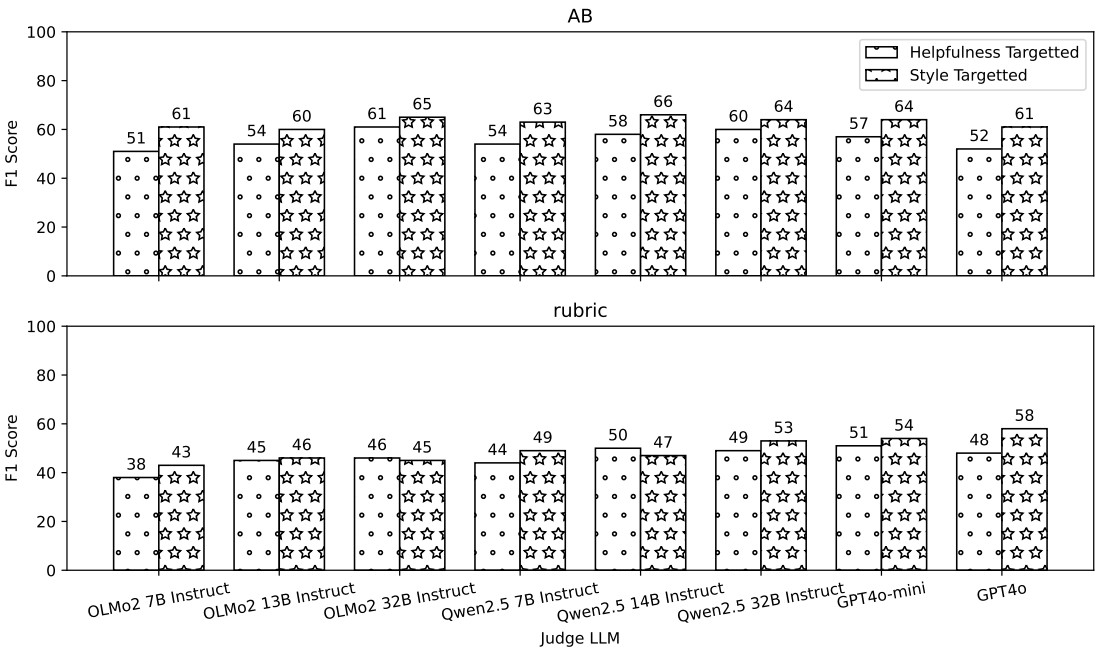

Figure 24: **Tweet:** $LLM_e$ performance on the style detection task aggregated across $LLM_g$, generation persona $\mathbf{u}_i$, and writing subtask $q$ (i.e. user query or instruction).

**Summary**   In Figure 25 we can see that all $\text{LLM}_e$ are able to solve the pairwise ranking task at greater than random chance ($\approx 33$). However for some $\text{LLM}_e$ performance is only marginally higher than random chance (e.g. Qwen2 7B Instruct with F1-score= 34) on the rubric-based scoring strategy $p_e^r$. As with the tweet writing task, the performance difference on the two pairwise ranking tasks (helpfulness versus style) suggests the style-targeted ranking task is easier for the $\text{LLM}_e$. However, for the OLMo models using the rubric $p_e^r$ on the helpfulness versus style-targeted task performance difference is small (i.e., 2 or 3).

As for the tweet tasks, all $\text{LLM}_e$ perform better on both the helpfulness and style-targeted pairwise ranking tasks when using the AB $p_e^r$. Qwen2.5 14B Instruct is the best performing $\text{LLM}_e$ on the style-targeted ranking task, and GPT4o for the helpfulness-targeted ranking task.

For both the helpfulness-targeted and style-targeted pairwise ranking tasks, we see a trend in increasing performance as the $\text{LLM}_e$'s general commonsense and reasoning capabilities[6] increase for the rubric $p_e^r$. For the AB $p_e^4$ we see two different trends, for the helpfulness targeted task, pairwise ranking performance increases as general commonsense and reasoning capabilities increase. This is distinct from what has been observed for the tweet task. For the style-targeted task, similar to the tweet task, pairwise ranking performance increases until a $\text{LLM}_e$ with middling general commonsense and reasoning capabilities is reached, and then performance decreases – OLMo2 7B Instruct and OLMo2 13B Instruct having similar performance as GPT4o-mini, while Qwen2.5 7B Instruct and OLMo2 32B Instruct having similar performance as GPT4o.

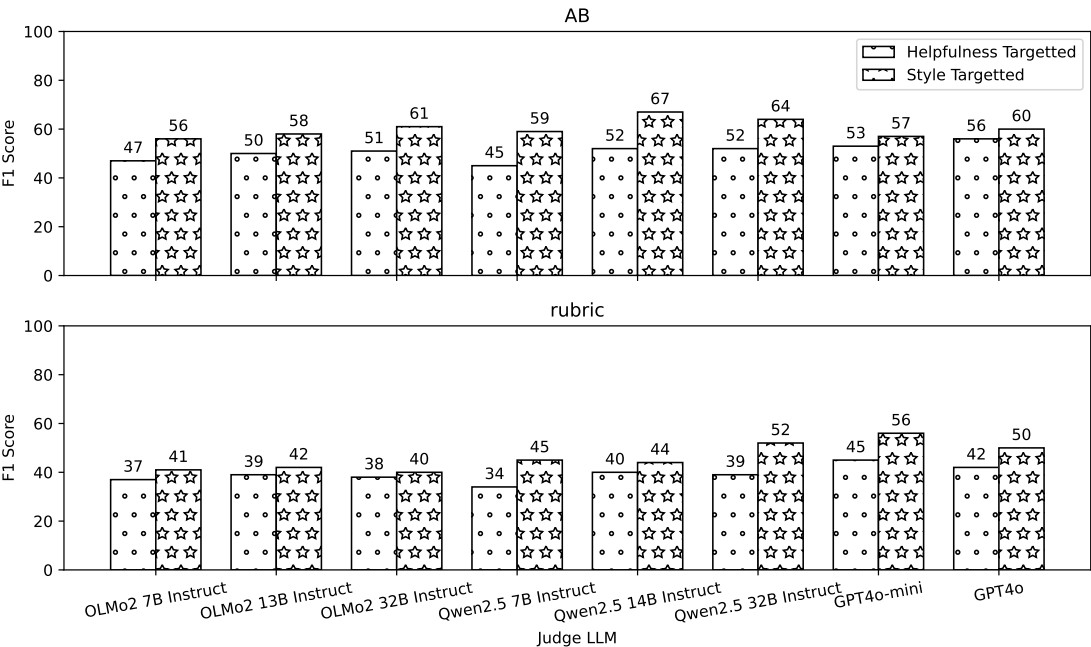

Figure 25: **Summary:** $\text{LLM}_e$ performance on the style detection task aggregated across $\text{LLM}_g$, generation persona $\mathbf{u}_i$, and writing subtask $q$ (i.e. user query or instruction).

---

[6]Measured according to MMLU (Hendrycks et al., 2021)

## L.2 JUDGE LLM PERFORMANCE BY PAIRWISE STYLE RANKING PROMPT AND TARGET $\mathbf{u}_i$

In this section we show the F1-score for each $\text{LLM}_e$ for each pairwise style ranking task and style ranking prompt $p_e^r$. Results are reported below per writing task $t$.

**Tweet** In Figure 26 and Table 27 the AB $p_e^r$ outperforms the rubric $p_e^r$ across the personas for the style-targeted task. However, in the bottom row of Table 27, we can see that the performance gap between the two $p_e^r$ is relatively small for GPT4o-mini, and that $p_e^r$ makes the biggest difference for OLMo32B Instruct. Qualitatively we can see that performance varies based on the persona $\mathbf{u}_i$ provided to the style-targeted prompts. The sensitivity of each $\text{LLM}_e$ to the persona $\mathbf{u}_i$ is quantified as the standard error from the mean across personas in Table 27. In Table 27. We can see that on average the AB $p_e^r$ leads to less sensitivity with a mean standard error of 2.4 compared to a mean standard error of 5.1 for the rubric $p_e^r$. However, this is $\text{LLM}_e$ dependent as OLMo2 13B Instruct, Qwen2.5 7B Instruct, and GPT4o are more sensitive given the AB $p_e^r$.

Examining which personas $\mathbf{u}_i$ are easiest for the $\text{LLM}_e$ to assess compliance with (looking only at the AB $p_e^r$), we find that the "playful and whimsical, rhyming and rhythmic, and sensory-focused" is the easiest $\mathbf{u}_i$ for the majority of $\text{LLM}_e$ (5 / 8) with "robotic and emotionless, telegraphic brevity, and legal precision" the easiest for two $\text{LLM}_e$, and one $\text{LLM}_e$ (Qwen2.5 14B Instruct) that performs equally well on the two. Looking at the personas each $\text{LLM}_e$ is least performance on we find that the majority of $\text{LLM}_e$ (5 / 8) are worst on the "inspirational and uplifting, journalistic, and rich descriptions" with "step-by-step instructional, encouraging and supportive, and visual spatial" as the worst for two $\text{LLM}_e$, and one LLM (Qwen2.5 7B Instruct) that performs equally poorly on both personas. We find that while on average the $\text{LLM}_e$ are worse at predicting the human rankings on the helpfulness targeted pairwise ranking task, for all but three $\text{LLM}_e$ there is a persona $\mathbf{u}_i$ for the style-targeted task where the $\text{LLM}_e$ performs worse.

Table 27: **Tweet:** The sensitivity of the $\text{LLM}_e$ to the style guidance provided when completing the pairwise quality evaluation. For each pairwise style ranking prompt $p_e^r$ and $\text{LLM}_e$ pair, the mean and standard error from the mean in F1-score per style guidance description is provided.

| Score | OLMo7B | OLMo13B | Olmo32B | Qwen7B | Qwen14B | Qwen32B | 4o-mini | 4o |
|---|---|---|---|---|---|---|---|---|
| AB | 59($\pm$2.4) | 58($\pm$3.0) | 64($\pm$2.1) | 61($\pm$2.7) | 64($\pm$2.4) | 63($\pm$1.9) | 59($\pm$2.7) | 63($\pm$2.3) |
| Rubric | 42($\pm$3.3) | 45($\pm$1.7) | 45($\pm$4.0) | 48($\pm$2.0) | 48($\pm$2.9) | 52($\pm$2.2) | 56($\pm$3.2) | 53($\pm$1.4) |
| AB - Rubric | 17 | 13 | 19 | 13 | 16 | 11 | 3 | 10 |

**Summary** In Figure 27 and Table 28 the AB $p_e^r$ outperforms the rubric $p_3^r$ across personas $\mathbf{u}$ for the style-targeted task. In general, the AB versus rubric performance gap per $\text{LLM}_e$ (Table 28 bottom row) is similar to the performance gap observed for tweet in Table 27, except the scoring strategy $p_e^r$ has the largest impact on performance for Qwen2.5 14B Instruct. Looking at the standard errors from the mean in Table 28, we can see that the pairwise style ranking performance of the $\text{LLM}_e$ is dependent on the persona $\mathbf{u}_i$. The AB $p_e^r$ scoring strategy leads to pairwise ranking performance that is less sensitive to the specific persona $\mathbf{u}_i$ than the rubric $p_e^r$ with a mean standard error of 2.45 versus 3.0. However, there are some $\text{LLM}_e$ (OLMo2 13B Instruct and Qwen2.5 7B Instruct) that are less sensitive to the persona $\mathbf{u}_i$ given the rubric $p_e^r$.

Examining which personas $\mathbf{u}_i$ are easiest for the $\text{LLM}_e$ to assess compliance with (looking only at the AB $p_e^r$), we find that the "playful and whimsical, rhyming and rhythmic, and sensory-focused" is tied with "robotic and emotionless, telegraphic brevity, and legal precision" as the easiest $\mathbf{u}_i$ for the $\text{LLM}_e$. The $\mathbf{u}_i$ the majority of $\text{LLM}_e$ (4/8) perform worst on is "step-by-step instructional, encouraging and supportive, and visual spatial" with "inspirational and uplifting, journalistic, and rich descriptions" as the worst for one $\text{LLM}_e$ and tied with "step-by-step instructional, encouraging and supportive, and visual spatial" for two $\text{LLM}_e$. We find that for all but one $\text{LLM}_e$ (OLMo2 13B Instruct) performance on the helpfulness-targeted pairwise ranking task is worse than any given persona $\mathbf{u}_i$ for the style-targeted task.

Table 28: **Summary:** The sensitivity of the $\text{LLM}_e$ to the style guidance provided when completing the pairwise quality evaluation. For each pairwise style ranking prompt $p_e^r$ and $\text{LLM}_e$ pair, the mean and standard error from the mean in F1-score per style guidance description is provided.

| Score | OLMo7B | OLMo13B | Olmo32B | Qwen7B | Qwen14B | Qwen32B | 4o-mini | 4o |
|---|---|---|---|---|---|---|---|---|
| AB | 54($\pm$2.8) | 56($\pm$2.9) | 59($\pm$1.9) | 57($\pm$3.7) | 63($\pm$2.8) | 61($\pm$2.0) | 56($\pm$1.7) | 59($\pm$1.6) |
| Rubric | 40($\pm$3.0) | 41($\pm$2.0) | 40($\pm$4.7) | 43($\pm$3.1) | 43($\pm$4.6) | 49($\pm$2.3) | 53($\pm$2.3) | 48($\pm$2.0) |
| AB - Rubric | 14 | 15 | 19 | 14 | 20 | 12 | 3 | 11 |

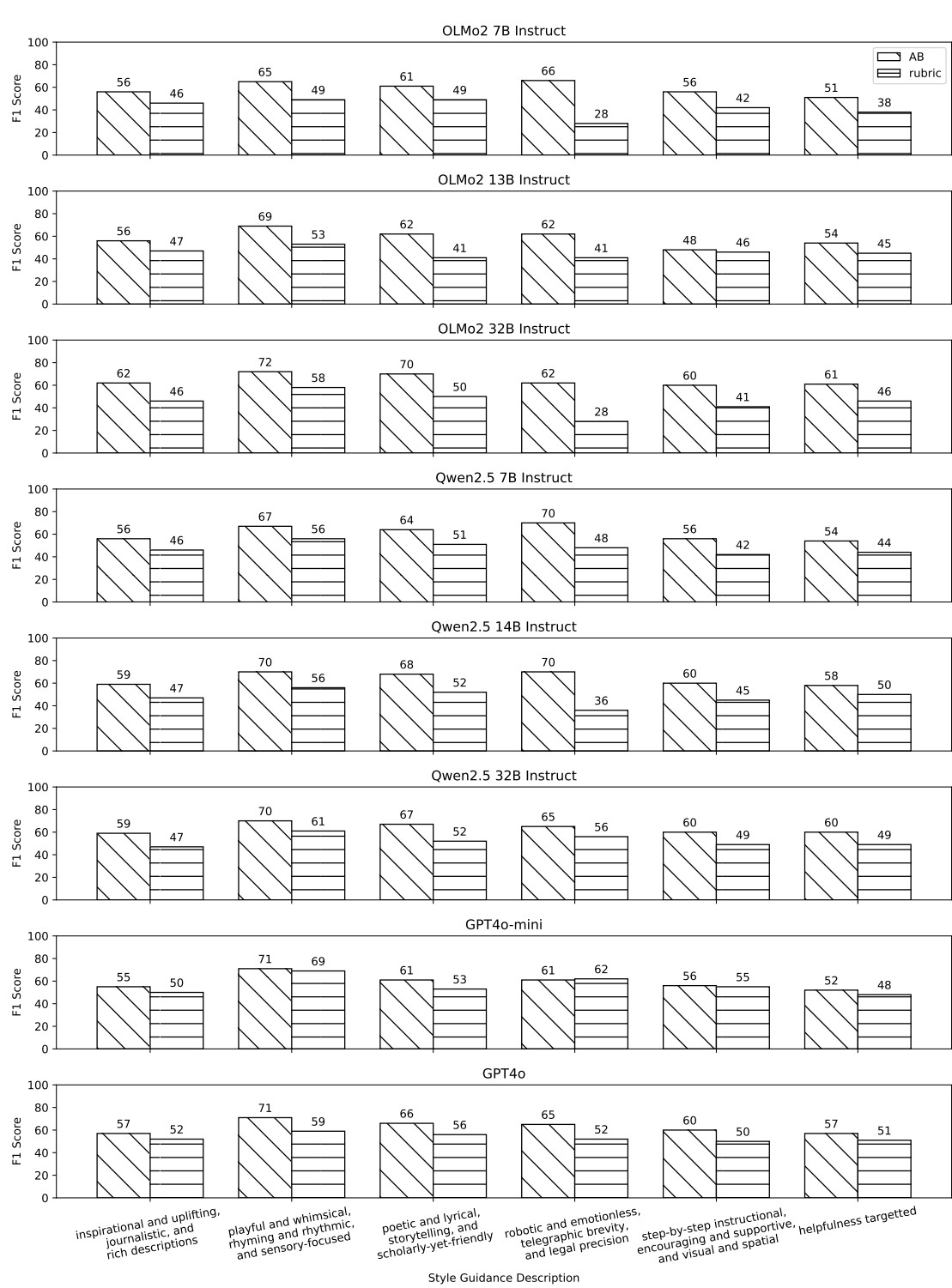

Figure 26: **Tweet:** LLM$_e$ performance on the pairwise style ranking task aggregated across LLM$_g$, generation persona $\mathbf{u}_i$, and writing subtask $q$ (i.e. user query or instruction). Performance is broken down based on the pairwise ranking task, and by the personas $\mathbf{u}_i$ for the style-targeted pairwise ranking task.

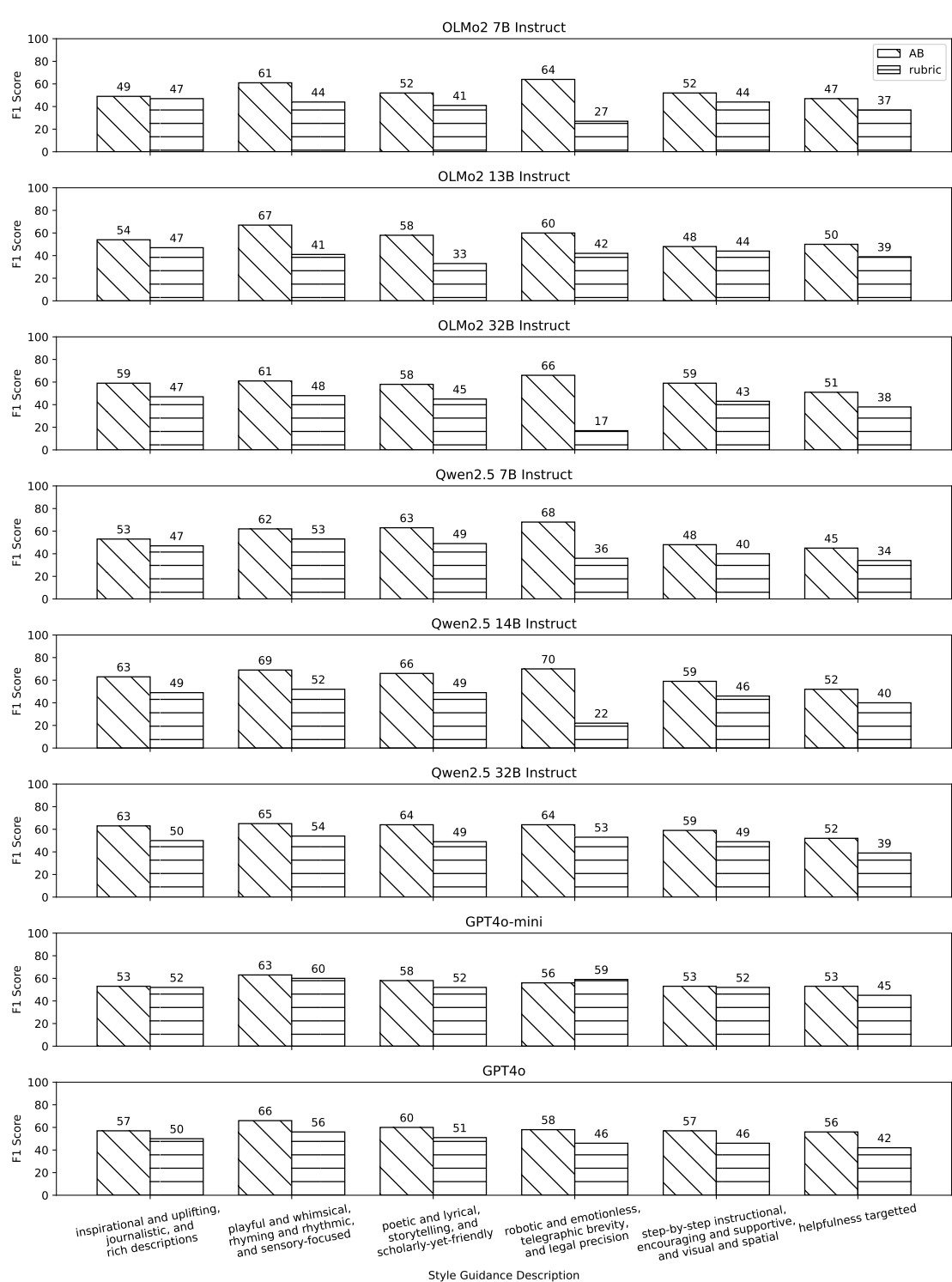

Figure 27: **Summary:** $LLM_e$ performance on the pairwise style ranking task aggregated across $LLM_g$, generation persona $\mathbf{u}_i$, and writing subtask $q$ (i.e. user query or instruction). Performance is broken down based on the pairwise ranking task, and by the personas $\mathbf{u}_i$ for the style-targeted pairwise ranking task.

### L.3 Judge LLM Label Counts by Pairwise Style Ranking Prompt

In this section we look at the frequency of the "a response", "b response", and "tie" labels per $\text{LLM}_e$ by the pairwise style ranking task and the prompt $p_e^r$.

**Tweet** In Figure 28 we can see that 19% of the time humans assign a tie label with completing the helpfulness-targeted pairwise style ranking task, and 25% of the time for the style-targeted task. For both task types, the humans assign a similar proportion of "a response" and "b response" labels (i.e., 0.41 versus 0.4 and 0.38 versus 0.37). The $\text{LLM}_e$ are more likely to assign a "tie" label when using the AB $p_e^r$, and almost never assign "tie" using the rubric $p_e^r$, where GPT4o-mini is the most likely at 7% and 9% of the time, respectively. We attribute the benefits of the AB $p_e^r$ to its increased ability to assign "tie" labels relative to the rubric $p_e^r$.

Looking at the trend in "tie" label frequencies across the $\text{LLM}_e$ by the $\text{LLM}_e$'s general commonsense and reasoning capabilities for the AB $p_e^r$, we see that the weakest of the $\text{LLM}_e$ (i.e., Qwen2.5 7B Instruct, OLMo2 7B Instruct, and OLMo2 13B Instruct with MMLU performance $\leq 75$) are the most likely to assign a "tie" label. This suggests the LLMs are not able to handle the complexity of the labelling prompt leaving their labels sensitive to the ordering of the responses in the prompt. The strongest of the $\text{LLM}_e$ (i.e., Qwen2.5 32B Instruct, GPT4o-mini, and GPT4o with MMLU performance $\geq 82$) are the least likely to assign a "tie" label. This suggests that such models are not well aligned with human annotator inabilities to differentiate between responses. Therefore, the $\text{LLM}_e$ with middling performance (i.e., Qwen2.5 14B Instruct and OLMO2 32B Instruct with MMLU performance of 80 and 77) as the best able to match human pairwise ranking. They are strong enough to not be too sensitive to response order in the prompt, but they do not differentiate between responses at a level of detail that exceeds humans. The similarity in "tie" label frequency between Qwen2.5 14B Instruct and OLMO2 32B Instruct and the human labels aligns with those two $\text{LLM}_e$ having the highest pairwise style ranking performance.

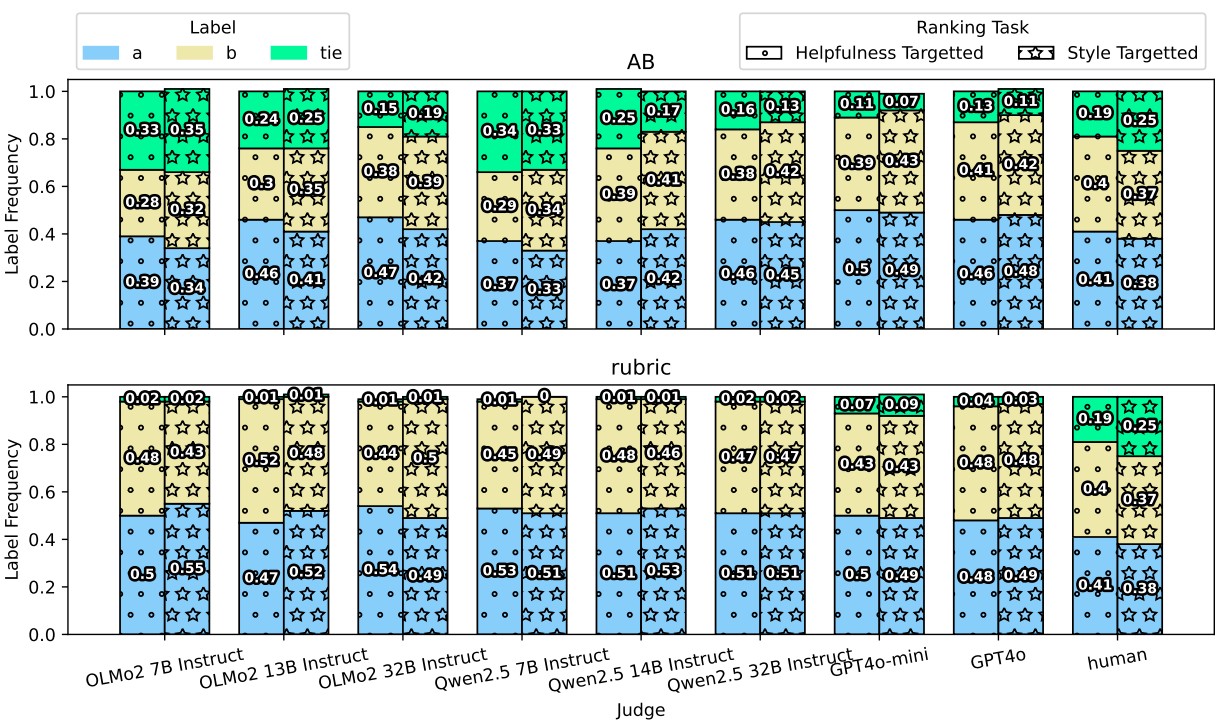

Figure 28: **Tweet:** $\text{LLM}_e$ label counts on the pairwise style ranking task aggregated across $\text{LLM}_g$, generation persona $\mathbf{u}_i$, and writing subtask $q$ (i.e. user query or instruction). Performance is broken down based on the pairwise ranking task, and by the personas $\mathbf{u}_i$ for the style-targeted pairwise ranking task.

To further support this hypothesis, we show the detection scores for $\text{LLM}_e \in \{$ OLMo-2-1124-7B-Instruct and o4-mini-2025-04-16$\}$ when conditioned on each $\mathbf{u}_i$. This is an example of a weak and a strong LLM respectively. These scores are shown in Figure 29. Notice that the weaker model (OLMo) is relatively invariant to the style on which the generator is conditioned and that styles are rarely detected (the image is sparse), whereas the strong model (4o-mini)

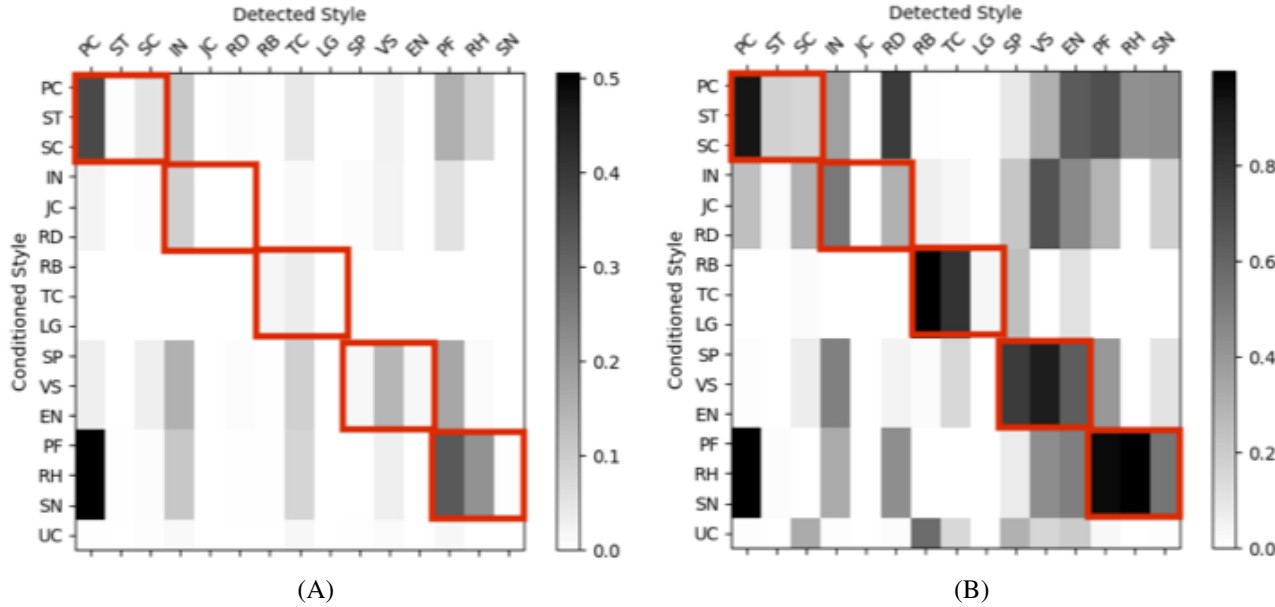

Figure 29: Binary detection scores conditioned on input styles for (A) OLMo-2-1124-7B-Instruct, and (B) o4-mini-2025-04-16 for the tweet task. OLMo-2-1124-7B-Instruct is a weaker model than o4-mini-2025-04-16, and we notice that the scores for OLMo are relatively invariant to the conditioning of the generator. Conversely 4o-mini better detects the styles that the generator is conditioned on. This is one explanation for the high number of ties assigned by weaker models, and low number of ties assigned by the stronger models.

better discriminates styles (the strongest detected styles are alone the block-diagonal, where they should be). This is one explanation for the high number of ties assigned by weaker models, and lower number of ties assigned by the stronger models.

**Summary**   In Figure 30 we can see that 23% of the time humans assign a "tie" label when completing the helpfulness-targeted pairwise ranking task, and 27% of the time when completing the style-targeted task. For both the helpfulness and style-targeted versions of the pairwise ranking task, the humans assign similar proportions of "response a" and "response b" labels with a slight bias towards "response a" labels on the helpfulness-targeted pairwise ranking task (i.e., 0.3 versus 0.46). As with the tweet writing task, the $LM_e$ are more likely to assign a "tie" label using the AB $p_e^r$ and have a maximum 'tie' rate of 5% using the rubric $p_e^r$ (lower than for the tweet task). We attribute the benefits of the AB $p_e^r$ to its increased ability to assign "tie" labels relative to the rubric $p_e^r$.

Looking at the rate of "tie" labels across $LLM_e$ by the $LLM_e$'s general commonsense and reasoning capabilities for the AB $p_e^r$, we see the LLMs most likely to assign a "tie" label (i.e. OLMo2 7B Instruct, OLMo2 13B Instruct, and Qwen2.5 7B Instruct) are have the weakest general commonsense and reasoning capabilities (MMLU performance $\leq 75$). The strongest of the $LLM_e$ (i.e., Qwen2.5 32B Instruct, GPT4o-mini, and GPT4o with MMLU performance $\geq 82$) are the least likely to assign a "tie" label. This suggests that the model's with the strongest general commonsense and reasoning capabilities are not the best aligned with how humans differentiate between two responses, especially on style-targeted ranking tasks.

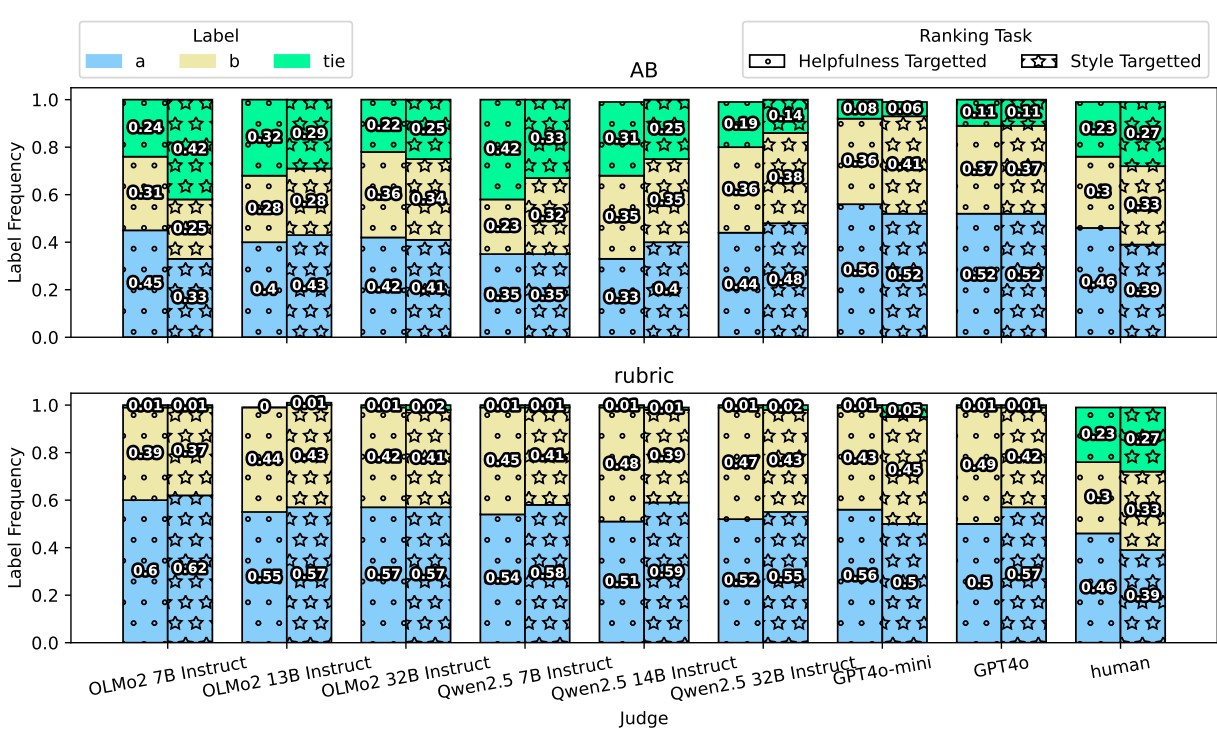

Figure 30: **Summary:** $LLM_e$ label counts on the pairwise style ranking task aggregated across $LLM_g$, generation persona $\mathbf{u}_i$, and writing subtask $q$ (i.e. user query or instruction). Performance is broken down based on the pairwise ranking task, and by the personas $\mathbf{u}_i$ for the style-targeted pairwise ranking task.

### L.4 JUDGE LLM LABEL COUNTS BY PAIRWISE STYLE RANKING PROMPT AND PERSONA $\mathbf{u}_i$

In this section we look at the frequency of the "a response", "b response", and "tie" labels per $\text{LLM}_e$ by the pairwise style ranking task, prompt $p_e^r$ and the persona used in the style-targeted ranking task.

**Tweet**   In Section L.2, we examined Figure 26 to identify that (looking only at the AB $p_e^r$), the "playful and whimsical, rhyming and rhythmic, and sensory-focused" is the easiest $\mathbf{u}_i$ for the majority of $\text{LLM}_e$ (5 / 8) to complete the pairwise ranking task with "robotic and emotionless, telegraphic brevity, and legal precision" the easiest for two $\text{LLM}_e$, and one $\text{LLM}_e$ (Qwen2.5 14B Instruct) that performs equally well on the two personas. The "inspirational and uplifting, journalistic, and rich descriptions" with "step-by-step instructional, encouraging and supportive, and visual spatial" personas were the most difficult for the $\text{LLM}_e$ to solve the pairwise ranking task for.

In Figure 31 we look at the label counts for each $\text{LLM}_e$ and the human annotators by pairwise ranking task (helpfulness versus style targeted) and the personas $\mathbf{u}_i$ for the style-targeted task. We find that for the human annotators, the proportion of each label is relatively consistent across style-targeted $\mathbf{u}_i$ ranging from 19% to 27% for "ties". The human annotators do assign the fewest number of "tie" labels on the helpfulness-targeted task and the "playful and whimsical, rhyming and rhythmic, and sensory-focused" style-targeted $\mathbf{u}_i$ pairwise ranking tasks. The "robotic and emotionless, telegraphic brevity, and legal precision" and "step-by-step instructional, encouraging and supportive, and visual spatial" style-targeted $\mathbf{u}_i$ received the highest number of "tie" labels. As "playful and whimsical, rhyming and rhythmic, and sensory-focused" and "robotic and emotionless, telegraphic brevity, and legal precision" are the style-targeted $\mathbf{u}_i$ the most $\text{LLM}_e$ performed best on, and they are split between receiving the least and the most "tie" labels from humans. This suggests that $\text{LLM}_e$ performance is not largely driven by the presence of "tie" labels that could indicate the task is harder for humans to solve. However, the "inspirational and uplifting, journalistic, and rich descriptions" and "step-by-step instructional, encouraging and supportive, and visual spatial", which were the most difficult $\mathbf{u}_i$ for the $\text{LLM}_e$, either match or nearly match the maximum number of human-annotator "tie" labels.

**Summary**   In Section L.2 we examined Figure 27 to identify that (looking only at AB $p_e^r$), the "playful and whimsical, rhyming and rhythmic, and sensory-focused" and "robotic and emotionless, telegraphic brevity, and legal precision" tied as the easiest personas $\mathbf{u}_i$ for the majority of $\text{LLM}_e$ to rank the response pair $(r_q^a, r_q^b)$. As with the tweet task, the "inspirational and uplifting, journalistic, and rich descriptions" and "step-by-step instructional, encouraging and supportive, and visual spatial" were the most difficult $\mathbf{u}_i$ for the $\text{LLM}_e$.

The label frequencies for the $\text{LLM}_e$ and the human annotators for each pairwise ranking task and persona $\mathbf{u}_i$ for the style-targeted task are provided in Figure 32. For the human annotators, the frequency of the "response a", "response b", and "tie" labels is roughly consistent across personas $\mathbf{u}_i$ with three of the $\mathbf{u}_i$ having "tie" rates of 0.22 or 0.24, and two of 0.34 and 0.32. The "tie" rate for the helpfulness-targeted pairwise ranking task is not the lowest, but is on the lower end at 0.23. There is no clear trend in the human "tie" rate between the personas $\mathbf{u}_i$ the $\text{LLM}_e$ performed best versus worst on. For example, "playful and whimsical, rhyming and rhythmic, and sensory-focused" received some the fewest "tie" labels, while "robotic and emotionless, telegraphic brevity, and legal precision" received some of the most. As with tweets, the there is not a strong connection between human "tie" rate and $\text{LLM}_e$ pairwise ranking performance.

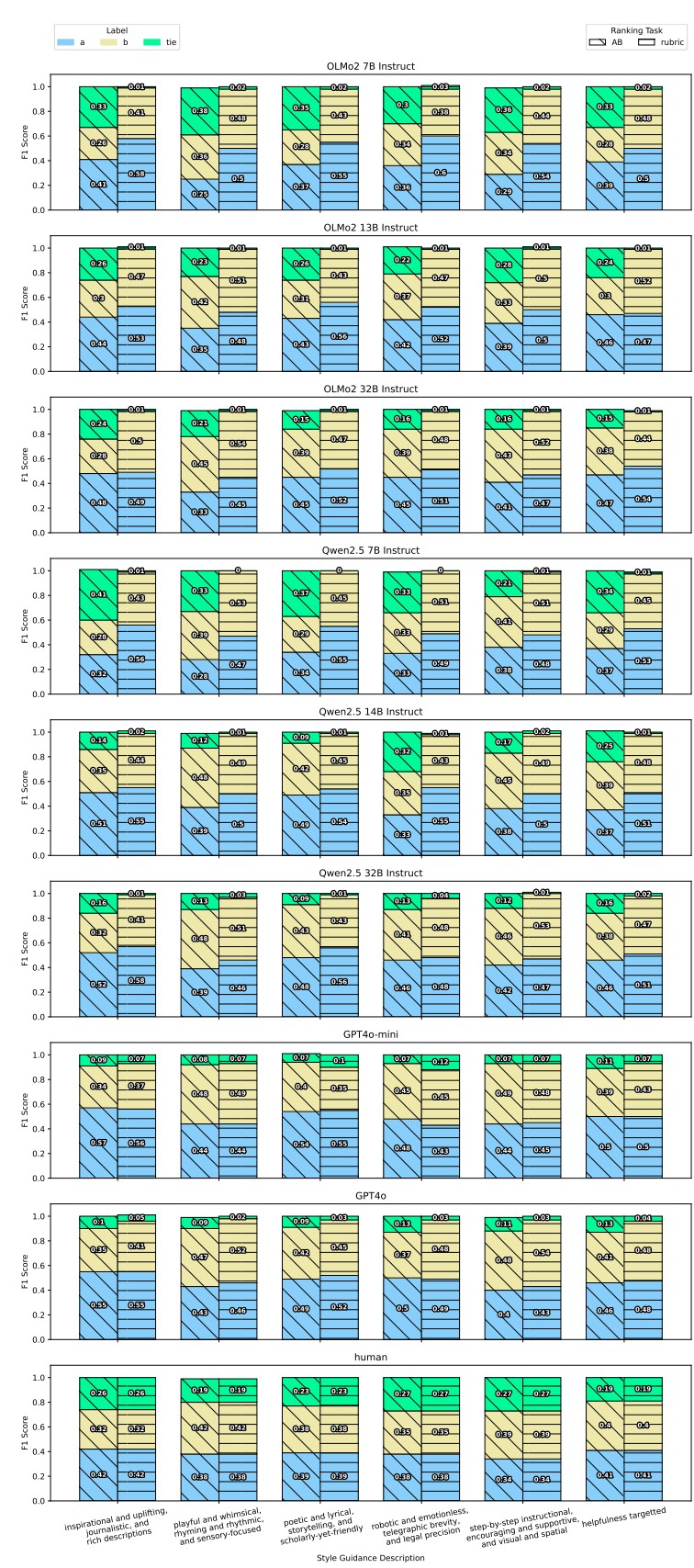

Figure 31: **Tweet:** LLM$_e$ label counts on the AB and rubric pairwise style ranking tasks aggregated across LLM$_g$, generation persona $\mathbf{u}_i$, and writing subtask $q$ (i.e. user query or instruction). Performance is broken down based on the pairwise ranking task (helpfulness targeted and style targeted), and by the personas $\mathbf{u}_i$ for the style-targeted pairwise ranking task.

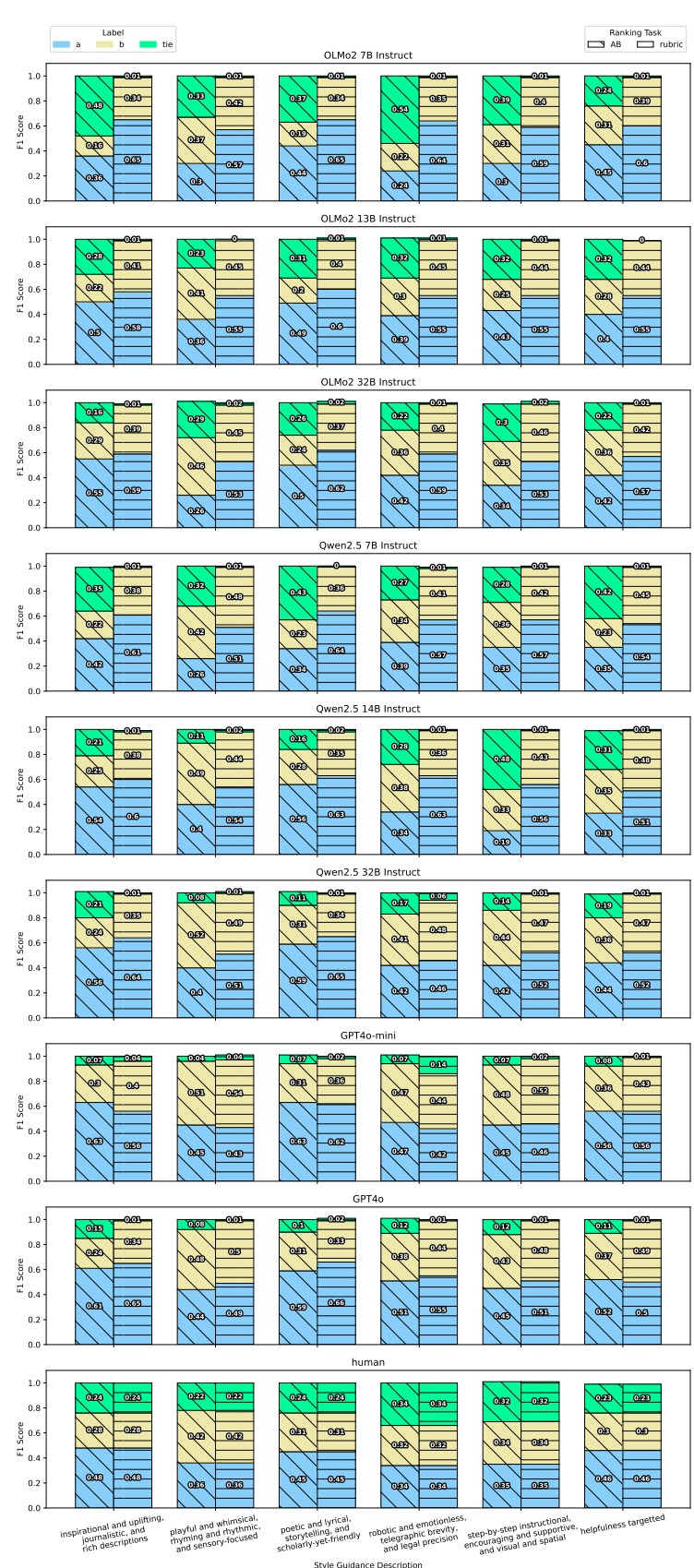

Figure 32: **Summary:** $LLM_e$ label counts on the AB and rubric pairwise style ranking tasks aggregated across $LLM_g$, generation persona $\mathbf{u}_i$, and writing subtask $q$ (i.e. user query or instruction). Performance is broken down based on the pairwise ranking task (helpfulness targeted and style targeted), and by the personas $\mathbf{u}_i$ for the style-targeted pairwise ranking task.

## L.5 Judge LLM Pairwise Ranking Performance by LLM$_g$ and LLM$_e$ Relationship – Detailed Results

In this section we provide detailed results for the impact the relationship between the LLM$_g$ and LLM$_e$ has on the pairwise ranking performance of the LLM$_e$.

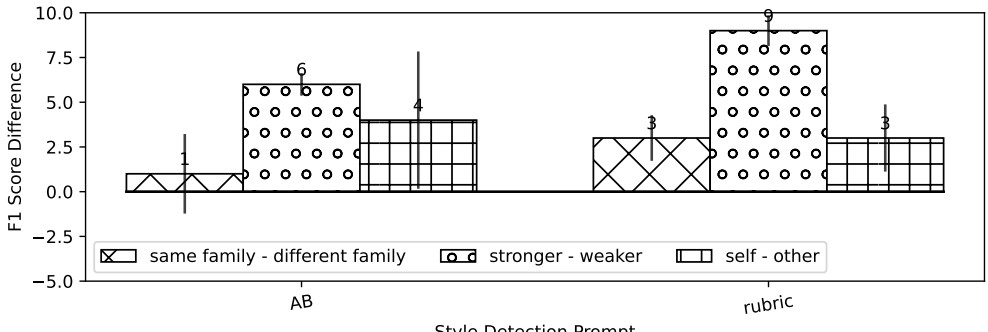

(a) Aggregated of Pairwise Ranking Task

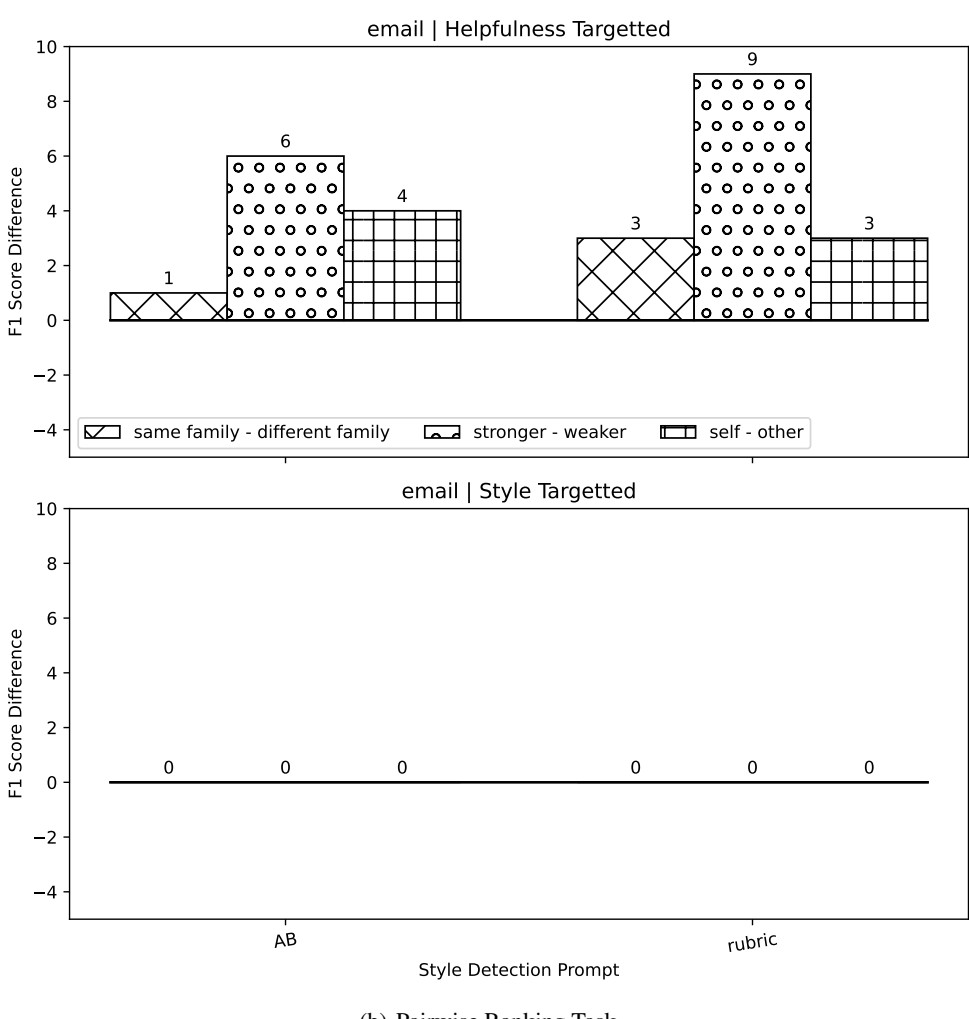

(b) Pairwise Ranking Task

Figure 33: **Email:** The LLM$_e$ performance as a function of the relationship between the LLM$_e$ and the LLM$_g$.

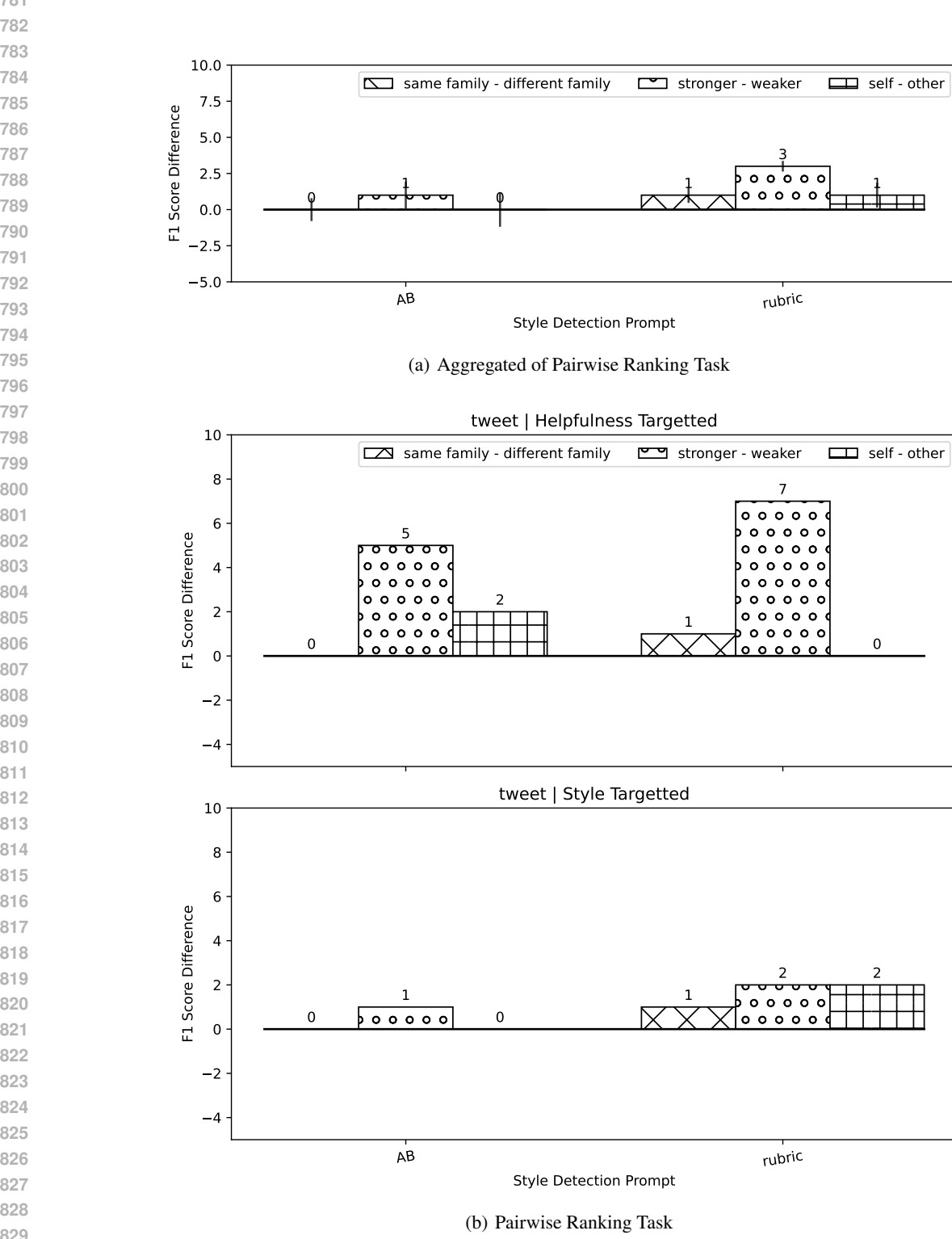

(a) Aggregated of Pairwise Ranking Task

(b) Pairwise Ranking Task

Figure 34: **Tweet:** The $\text{LLM}_e$ performance as a function of the relationship between the $\text{LLM}_e$ and the $\text{LLM}_g$.

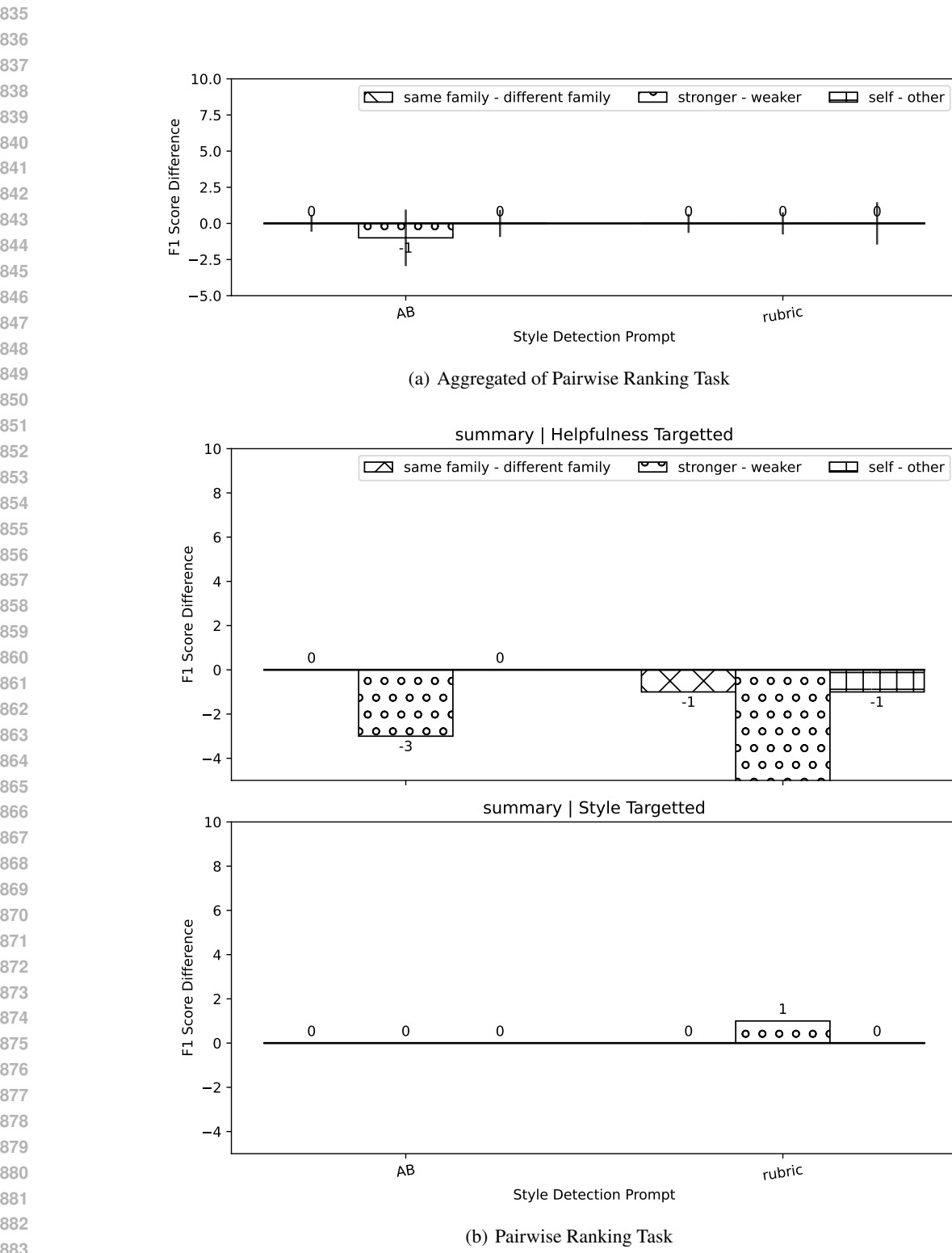

(a) Aggregated of Pairwise Ranking Task

(b) Pairwise Ranking Task

Figure 35: **Summary:** The LLM$_e$ performance as a function of the relationship between the LLM$_e$ and the LLM$_g$.

