# OpenReview forum: "Evaluating the Evaluators: Investigating LLM Judges for Personalized Writing Style Assessment"
_ICLR.cc/2026/Conference — ICLR 2026 Conference Withdrawn Submission_

### Official Review · Reviewer_MmLU · 2025-10-27

**Soundness:** 3
**Presentation:** 3
**Contribution:** 2
**Rating:** 4
**Confidence:** 4

**Summary:**

The submission evaluates through a set of experiments whether LLMs can be used to evaluate the presence of certain writing styles in generated text. The authors first compiled a large-scale human annotation dataset for 15 writing styles, and then evaluated different prediction protocols with LLMs, with consistent findings across evaluated LLMs such as binary working better than likert or probability prediction. That LLMs are mostly consitent in their ratings, and that the most performant LLMs tend to "overthink" their response, leading to less correlation with users.

**Strengths:**

- The large-scale data annotation could prove useful to others
- The experiments are thorough, involving multiple domains and many LLMs, with findings that might be useful to the community.

**Weaknesses:**

- The scope of the work is limited due to a restrictive choice of 5 writing styles which are compounds of a total of 15 individual styles. It is not clear how often a real user would require from an LLM to write in the style of "robotic and emotionless and telegraphic brevity and legal precision". It is not entirely clear how these were selected. In general, these styles seem more concrete in that they could be evaluated based on some structural elements, but not all styles in the wild are this way. The claims of the paper should therefore be more limited to such writing styles that are easy to describe and more structural in nature. Imagine instead if a user asks to write in the style of Pablo Neruda. Would the same methodology be able to be applied?
- One of the main findings of the work, namely that more powerful LLMs (in terms of reasoning/instruction following) are worse at stylistic judgement because it is more detail-oriented than the annotators is only one possible interpretation of the result. Another way to interpret it is more negative: perhaps the annotators that participated in the data collection were not given enough time to review in-depth the text they were judging, leading to the annotation based on shallow signals. In turn, an LLM capable of looking beyond the shallow structural signal does not correlate well with the high-level label from the annotator. There would need to be clarification on how much time annotators spent reading each annotate sample, and if they were compensated accordingly for this time. If the dataset is mostly focused on easy-to-annotate surface-level style, why is that of importance?

**Questions:**

- Please see the questions in Weaknesses section.
- It's unclear what the ceiling of performance on the prediction task is. Since the annotators reach moderate-ish agreement on the various styles, it would be interesting to know what the performance ceiling is for models, and how far the best-performing LLM is from such a performance.

---

### Official Review · Reviewer_spdk · 2025-10-31

**Soundness:** 2
**Presentation:** 2
**Contribution:** 3
**Rating:** 0
**Confidence:** 3

**Summary:**

The paper studies how well LLMs can judge writing style when generators are prompted with personas. It evaluates eight judge models on two tasks style detection (is a style present?) and pairwise style ranking (which of two outputs better matches a persona) across three writing tasks (emails, tweets, summaries) and text produced by four generators. The authors collect a sizable dataset. Key findings: (i) judge performance correlates strongly with general LLM ability for style detection; (ii) task matters (ex: email is easiest); (iii) A/B comparison beats rubric scoring for pairwise ranking and (iv) for detection, strongest LLMs are best, but for ranking, mid-strength models can outperform the very strongest, which over-attend to distinctions humans ignore.

**Strengths:**

- Comprehensive, carefully designed evaluation spanning models, tasks, scoring schemes, and generator–judge relationships; large human-labeled dataset with repeated annotations.
- Clear actionable takeaways (use strongest LLM for detection; prefer A/B for ranking; use a stronger judge than the generator).
- Insightful analysis of why very strong judges can be worse rankers (focus on differences humans deem irrelevant) and of judge self-consistency across prompts.

**Weaknesses:**

- **It appears that the authors have manipulated the ICLR template. The margins on the sides are smaller in this submission.** This has allowed them to have more space than other papers and breaches the author instructions. In my opinion, this should result in a desk rejection.

- Evaluator leakage risk: The same LLM families appear as generators and judges. Although cross-family analyses are included, stronger isolation (e.g. disjoint families, adversarial formatting controls) would reduce bias.

- Rubric ranking underperforms A/B and rarely outputs "tie" suggesting rubric prompts (and tie handling) may be mis-calibrated. More robust rubric designs could change conclusions.

- Human agreement on style is only fair/moderate (~0.2–0.6), and 3-point labels are binarized for detection. Both choices may compress nuance. Significance tests and per-style variance reporting could be stronger.

**Questions:**

- Can the authors report per-style F1 with confidence intervals, and run bootstrap tests to compare scoring schemes?

- For ranking can the authors add a calibrated rubric (explicit tie prior, ordinal anchors) and report how often each judge uses "tie" vs humans?

- Can the authors evaluate each generator exclusively with judges from other families, and include an unseen-family judge?

---

### Official Review · Reviewer_8fCw · 2025-10-31

**Soundness:** 3
**Presentation:** 3
**Contribution:** 2
**Rating:** 4
**Confidence:** 4

**Summary:**

This paper studies how well LLMs can act as evaluators for assessing writing style alignment in personalized text generation. The authors build a large benchmark with over 350k human and LLM annotations across three writing tasks (emails, tweets, summaries) and two evaluation types: style detection (identifying if a target style is present) and pairwise style ranking. They test eight evaluator models and find that style detection accuracy correlates strongly with general LLM ability. However, for pairwise ranking, mid-sized models perform better than the strongest ones, since very capable models attend to details humans overlook. The study concludes that stronger models are ideal for classification-style detection, while moderate-strength models align better with human judgments in comparative evaluation.

**Strengths:**

- Addresses a relevant problem in LLM evaluation: whether LLMs can serve as reliables judges for subjective, stylistic dimensions of writing.
- Collects a big corpus of annotations that is hopefully being published and made available. But also here the long-term benefit of the LLM-written annotations remain limited as new LLMs are released often.
- Insightful analysis with recommendations when to use what models. However, it is still questionable how these results would generalize to new LLMs.
- Extensive appendix with a lot of details

**Weaknesses:**

- Limited impact as the paper just evaluates LLM-written text with LLMs and one of the main takeaways is that the strongest LLM has also the best results (with the exception of comparative evaluations).
- Unclear statements already in the abstract (line 21). Highest by 28%? (Line 22)
- Table 1 inter-annotator agreement is rather low questioning the significance of the results
- Limited task diversity. It would be interesting to see results on more advanced tasks like story writing, essays,…
- Relevant related work missing:
	- Ostheimer, P., Nagda, M., Kloft, M., & Fellenz, S. (2023). Text style transfer evaluation using large language models.
	- Chiang, C. H., & Lee, H. Y. (2023). Can large language models be an alternative to human evaluations?

**Questions:**

See weaknesses above:
- Why is inter-annotator agreement so low and do you think this is a problem?
- Have you tried to evaluate additional tasks?

**Details Of Ethics Concerns:**

The template was violated (margins are smaller than allowed)

---

### Official Review · Reviewer_9AYF · 2025-11-02

**Soundness:** 2
**Presentation:** 2
**Contribution:** 2
**Rating:** 2
**Confidence:** 5

**Summary:**

In this paper authors consider the problem of determining how well large language models (LLMs) are able to judge LLM-generated text when a generator is prompted to align with a specific writing style.They evaluate performance on two judge tasks: style detection and style
quality pairwise ranking.They collected human style detection and pairwise ranking labels on text generated from four models for three generation tasks (email, tweet, and summary writing) and find that judge quality correlates strongly with general LLM ability
measured using MMLU and  varies by writing task. They also find that LLM evaluators are more consistent and reliable when using AB comparisons rather than rubric-based scoring for style ranking. Finally, they find that for style detection, using the LLM with the strongest general capabilities is best, however this is not true for style quality pairwise ranking

**Strengths:**

Even though I am not sure what to make of the low human agreement for the data , I assume the data can be useful
For language models to understand and detect style while subjective is a challenging task which authors tackle here

**Weaknesses:**

I am struggling a bit in terms of why this is an ICLR submission. The insights are pretty generic and isnt something catered to the style task. I assume this kind of idiosyncratic behavior of LLM as judges will hold for other tasks so from that perspective its very narrow and not of broader relevance.

The way style is defined and for the task its pretty narrow and will have likely very little impact outside the scope of the publication

Where do you get these 15 style types from ? Its pretty adhoc. There are four main types of writing: expository, descriptive, persuasive, and narrative afaik.

The agreement for half the styles are very low making it unreliable. If ground truth is noisy whole experiment becomes pointless

The fact that  judge performance varied significantly by the writing task (e.g., performance was highest for email) suggests the findings (like which prompt is best) are highly task-dependent and may not generalize to other formats

No discussion on OOD style. What about literary style ?

**Questions:**

NA

---

### Author Response · Authors · 2025-12-01
**Withdrawal of Paper**

We thank the reviewers for their feedback. We have learned about how about we need to better tell story around the motivation for this work.

As Reviewer spdk identified, the margin in the paper was modified. We spent a large chunk of time trying to figure out how this happened. None of the authors modified the ICLR 2026 template. In formatting the prompts in the appendix we accidentally changed the paper margins. Due to this mistake, we are withdrawing the paper.

---

### Note · Authors · 2025-12-01

**Comment:**

We are withdrawing the paper due to accidentally changing the margin when formatting our LLM prompts in the appendix.

**Withdrawal Confirmation:**

I have read and agree with the venue's withdrawal policy on behalf of myself and my co-authors.